# Dynamics of co-substrate pools can constrain and regulate metabolic fluxes

Robert West[1†], Hadrien Delattre[1†], Elad Noor[2], Elisenda Feliu[3]*, Orkun S Soyer[1]*

[1]School of Life Sciences, University of Warwick, Warwick, United Kingdom; [2]Department of Plant and Environmental Sciences, Weizmann Institute of Science, Rehovot, Israel; [3]Department of Mathematics, University of Copenhagen, Copenhagen, Denmark

**Abstract** Cycling of co-substrates, whereby a metabolite is converted among alternate forms via different reactions, is ubiquitous in metabolism. Several cycled co-substrates are well known as energy and electron carriers (e.g. ATP and NAD(P)H), but there are also other metabolites that act as cycled co-substrates in different parts of central metabolism. Here, we develop a mathematical framework to analyse the effect of co-substrate cycling on metabolic flux. In the cases of a single reaction and linear pathways, we find that co-substrate cycling imposes an additional flux limit on a reaction, distinct to the limit imposed by the kinetics of the primary enzyme catalysing that reaction. Using analytical methods, we show that this additional limit is a function of the total pool size and turnover rate of the cycled co-substrate. Expanding from this insight and using simulations, we show that regulation of these two parameters can allow regulation of flux dynamics in branched and coupled pathways. To support these theoretical insights, we analysed existing flux measurements and enzyme levels from the central carbon metabolism and identified several reactions that could be limited by the dynamics of co-substrate cycling. We discuss how the limitations imposed by co-substrate cycling provide experimentally testable hypotheses on specific metabolic phenotypes. We conclude that measuring and controlling co-substrate dynamics is crucial for understanding and engineering metabolic fluxes in cells.

**\*For correspondence:**
efeliu@math.ku.dk (EF);
O.Soyer@warwick.ac.uk (OSS)

[†]These authors contributed equally to this work

**Competing interest:** The authors declare that no competing interests exist.

## Editor's evaluation

This manuscript presents an important mathematical analysis of metabolic "co-substrates" and how their cycling can affect metabolic fluxes. Through mathematical analysis of simple network motifs, it shows the impact of co-substrate cycling on constraining metabolic fluxes. The combination of mathematical modeling and comparisons with existing data from previous studies offers convincing support for the potential biological relevance of co-substrate cycling. The work will be of interest to researchers who study microbial metabolism and metabolic engineering.

## Introduction

Dynamics of cell metabolism directly influences individual and population-level cellular responses. Examples include metabolic oscillations underpinning the cell cycle (*Papagiannakis et al., 2017*; *Murray et al., 2007*) and metabolic shifts from respiration to fermentation, as observed in cancer phenotypes (*Warburg, 1956*; *Diaz-Ruiz et al., 2009*; *Carmona-Fontaine et al., 2013*) and cell-to-cell cross-feeding (*Ponomarova et al., 2017*; *Campbell et al., 2015*; *Großkopf et al., 2016*). Predicting or conceptualising these physiological responses using dynamical models is difficult due to the large size and high connectivity of cellular metabolism. Despite this complexity, however, it is possible that

**eLife digest** Metabolism powers individual cells and ultimately the body. It comprises a sequence of chemical reactions that cells use to break down substances and generate energy. These reactions are catalyzed by enzymes, which are proteins that speed up the rate of the reaction. Many reactions also involve co-substrates, which are themselves transformed by individual reactions but are eventually converted back into their original form in a series of steps. This process is known as co-substrate cycling.

Scientists have long been interested in understanding what controls the rate at which metabolic reactions and metabolic pathways convert a substance into a final product. This is a difficult subject to study because of the complexity of the metabolic pathways, with their branched, linear or coupled structures. In the past, researchers have looked at the influence of enzymes on the rate of a metabolic pathway, but less has been known about the effect of co-substrate cycling.

To find out more, West, Delattre et al. developed a series of mathematical models to describe different types of metabolic pathways in terms of the number of metabolites that enter and leave it, including the influence of co-substrates.

They found that co-substrate cycling, when involved in a metabolic reaction, limits the speed with which the reaction happens. This is distinct from the limit that enzymes impose on the speed of the reaction. It depends on the total amount of co-substrates in the cell: changing the number of co-substrates in the cell influences the speed at which the metabolic reaction takes place.

This study has increased our understanding of how metabolic pathways work, and what controls the speed at which reactions take place. It opens up a new potential method for explaining how cells control metabolic reaction rates and how metabolic substrates can be directed across different pathways. This research is likely to inspire future research into the influence of co-substrates in different cell types and conditions.

cellular metabolism features certain 'design principles' that determine the overall dynamics. There is ongoing interest in finding such simplifying principles.

A key concept for understanding the dynamics of any metabolic system is that of 'reaction flux', which is a measure of the rate of biochemical conversion in a given reaction. To identify possible limitations on reaction fluxes, early studies focused on linear pathways involving ATP production and studied their dynamics under the optimality assumption of maximisation of overall pathway flux under limited enzyme levels available to the pathway (*Heinrich et al., 1991*). The resulting theory predicted a trade-off between pathway flux vs. yield (i.e. rate of ATP generation vs. amount of ATP generated per metabolite consumed by the pathway) in linear pathways (*Heinrich and Hoffmann, 1991*). This theory is subsequently used to explain the emergence of different metabolic phenotypes (*Pfeiffer et al., 2001*). In related studies, models pertaining to flux optimisation and enzyme levels being a key limitation are used to explain the structure of different metabolic pathways (*Flamholz et al., 2013*), and the metabolic shifting from respiration to fermentative pathways under increasing glycolysis rates (*Großkopf et al., 2016*; *Basan et al., 2015*; *Majewski and Domach, 1990*). There are, however, increasing number of studies suggesting that enzyme levels alone might not be sufficient to explain observed flux levels. For example, it was shown that the *maximal* value of the apparent activities ($k_{app}^{max}$) of an enzyme, derived using measured enzyme levels and fluxes under different conditions, was a good estimate for the specific activity of that enzyme *in vitro* ($k_{cat}$) (*Davidi et al., 2016*). However, individual estimates from each condition (i.e. individual $k_{app}$ values) were commonly lower than the specific activity – suggesting that the flux is limited by something other than enzyme levels under those conditions. Other studies have shown that metabolic flux changes, caused by perturbations in media conditions, are not explained solely by changes in expression levels of enzymes (*Chubukov et al., 2013*; *Gerosa et al., 2015*).

Another conceptual framework emphasized the importance of cyclic reaction motifs, particularly those involving so-called co-substrate pairs, such as ATP / ADP or NAD(P)H / NAD(P)+, as a key to understanding metabolic system dynamics (*Reich and Sel'kov, 1981*). This framework is linked to the idea of considering the supply and demand structures around specific metabolites as regulatory blocks within metabolism (*Hofmeyr and Cornish-Bowden, 2000*). For example, the total pool of ATP

and its derivates (the 'energy charge') is suggested as a key determinant of physiological cell states (*Atkinson, 1968*). Inspired by these ideas, theoretical studies have shown that metabolic systems featuring metabolite cycling together with allosteric regulation can introduce switch-like and bistable dynamics (*Okamoto and Hayashi, 1983*; *Hervagault and Cimino, 1989*), and that metabolite cycling motifs introduce total co-substrate level as an additional control element in metabolic control analysis (*Hofmeyr et al., 1986*; *Sauro, 1994*). Specific analyses of ATP cycling in the glycolysis pathway, sometimes referred to as a 'turbo-design', and metabolite cycling with autocatalysis, as seen for example in the glyoxylate cycle, have shown that these features constrain pathway fluxes (*Koebmann et al., 2002*; *Teusink et al., 1998*; *van Heerden et al., 2014*; *Hatakeyama and Furusawa, 2017*; *Barenholz et al., 2017*; *Kurata, 2019*). Taken together, these studies indicate that metabolite cycling, in general, and co-substrate cycling specifically, could provide a key 'design feature' in cell metabolism, imposing certain constraints or dynamical properties to it.

Towards better understanding the role of co-substrate cycling in cell metabolism dynamics, we undertook here an analytical and simulation-based mathematical study together with analyses of measured fluxes. We created models of enzymatic reaction systems featuring co-substrate cycling, abstracted from real metabolic systems such as glycolysis, nitrogen-assimilation, and central carbon metabolism. We found that co-substrate cycling introduces a fundamental constraint on reaction flux. In the case of single reaction and short linear pathways, we were able to derive a mathematical expression of the constraint, showing that it relates to the pool size and turnover rate of the co-substrate. Analysing measured fluxes, we find that several of the co-substrate featuring reactions in central carbon metabolism carry lower fluxes than expected from the kinetics of their primary enzymes, suggesting that these reactions might be limited by co-substrate cycling. In addition to its possible constraining role, we show that co-substrate cycling can also act as a regulatory element, where control of co-substrate pool size can allow control of flux dynamics across connected or branching pathways. Together, these findings show that co-substrate cycling can act both as a constraint and a regulatory element in cellular metabolism. The resulting theory provides testable hypotheses on how to manipulate metabolic fluxes and cell physiology through the control of co-substrate pool sizes and turnover dynamics, and can be expanded to explain dynamic measurements of metabolite concentrations in different perturbation experiments.

## Results

### Co-substrate cycling represents a ubiquitous motif in metabolism with co-substrate pools acting as 'conserved moieties'

Certain pairs of metabolites can be interconverted via different reactions in the cell, thereby resulting in their 'cycling'. This cycling creates interconnections within metabolism, spanning either multiple reactions in a single, linear pathway, or multiple pathways that are independent or are branching from common metabolites. For example, in glycolysis, ATP is consumed in reactions mediated by the enzymes glucose hexokinase and phosphofructokinase, and is produced by the downstream reactions mediated by phosphoglycerate and pyruvate kinase (*Appendix 1—figure 1A*). In the nitrogen assimilation pathway, the $NAD^+$ / NADH pair is cycled by the enzymes glutamine oxoglutarate aminotransferase and glutamate dehydrogenase (Appendix Dynamics of co-substrate pools can constrain and regulate metabolic fluxes - *Appendix 1—figure 1B*). Many other cycling motifs can be identified, involving either metabolites from the central carbon metabolism or metabolites that are usually referred to as co-substrates. Examples for the latter include NADPH, FADH2, GTP, and Acetyl-CoA and their corresponding alternate forms, while examples for the former include the tetrahydrofolate (THF) / 5,10-Methylene-THF and glutamate / $\alpha$-ketoglutarate (akg) pairs involved in one-carbon transfer and in amino acid biosynthesis pathways, respectively (*Appendix 1—figure 1C and D*). For some of these metabolites, their cycling can connect many reactions in the metabolic network. Taking ATP (NADH) as an example, there are 265 (118) and 833 (601) reactions linked to the cycling of this metabolite in the genome-scale metabolic models of *Escherichia coli* and human respectively models iJO1366 (*Orth et al., 2011*) and Recon3d (*Brunk et al., 2018*).

We notice here that many of the co-substrate involving cycling reactions can be abstracted as a simplified motif as shown in (*Figure 1A*). This abstract representation highlights the fact that the total pool-size involving all the different forms of a cycled metabolite can become a conserved quantity.

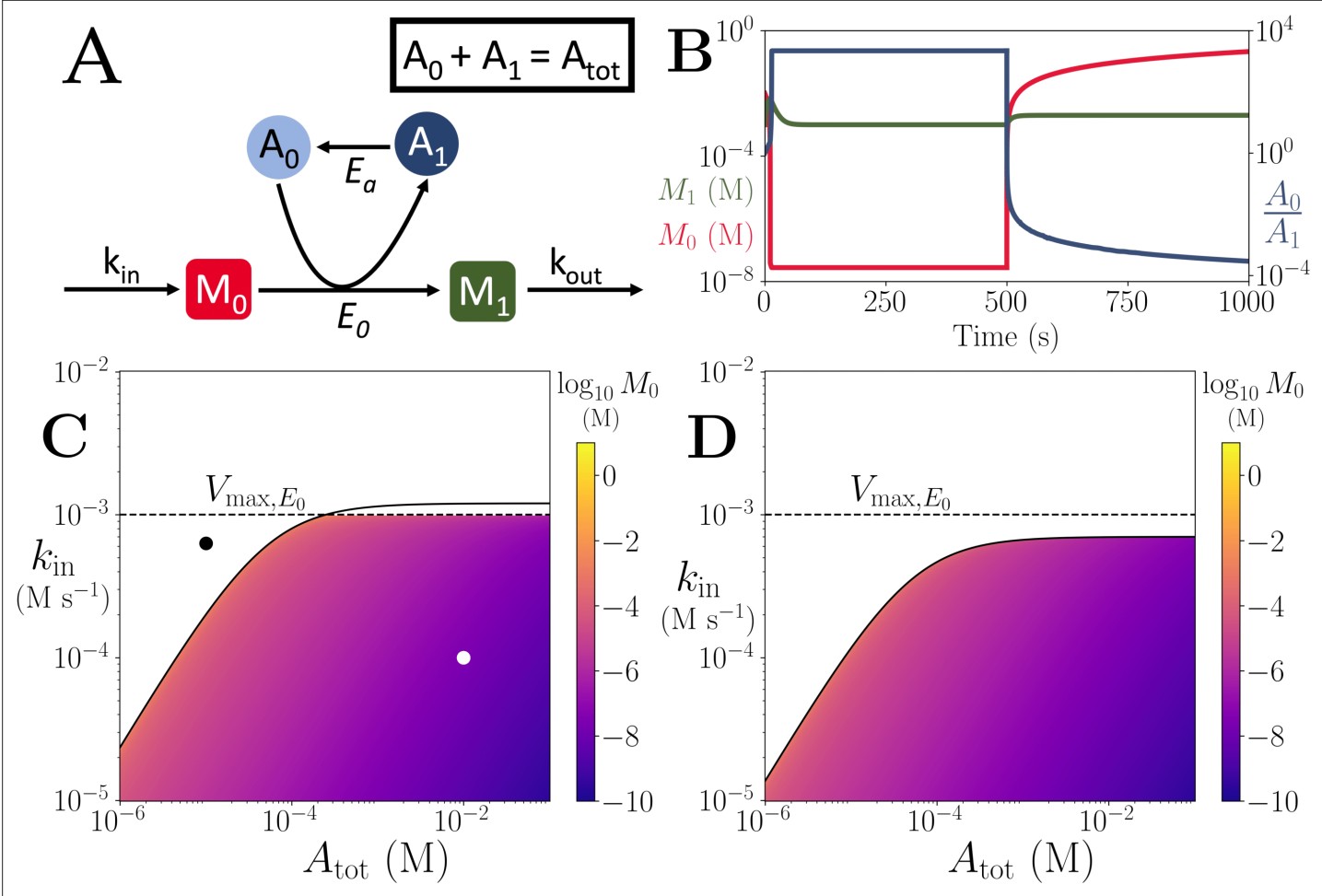

**Figure 1.** Motif, time-series and threshold in a single co-substrate involving reaction. (**A**) Cartoon representation of a single irreversible reaction with co-substrate cycling (see Appendices for other reaction schemes). The co-substrate is considered to have two forms $A_0$ and $A_1$. (**B**) Concentrations of the metabolites $M_0$ (red) and $M_1$ (green), and the $A_0/A_1$ ratio (blue) are shown as a function of time. At $t = 500$, the parameters are switched from the white dot in panel (**C**) (where a steady state exists) to the black dot (where we see continual build-up of $M_0$ and decline of $A_0$ without steady state). (C & D) Heatmap of the steady state concentration of $M_0$ as a function of the total co-substrate pool size ($A_{\text{tot}}$) and inflow flux ($k_{in}$). White area shows the region where there is no steady state. On both panels, the dashed line indicates the limitation from the primary enzyme, $k_{in} < V_{\max,E_0}$, and the solid line indicates the limitation from co-substrate cycling, $k_{in} < A_{\text{tot}} V_{\max,E_a}/(K_{M,E_a} + A_{\text{tot}})$. In panel (**C**), there is a range of $A_{\text{tot}}$ values for which the first limitation is more severe than the second. In contrast, in panel (**D**), the second limitation is always more severe than the first. In (B & C) the parameters used for the primary enzyme (for the reaction converting $M_0$ into $M_1$) are picked from within a physiological range (see *Supplementary file 1*) and are set to: $E_{\text{tot}} = 0.01$ mM, $k_{cat} = 100/s$, $K_{M,E_0} = K_{M,E_a} = 50\,\mu M$, while $k_{out}$ is set to 0.1/s. The $E_{\text{tot}}$ and $k_{cat}$ for the co-substrate cycling enzyme are 1.2 times those for the primary enzyme. In panel (**D**) the parameters are the same except for the $E_{\text{tot}}$ and $k_{cat}$ of the co-substrate cycling enzyme, which are set to 0.7 times those for the primary enzyme.

This would be the case even when we consider biosynthesis or environmental uptake of co-substrates, as the total concentration of a cycled metabolite across its different forms at steady state would then be given by a constant defined by the ratio of the influx and outflux rates (see Appendices 2 and 3). In other words, the cycled metabolite would become a 'conserved moiety' for the rest of the metabolic system and can have a constant 'pool size'. Supporting this, temporal measurement of specific co-substrate pool sizes shows that ATP and GTP pools are constant under stable metabolic conditions, but can rapidly change in response to external perturbations, possibly through inter-conversions among pools rather than through biosynthesis (*Walther et al., 2010*).

## Co-substrate cycling introduces a limitation on reaction flux

To explore the effect of co-substrate cycling on pathway fluxes, we first consider a didactic case of a single reaction. This reaction converts an arbitrary metabolite $M_0$ to $M_1$ and involves co-substrate

cycling (*Figure 1A*). For co-substrate cycling, we consider additional 'background' enzymatic reactions that are independent of $M_0$ and can also convert the co-substrate (denoted $E_a$ on *Figure 1A*). We use either irreversible or reversible enzyme dynamics to build an ordinary differential equation (ODE) kinetic model for this reaction system and solve for its steady states analytically (see Methods and Appendix 3). In the case of using irreversible enzyme kinetics, we obtain that the steady state concentration of the two metabolites, $M_0$ and $M_1$ (denoted as $m_0$ and $m_1$) are given by:

$$m_0 = \frac{\alpha \, k_{\text{in}} K_{M,E_0}}{(V_{\text{max},E_0} - k_{\text{in}})(V_{\text{max},E_a} A_{\text{tot}} - k_{\text{in}}(K_{M,E_a} + A_{\text{tot}}))}$$

$$m_1 = \frac{k_{\text{in}}}{k_{\text{out}}}$$

(1)

where $k_{in}$ and $k_{out}$ denote the rate of in-flux of $M_0$, and out-flux of $M_1$, either in-and-out of the cell or from other pathways, and $A_{\text{tot}}$ denotes the total pool size of the cycled metabolite (with the different forms of the cycled metabolite indicated as $A_0$ and $A_1$ in *Figure 1A*). The parameters $V_{\text{max},E_0}$ and $V_{\text{max},E_a}$ are the maximal rates (i.e. $V_{\text{max}} = k_{\text{cat}} E_{\text{tot}}$) for the enzymes catalysing the conversion of $A_0$ and $M_0$ into $A_1$ and $M_1$ (enzyme $E_0$), and the turnover of $A_1$ into $A_0$ (enzyme $E_a$), respectively, while the parameters $K_{M,E_0}$ and $K_{M,E_a}$ are the individual or combined Michaelis-Menten coefficients for these enzymes' substrates (i.e. for $A_0$ and $M_0$ and $A_1$, respectively). The term $\alpha$ is (in this case where all reactions are irreversible) equal to $V_{\text{max},E_a} - k_{\text{in}}$, and in general is a positive expression comprising $k_{in}$, and the Michaelis-Menten coefficients and the $V_{\text{max}}$ parameters of the background enzymes in the model (see Appendix 3, *Equations 7; 9; 11*). The steady states for the model with all enzymatic conversions being reversible, and for a model with degradation and synthesis of $A_0$ and $A_1$, are given in Appendix 3. The steady state solutions of these alternative models are structurally akin to (*1*), and do not alter the qualitative conclusions we make in what follows.

A key property of (*1*) is that it contains terms in the denominator that involve a subtraction. The presence of these terms introduces a limit on the parameter values for the system to attain a positive steady state. Specifically, we obtain the following conditions for positive steady states to exist:

$$k_{\text{in}} < V_{\text{max},E_0} \quad \text{and} \quad k_{\text{in}} < \frac{A_{\text{tot}} V_{\text{max},E_a}}{K_{M,E_a} + A_{\text{tot}}}.$$

(2)

Additionally, the 'shape' of (*1*) indicates a 'threshold effect' on the steady state value of $m_0$, where it would rise towards infinity as $k_{in}$ increases towards the lower one among the limits given in (*2*) (see *Figure 1B*).

Why does (1) show this specific form, leading to these limits? We find that this is a direct consequence of the steady state condition, where metabolite production and consumption rates need to be the same at steady state. In the case of co-substrate cycling, the production rate of $M_0$ is given by $k_{in}$, while its consumption rate is a function of the $V_{\text{max},E_0}$ and the concentration of $A_0$. In turn, the concentration of $A_0$ is determined by its re-generation rate (which is a function of $K_{M,E_a}$ and $V_{\text{max},E_a}$) and the pool size ($A_{\text{tot}}$). This explains the inequalities given in (*2*) and shows that a cycled co-substrate creates the same type of limitation (mathematically speaking) on the flux of a reaction it is involved in, as that imposed by the enzyme catalysing that reaction ($E_0$ in this example) (see *Figure 1C & D*). We also show that considering the system shown in *Figure 1A* as an enzymatic reaction without co-substrate cycling leads to only the constraint $k_{\text{in}} < V_{\text{max},E_0}$, while considering it as a non-enzymatic reaction with co-substrate cycling only, leads to only the constraint $k_{\text{in}} < A_{\text{tot}} V_{\text{max},E_a}/(K_{M,E_a} + A_{\text{tot}})$ becoming the sole limitation on the system (see Appendix 3). In other words, the two limitations act independently.

To conclude this section, we re-iterate its main result. The flux of a reaction involving co-substrate cycling is limited either by the kinetics of the primary enzyme mediating that reaction, or by the turnover rate of the co-substrate. The latter is determined by the co-substrate pool size and the kinetics of the enzyme(s) mediating its turnover.

## Co-substrate cycling causes a flux limit on linear metabolic pathways

We next considered a generalised, linear pathway model with $n + 1$ metabolites and arbitrary locations of reactions for co-substrate cycling, for example as seen in upper glycolysis (*Appendix 1—figure 1A*). In this model, we only consider intra-pathway metabolite cycling, i.e. the co-substrate is consumed

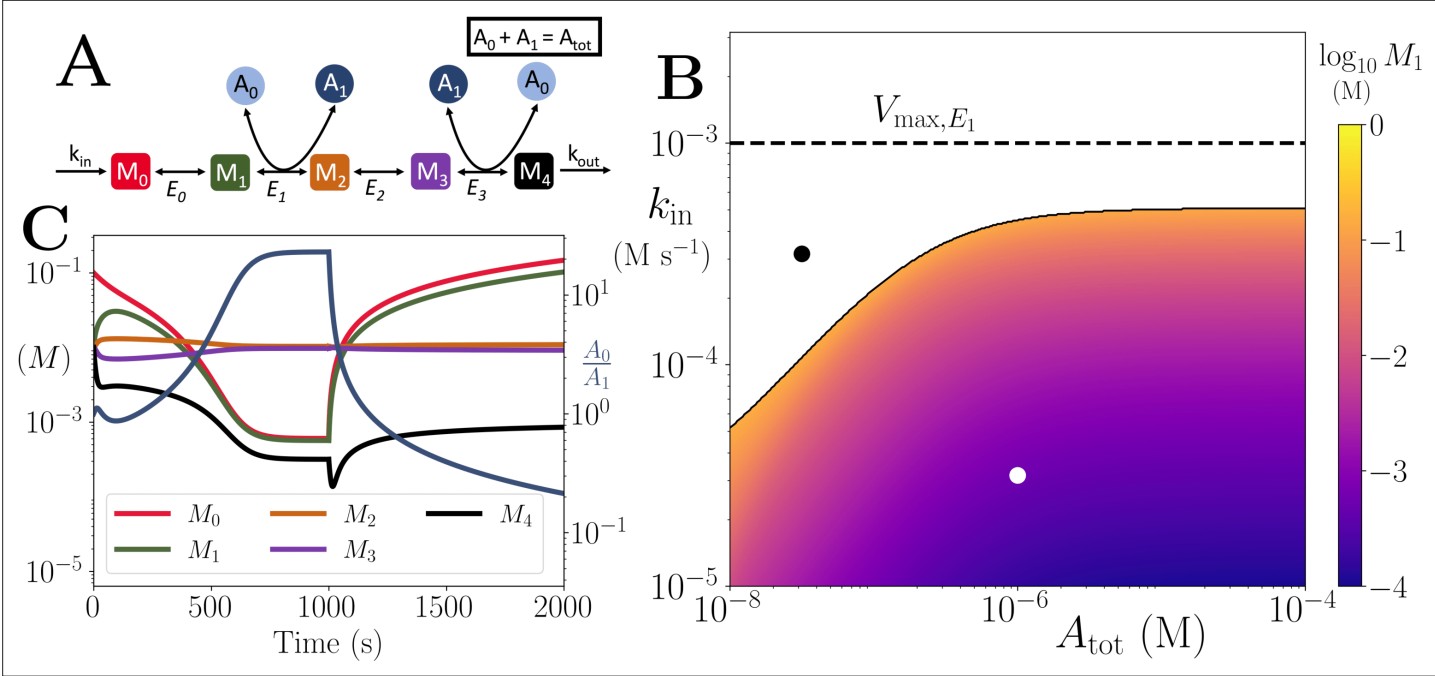

**Figure 2.** Motif, time-series and thresholds for the linear pathway model with $n = 4$. (**A**) Cartoon representation of a chain of reversible reactions with co-substrate cycling occurring solely inter-pathway. The co-substrate is considered to have two forms $A_0$ and $A_1$. (**B**) Heatmap of the steady state concentration of $M_0$ as a function of the total metabolite pool size ($A_{tot}$) and inflow rate constant ($k_{in}$). White area shows the region where there is no steady state. The dashed and solid lines indicate the limitations arising from primary enzyme ($E_1$ in this case) and co-substrate cycling, respectively, as in **Figure 1**. (**C**) Concentrations of $M_{0-4}$, and $A_0/A_1$ ratio as a function of time (with colours as indicated in the inset). At $t = 1000$ s, the parameters are switched from the white dot in panel (**B**) (where a steady state exists) to the black dot (where we see build-up of all substrates that are produced before the first co-substrate cycling reaction, and continued decline of $A_0$). The parameters used are picked from within a physiological range (see **Supplementary file 1**) and are set to: $E_{tot} = 0.01$ mM, $k_{cat} = 100/s$, $K_M = 50\,\mu m$, for all reactions, and $k_{out} = 0.1/s$.

and re-generated solely by the reactions of the pathway. Here, we show results for this model with 5 metabolites as an illustration (**Figure 2A**), while the general case is presented in Appendix 4.

We find the same kind of threshold dynamics as in the single reaction case. When $k_{in}$ is above a threshold value, the metabolite $M_0$ accumulates towards infinity and the system does not have a steady state (**Figure 2B**). A numerical analysis, as well as our analytical solution, reveals that the accumulation of metabolites applies to all metabolites upstream of the first reaction with co-substrate cycling (**Figure 2C** and Appendix 4). Additionally, metabolites downstream of the cycling reaction accumulate to a steady state level that does not depend on $k_{in}$ (**Figure 2C** and **Appendix 3—figure 1**). In other words, pathway output cannot be increased further by increasing $k_{in}$ beyond the threshold. Finally, as $k_{in}$ increases, the cycled metabolite pool shifts towards one form and the ratio of the two forms approaches zero (**Figure 2C**).

An analytical expression for the threshold for $k_{in}$, like shown in (**2**), could not be derived for linear pathways with $n > 3$, but our analytical study indicates that (i) the threshold is always linked to $A_{tot}$ and enzyme kinetic parameters, and (ii) the concentration of all metabolites upstream (downstream) to the reaction coupled to metabolite cycling will accumulate towards infinity (a fixed value) as $k_{in}$ approaches the threshold (see Appendix 4). In **Figure 2**, we illustrate these dynamics with simulations for a system with $n = 4$.

We also considered several variants of this generalised linear pathway model, corresponding to biologically relevant cases as shown in **Appendix 1—figure 1**. These included (i) intra-pathway cycling of two different metabolites, as seen with ATP and NADH in combined upper glycolysis and fermentation pathways (Appendix 5), (ii) different stoichiometries for consumption and re-generation reactions of the cycled metabolite, as seen in upper glycolysis (Appendix 6), and (iii) cycling of one metabolite interlinked with that of another, as seen in nitrogen assimilation (Appendix 7). The results in the Appendices confirm that all these cases display similar threshold dynamics, where the threshold point is a function of the co-substrate pool size and the enzyme kinetics.

## Cycled metabolite related limit could be relevant for specific reactions from central metabolism

Based on flux values that are either experimentally measured or predicted by flux balance analysis (FBA), many reactions from the central carbon metabolism of the model organism *Escherichia coli* are shown to have lower flux than expected from the kinetics of their immediate enzymes (i.e. $V_{max}$) (*Davidi et al., 2016*). This finding is based on calculating $V_{max}$ from *in vitro* measured $k_{cat}$ values of specific enzymes and their *in vivo* levels based on proteomics studies in *E. coli* (see Materials and methods). The flux and enzyme concentration data were from other studies which measured them during the exponential phase in *E. coli* growing on minimal media supplemented with various carbon sources (*Schmidt et al., 2016*; *Gerosa et al., 2015*). If we consider measured fluxes for each reaction as a proxy for $k_{in}$ (notice that these two would be equal at steady state), we can conclude from the fact that there were no observed substrate accumulation in these reactions, as an indication for the analysed reactions carrying fluxes below the first limit identified above in (*2*). There could be several explanations for this observation of measured fluxes being lower than the limit set by measured enzyme kinetics and level. One simple explanation could be that there is a discrepancy between *in vitro* measured enzyme kinetics and *in vivo* realised ones. Alternatively, this discrepancy can be low, but the lower flux could be arising because there are additional limiting factors other than the enzymes mediating the main reaction. Among such additional limiting factors, substrate limitation and thermodynamic effects are shown to partially explain observed lower fluxes in some reactions (*Davidi et al., 2016*; see also below results). Here, we highlight that the presented theory shows that an additional possible limitation could be the co-substrate pool size and turnover dynamics.

To explore this possibility, we re-analysed the flux values compiled previously (*Davidi et al., 2016*; *Gerosa et al., 2015*) and focused solely on reactions that are linked to ATP, NADH, or NADPH pools (see Materials and methods and *Supplementary file 1*). The resulting dataset contained fluxes, substrate concentrations, and enzyme levels for 45 different reactions determined under 7 different conditions along with turnover numbers and kinetic constants of the corresponding enzymes. In total, we gathered 49 combinations of enzyme-flux-$k_{cat}$ values with full experimental data and 259 combinations with only FBA-predicted flux values. We compared the flux values that would be expected from the primary enzyme limit identified above, under all conditions analysed (*Figure 3A*), and in addition checked whether the saturation effect of the primary substrate could explain the difference (*Appendix 8—figure 1*). We found that in both cases, about 80% of these reactions carry flux lower than what is expected from enzyme kinetics (*Appendix 8—figure 2*), suggesting that the limits imposed by co-factor dynamics might be constraining the flux further. The low number of the cases where the flux exceeds the limit might be due to uncertainties in measurement of flux, enzyme or substrate level.

It is also possible that observed lower fluxes are due to thermodynamic limitations. This is very difficult to analyse without more data, as calculating reaction thermodynamics requires knowledge of concentrations for *all* substrates and products, as well as enzyme Michaelis-Menten constants in both forward and backward directions. This information is currently not available except for few of the reactions among the ones we analysed. Nevertheless, to give as much insight as possible on the thermodynamic effect, we analysed the physiological Gibbs free energy (the $\Delta_r G'$ is calculated assuming that all reactants are at 1 mM and pH = 7) against the normalized flux – $v/(E_0 \cdot k_{cat})$ (*Appendix 8—figure 3*). This shows that although in few cases, such as malate dehydrogenase (MDH), the normalised flux seems to be greatly reduced by the thermodynamic barrier, the general picture is that there is little correlation between reaction flux and thermodynamics.

We have also checked the relation between fluxes and co-substrate pool sizes. Co-substrate pool sizes do change between different conditions, and we note that such changes cannot be due to flux changes in co-substrate utilising reactions. But, on the other hand, changes in pool size can affect flux in those reactions, where co-substrate dynamics is limiting (as predicted by the theory). For both measured and FBA-predicted fluxes, we find that several reactions show significant correlation between flux and co-substrate pool size (see *Figure 3B–D*, see also *Appendix 8—table 1* and *Appendix 8—figure 4*). In the case of FBA-predicted fluxes, however, we note that these results can be confounded due to additional, flux-to-flux correlations and correlations between pool sizes and growth rate. Among reactions with measured fluxes, the three reactions with high correlation to pool size are those mediated by malate dehydrogenase (*MDH*), linked with NADH pool, phosphoglycerate

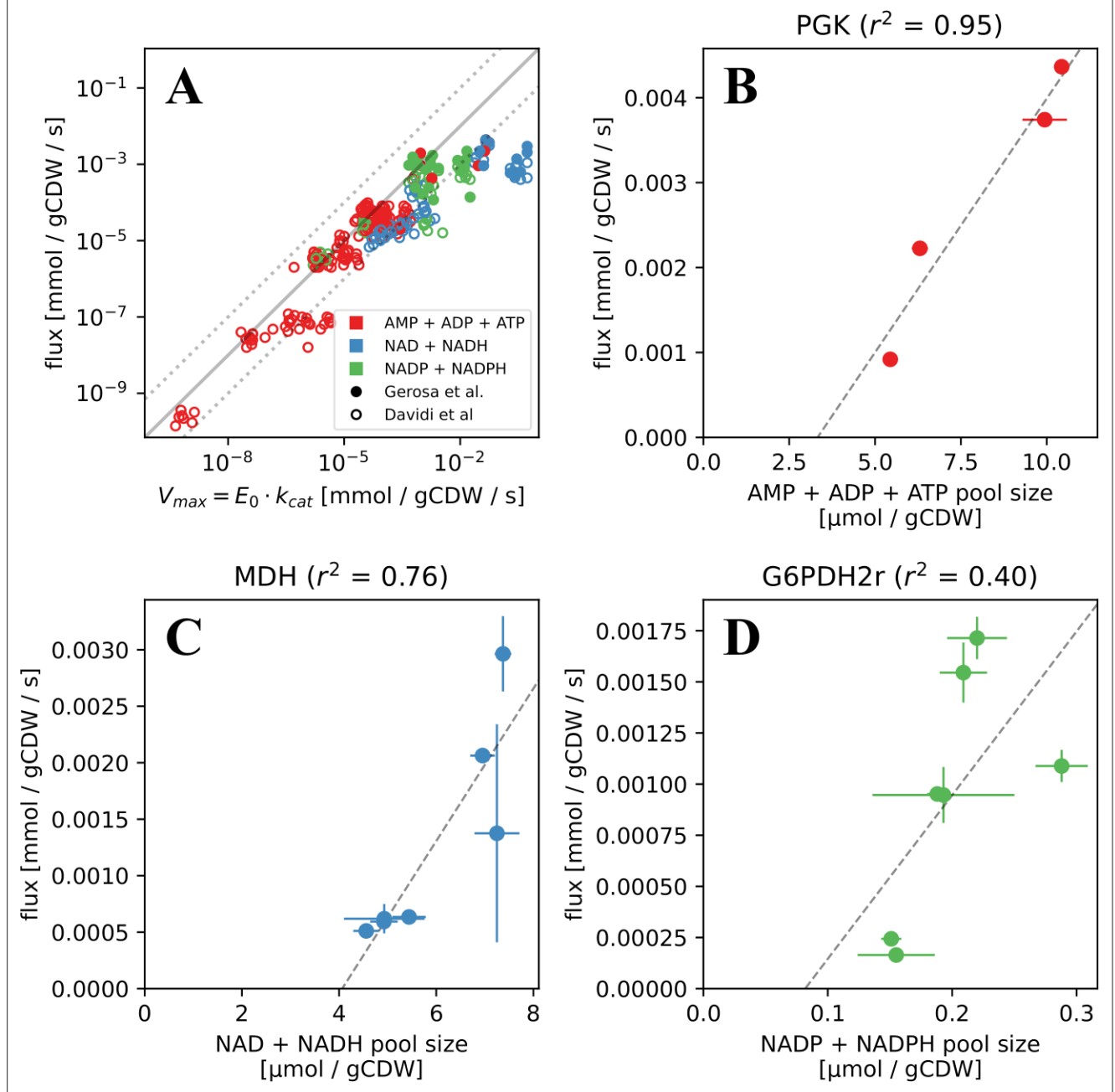

**Figure 3.** Measured and FBA-predicted flux values are typically lower than the calculated primary enzyme threshold. (**A**) Measured and FBA-predicted flux values (from *Davidi et al., 2016*; *Gerosa et al., 2015*) plotted against the calculated primary enzyme kinetic threshold (first part of eq. (1)). Notice that there are 7 points for each reaction, corresponding to the different experimental conditions under which measurements or FBA modelling was done (see *Supplementary file 1* for data, along with reaction names and metabolites involved). (**B–D**) Measured flux values under different experimental conditions (from *Gerosa et al., 2015*) for select reactions plotted against the corresponding co-substrate pool size. Panels B to D show reactions for phosphoglycerate kinase (*PGK*), malate dehydrogenase (*MDH*), and glucose-6-phosphate dehydrogenase (*G6PDH*). Each point on these panels is a separate flux measurement under a different environmental condition, where the co-substrate pool size is also measured. Error bars represent standard deviations of flux and metabolite measurements as they appear in the dataset from Gerosa et al. Point colours represent co-substrate type and are as shown in the legend to panel A. Lines show the best linear fit with the corresponding normalised RMSE shown in the panel title.

kinase (*PGK*), linked with the ATP pool, and glucose-6-phosphate dehydrogenase (*G6PDH*) linked with the NADPH pool.

In summary, these results show that for reactions involving co-substrate cycling (1) measured fluxes are lower than those predicted by kinetics of the primary enzyme (i.e. enzyme involved in substrate

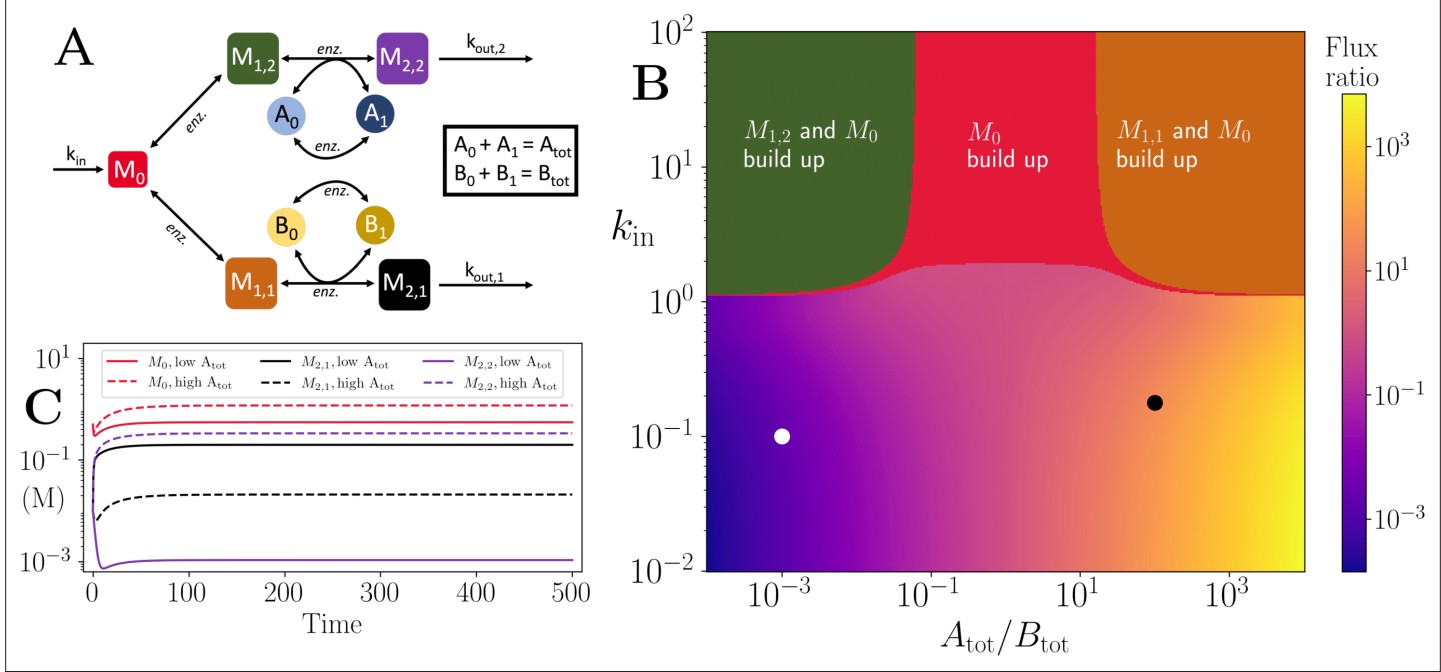

**Figure 4.** Motif, heatmap and time-series for the branching pathway model. (**A**) Cartoon representation of two branching pathways from the same upstream metabolite. The two branches are linked to separate co-substrate pools, $A$ and $B$. Note that pathway independent turnover of the co-substrates is included in the model (see **Figure 4—source code 1**). (**B**) The pathways' flux ratio (i.e. flux into $M_{2,2}$ divided by flux into $M_{2,1}$) shown in colour mapping, against the ratio of co-substrate pool sizes, $A_{tot}$ and $B_{tot}$, and the influx rate, $k_{in}$, into the upstream metabolite. In the block colour areas, the system has no steady state and the indicated metabolite(s) $M_0$ and one of the metabolites $M_{1,2}$ or $M_{1,1}$ accumulate towards infinity. (**C**) Concentrations of upstream and branch-endpoint metabolites over time, coloured as shown in the inset of the panel. The solid lines show results using parameters indicated by the white dot in panel (**B**), where $B_{tot} > A_{tot}$, while the dashed lines show results using parameters indicated by the black dot in panel (**B**), where $A_{tot} > B_{tot}$. For both simulations, all kinetic parameters are arbitrarily set to 1, apart from the pathway-independent co-substrate recycling ($V_{max,E_a}$ and $V_{max,E_b}$) that is set to 10 (see **Figure 4—source code 1**).

The online version of this article includes the following source code for figure 4:

**Source code 1.** Python implementation of branched pathway model, presented in **Figure 4**.

conversion) alone, and (2) there is – for some reactions – a correlation between flux and co-substrate pool size. Both observations *could* indicate co-substrate pool sizes and/or co-substrate cycling dynamics being a limiting factor for flux. We can not state this as a certainty, however, as there are possibly other factors acting as the extra limitation, including thermodynamic effects. These points call for further experimental analysis of co-substrate cycling within the study of metabolic system dynamics.

## Co-substrate cycling allows regulation of branch point fluxes

In addition to its possible constraining effects on fluxes, we wondered if co-substrate dynamics can offer a regulatory element in cellular metabolism. In particular, co-substrate cycling can commonly interconnect two independent pathways, or pathways branching from the same upstream metabolite, where it could influence flux distributions among those pathways. To explore this idea, we considered a model of a branching pathway, with each branch involving a different co-substrate, $A$ and $B$ (**Figure 4A** and Appendix 1). This scenario is seen in the synthesis of certain amino acids that start from a common precursor but utilise NADH or NADPH, for example Serine and Threonine.

We hypothesised that regulating the two co-substrate pool sizes, $A_{tot}$ and $B_{tot}$, could allow regulation of the fluxes on the two branches. To test this hypothesis, we ran numerical simulations with different co-substrate pool sizes and influx rates into the branch point. We found that the ratio of fluxes across the two branches can be regulated by changing the ratio of $A_{tot}$ to $B_{tot}$ (**Figure 4B**). The regulation effect is seen with a large range of $k_{in}$ values, but the threshold effect is still present with high enough $k_{in}$ values leading to loss of steady state and metabolite build up. In that case, the resulting metabolite build-up can affect either branch depending on which co-substrate has the lower

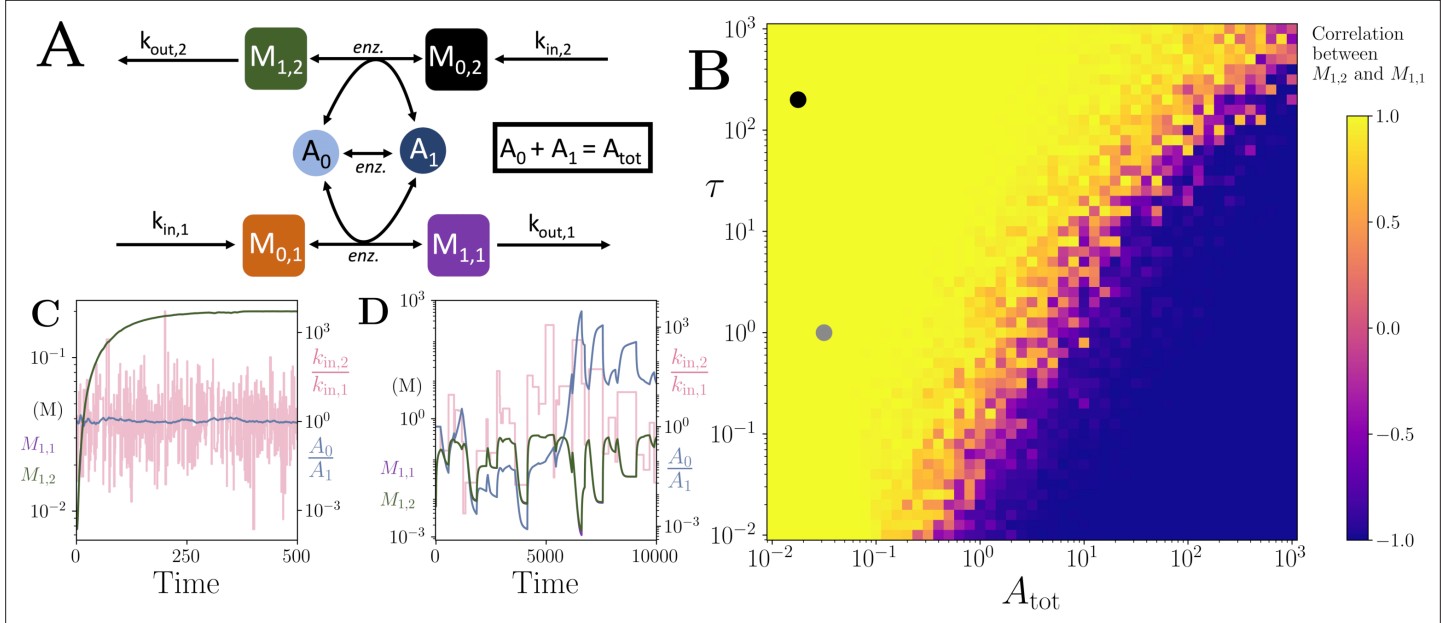

**Figure 5.** Motif, heatmap and time-series for the coupled pathway model with noisy influxes. (**A**) Cartoon representation of two pathways coupled via the same co-substrate cycling. The two forms of the co-substrate are indicated as $A_0$ and $A_1$. It is converted from $A_0$ to $A_1$ on the lower pathway, and from $A_1$ to $A_0$ in the upper pathway. The presented results are for a model with reversible enzyme kinetics, while the results from a model with irreversible enzyme kinetics are shown in **Appendix 10—figure 2**. (**B**) Correlation coefficient of the two pathway product metabolites, $M_{1,2}$ and $M_{1,1}$, as a function of the total amount of co-substrate ($A_{tot}$) and the extent of fluctuations in the two pathway influxes, $k_{in,1}$ and $k_{in,2}$. The influx fluctuation is characterised by a waiting time that is exponentially distributed with mean $\tau$, after which the log ratio of the $k_{in}$ values is drawn from a standard normal distribution. The mean of the $k_{in}$ values is set to be 0.1 and the pathway-independent cycling occurs at a much lower rate compared to the other reactions (see **Figure 5—source code 1**). (**C**) Concentrations of metabolites $M_{1,2}$ (green) and $M_{1,1}$ (magenta), pathway influx ratio (pink), and $A_0/A_1$ ratio (blue) as a function of time. The simulation is run with parameters corresponding to the grey dot in (**B**) where the products are correlated, and the rate of $k_{in}$ fluctuations is on a similar timescale to the other reactions. The system is largely unresponsive to the noise, and the $M_{1,2}$ and $M_{1,1}$ are fully correlated (i.e. the green and magenta curves overlap). (**D**) Concentrations of metabolites $M_{1,2}$ (green) and $M_{1,1}$ (magenta), pathway influx ratio (pink), and $A_0/A_1$ ratio (blue) as a function of time. The simulation is run with parameters corresponding to the black dot in (**B**) where the products are correlated, but the fluctuations in $k_{in}$ values occur at a much lower rate than the other reactions. The system is responsive to the noise, yet the $M_{1,2}$ and $M_{1,1}$ are fully correlated (i.e. the green and magenta curves overlap). For both simulations, all kinetic parameters are arbitrarily set to 1, apart from the pathway-independent co-substrate recycling ($V_{max,E_a}$) that is set to 0.01 (see **Figure 5—source code 1**).

The online version of this article includes the following source code for figure 5:

**Source code 1.** Python implementation of connected pathway model, presented in **Figure 5**.

pool size (see upper corner regions on **Figure 4B**). There is also a regime of only the upstream, branch point metabolite building-up, but this happens only when all reactions are considered as reversible and the extent of it depends on turnover rates of the two co-substrates (**Appendix 9—figure 1**).

In the no-build-up, steady state regime, changing the pool size ratio of the two co-substrates causes a change in fluxes and metabolite levels, The change in flux ratio is of the same order as the change in pool size ratio (**Figure 4C & D**), while the change in the ratio of metabolite levels is in general less. This relation between pool size ratio and flux ratio on each branch is unaffected by the value of $k_{in}$. We also evaluated the level of regulation that can be achieved by varying the turnover rates of $A$ and $B$. The flux regulation effect in this case is weaker, unless the difference in the turnover rates is large and the influx rate is close to the threshold (**Appendix 9—figure 2**).

## Inter-pathway co-substrate cycling limits maximum influx difference and allows for correlating pathway outfluxes despite influx noise

We next considered a simplified model of two independent pathways interconnected by a single co-substrate pool (**Figure 5A** and Appendix 10). This model can represent several different processes in metabolism, for example the coupling between the TCA cycle and the respiratory electron transfer chain, through NADH generation and consumption respectively, or the coupling between the pentose

phosphate pathway and some amino acid biosynthesis pathways (notably Methionine), through NADPH generation and consumption respectively. We hypothesised that such inter-pathway co-substrate cycling might cause; (1) the co-substrate related limit to relate to difference in pathway influxes, rather than input into one pathway, and (2) a coupling of the pathway output fluxes against influx fluctuations, such that the output fluxes remain correlated to each other, despite differences in influx levels.

To address the first hypothesis, we used analytical methods and explored the relation between the systems' ability to reach steady state and system parameters. We found that, indeed, for this coupled system, the ability to reach steady state depends on the influx difference between two pathways, as well as the total pool size and the kinetic parameters relating to pathway-independent turn-over of the co-substrate (see Appendix 10). In other words, for two pathways coupled via co-substrate cycling, the cycling-dependent flux limit for each is not determined by their own influx but rather on how different this is to the coupled pathways' influx (*Appendix 10—figure 1*).

To test the second hypothesis about the output coupling, we considered the correlation of the steady-state outputs of the pathways with random fluctuations in their influx (*Figure 5B*). We found that there is either a high level of anti-correlation or correlation between pathway outputs for all pool sizes tested (blue and yellow regions in *Figure 5B*). As the pool size decreases, the system reaches a point where there is a transition from anti-correlation to high correlation in product outputs (blue to yellow region in *Figure 5B*). At low pool sizes, pathway outputs are fully correlated despite significant fluctuation in pathway influx (*Figure 5C & D*). Within this correlated regime, we identified two different sub-regimes. The first is a regime where the metabolite concentrations stay relatively constant despite the influx noise (*Figure 5C*). This regime arises because the influx fluctuations are occurring at a much faster rate than the pathway kinetics and the system is rather non-responsive to influx noise. In a second regime, the influx noise is at time scales comparable to pathway kinetics. Here, the metabolite concentrations can readily change with the influx changes, and the system is 'responsive', yet the output levels are still correlated (*Figure 5D*). This regime is directly a result of co-substrate cycling dynamics. Because the turnover of co-substrate is essentially coupling the two pathways, their outputs become directly correlated. This effect does not depend on whether pathway reactions are modelled as reversible or irreversible, and the results here for $V_{\mathrm{max},E_a} = 0.01$ are representative of those for $V_{\mathrm{max},E_a} = 0$ (see *Appendix 10—figure 2*). As we increase the rate of the assumed background reaction, that is pathway-independent turnover of the co-substrate, we find that these results remain qualitatively the same, but the transition point from anti- correlation in outputs to correlation, shifts to lower $A_{\mathrm{tot}}$ values (*Appendix 10—figure 2*).

These results show that coupling by co-substrate cycling can introduce a limit on influxes of independent pathways or metabolic processes. Furthermore, such coupling can allow high correlation in the pathway outputs, despite significant noise in the inputs of those pathways. These effects are most readily seen where the turnover of the coupling co-substrate by other processes is low. We note that an example case for such a scenario is the coupling of respiration and oxidative phosphorylation, where transmembrane proton cycling takes the role of the cycled co-substrate (*Stucki, 1980*).

## Discussion

We presented a mathematical analysis of metabolic systems featuring co-substrate cycling and showed that such cycling introduces a threshold effect on system dynamics. As the pathway's influx rate, $k_{in}$, approaches a threshold value, the steady state concentrations of metabolites that are upstream of a reaction linked to co-substrate cycling, increase towards infinity and the system cannot reach steady state. Specifically, for reactions involving co-substrates, there are two thresholds on influx rate, one relating to the kinetics of the enzyme directly mediating that reaction, and another relating to the kinetics of the enzymes mediating the turnover of the co-substrate and the pool size of that co-substrate.

This second, additional constraint arising from co-substrate cycling can be directly relevant for cell physiology. We particularly note that this threshold can be highly dynamic, and condition- and cell-dependent. When cells have a permanently or occasionally lowered total co-substrate pool size (i.e. lower $A_{\mathrm{tot}}$), or when they are placed under challenging conditions (e.g. high carbon- or nitrogen-source concentrations) causing higher $k_{in}$ values across various pathways, their metabolic systems can be close to the threshold presented here. While both $k_{in}$ and $A_{\mathrm{tot}}$ can be adjusted in the long

term, for example by reducing substrate transporter expression or increasing co-substrate biosynthesis, there can be short term impact on cells experiencing significant flux limitations and metabolite accumulations.

Comparing measured flux data against estimated flux values based on measured enzyme levels from proteomics and enzyme kinetics from *in vitro* studies, we have provided support that fluxes in co-substrate linked reactions could indeed by limited by co-substrate pool dynamics under physiological conditions. This analysis was based on the model organism *E. coli* and is limited even for this organism due to limited flux and proteomics data. For example, the data compiled here contained 14 co-substrate reactions with experimentally measured fluxes, but only half of these could be used due to lack of measurement on enzyme concentrations. We hope that the presented theory will provide motivation to further expand the available data sets, especially for reactions relating to co-substrate linked reactions. In this quest, we expect that the expansion of measurements to eukaryotic cells to be particularly challenging due to organelle-specific pools, but some progress is being made to achieve at least mitochondrial and cytosolic measurements (*Chen et al., 2016*). Despite the current limitations, our data-based analyses highlighted three key reactions, that are possibly limited by co-substrate dynamics, and that are mediated by phosphoglycerate kinase (*PGK*), malate dehydrogenase (*MDH*), and glucose-6-phosphate dehydrogenase (*G6PDH*) and linked to ATP, NADH, and NADPH pools. Possible flux limitation of these reactions by co-substrate dynamics can also be subjected to further experimental study -– as we discuss further below.

Overall, the presented theoretical results could contribute to our understanding of two commonly observed metabolic dynamics that arise under increasing or high substrate concentrations, and that are shown to cause either 'substrate-induced death' (*van Heerden et al., 2014*) or 'overflow metabolism'. The latter usually refers to a respiration-to-fermentation switch under respiratory conditions (e.g. the Warburg and Crabtree effects [*Warburg, 1956*; *Diaz-Ruiz et al., 2009*; *Basan et al., 2015*; *Meyer et al., 1984*]), but other types of overflow metabolism, involving excretion of amino acids and vitamins, has also been observed (*Ponomarova et al., 2017*; *Jiang et al., 2018*). Several arguments have been put forward to explain these observations, including osmotic effects arising from high substrate concentrations causing cell death and limitations in respiratory pathways or cell's protein resources causing a respiration-to-fermentation switch (*Diaz-Ruiz et al., 2009*; *Majewski and Domach, 1990*; *Basan et al., 2015*). Notwithstanding the possible roles of these processes, the presented theory leads to the hypothesis that both substrate-induced death and metabolite excretions could relate to increasing substrate influx rate reaching close to the limits imposed by co-substrate dynamics. There is experimental support for this hypothesis in the case of both observations. Substrate-induced death and associated mutant phenotypes are linked to the dynamics associated with ATP regeneration in glycolysis (*Teusink et al., 1998*; *Koebmann et al., 2002*; *van Heerden et al., 2014*). Based on that finding, it has been argued that cells aim to avoid the threshold dynamics through allosteric regulation of those steps of the glycolysis that involve ATP consumption (*Teusink et al., 1998*). In the case of respiration-to-fermentation switch, it has been shown that the glucose influx threshold, at which fermentative overflow starts, changes upon introducing additional NADH conversion reactions in both yeast and *E. coli* populations (*Vemuri et al., 2006*; *Vemuri et al., 2007*). In another supportive case, sulfur-compound excretions are linked to alterations in the NADPH pool through changes in the amino acid metabolism (*Olin-Sandoval et al., 2019*; *Green et al., 2020*).

Dynamical thresholds relating to co-substrate pools would be relevant for all co-substrates, and not just for ATP or NADH, which have been the focus of most experimental studies to date. We would expect that altering kinetics of enzymes involved in co-substrate cycling can have direct impact on cell physiology, and in particular on metabolic excretions. This prediction can be tested by exploring the effect of mutations on enzymes linked to co-substrate consumption and production, or by altering co-substrate pool sizes and assessing effects of such perturbations on the dynamics of metabolic excretions. These tests can be experimentally implemented by introducing additional enzymes specialising in co-substrate consumption or production (e.g. ATPases, oxidases, or other) and controlling their expression. It would also be possible to monitor co-substrate pool sizes in cells in real time by using fluorescent sensors on key metabolites such as ATP or glutamate, or by measuring autofluorescence of certain pool metabolites, such as NAD(P)H, under alterations to influx rate of glucose or ammonium.

Besides acting as a flux constraint, we find that co-substrate pools can also allow for regulation of pathway fluxes through regulation of pool size or turnover dynamics. We find that such regulation can take the form of balancing inter-connected pathways, thereby ensuring correlation between outputs of different metabolic processes, or regulating flux across branch points. Regulation of fluxes through co-substrate pools can act to adjust metabolic fluxes at time scales shorter than possible via gene regulation, and possibly at similar time scales as with allosteric regulation – especially when considering pool size alterations through exchange among connected pools. Possibility of such a regulatory role has been indicated experimentally, where total ATP pool size is found to change when yeast cells are confronted with a sudden increase in glucose influx rate (*Walther et al., 2010*). In that study, the change in the ATP pool is found to link to the purine metabolism pathways, which are linked to several conserved moieties; GTP, ATP, NAD, NADP, S-adenosylmethionine, and Coenzyme A. These findings suggest that cells could dynamically alter pool sizes associated with different parts of metabolism, limiting flux through some pathways, while allowing higher flux in others, and thereby shifting the metabolites from the latter to the former. This could provide a dynamic self-regulation and the pool sizes of key co-substrates could be seen as 'tuning points' controlling a more complex metabolic system. We thus propose further experimental analyses focusing on co-substrate pool sizes and turnover dynamics to understand and manipulate cell physiology.

## Materials and methods
### Model of a single reaction with co-substrate cycling
The metabolic system shown in *Figure 1A* comprises the following biochemical reactions:

$$\xrightarrow{k_{\text{in}}} M_0$$
$$M_1 \xrightarrow{k_{\text{out}}}$$
$$M_0 + A_0 \longleftrightarrow M_1 + A_1$$
$$A_0 \longleftrightarrow A_1$$
$$(3)$$

where metabolites are denoted by $M_i$ and the different forms of the co-substrate are denoted by $A_i$. We assume additional conversion between $A_1$ and $A_0$, mediated through other enzymatic reactions. The parameters $k_{in}$, and $k_{out}$ denote the in- and out- flux of $M_0$ and $M_1$ respectively, from and to other pathways or across cell boundary. The ordinary differential equations (ODEs) for the system shown in (*3*) (and *Figure 1A*), using irreversible Michaelis-Menten enzyme kinetics would be:

$$\frac{dm_0}{dt} = k_{\text{in}} - \frac{V_{\text{max},E_0}\, a_0 m_0}{K_{M,E_0} + a_0 m_0}$$

$$\frac{dm_1}{dt} = \frac{V_{\text{max},E_0}\, a_0 m_0}{K_{M,E_0} + a_0 m_0} - k_{\text{out}} m_1$$

$$\frac{da_0}{dt} = \frac{V_{\text{max},E_a}\, a_1}{K_{M,E_a} + a_1} - \frac{V_{\text{max},E_0}\, a_0 m_0}{K_{M,E_0} + a_0 m_0}$$
$$(4)$$

$$\frac{da_1}{dt} = \frac{V_{\text{max},E_0}\, a_0 m_0}{K_{M,E_0} + a_0 m_0} - \frac{V_{\text{max},E_a}\, a_1}{K_{M,E_a} + a_1}$$

where $m_i$ and $a_i$ denote the concentrations of $M_i$ and $A_i$ respectively, $K_M$ denotes a composite parameter of the Michaelis-Menten coefficients of the enzyme for its substrates, and $V_{\text{max}}$ is the total enzyme concentration times its catalytic rate (i.e. $V_{\text{max}} = k_{\text{cat}} E_{\text{tot}}$) (see *Appendix 11—table 1* for a list of parameters and their units). We further have the conservation relation $a_0 + a_1 = A_{\text{tot}}$, where $A_{\text{tot}}$ is a constant. This assumption would be justified when influx of any form of the cycled metabolite into the system is independent of the rest of the metabolic system (see further discussion and analysis in Appendix 2). The steady states of (*4*) can be found by setting the left side equal to zero and performing algebraic re-arrangements to isolate each of the variables. The resulting analytical expressions for steady state metabolite concentration are shown in (*2*), and in Appendix 3 for this model with reversible enzyme kinetics, as well as for other models.

## Symbolic and numerical computations

For all symbolic computations, utilised in finding steady state solutions and deriving mathematical conditions on rate parameters, we used the software Maple 2021, as well as theoretical results presented in *Torres and Feliu, 2021*. To run numerical simulations of select systems, we used Python packages with the standard solver functions. All numerical simulations were performed in the Python environment. The main model simulation files relating to *Figures 4 and 5* are provided as , while all remaining simulation and analysis scripts are made available through a dedicated repository (*West et al., 2023*).

## Reaction fluxes and enzyme kinetic parameters

To support the model findings on co-substrate pools acting as a possible limitation on reaction fluxes, we analysed measured and FBA-derived flux data collated previously (*Davidi et al., 2016*; *Gerosa et al., 2015*). We focused our analyses on reactions involving co-substrates only. We compared measured (or FBA-derived) fluxes to flux thresholds based on enzyme kinetics (i.e. first condition in *Eq. 2*). To calculate the latter, we used data on enzyme kinetics and levels as collated in *Davidi et al., 2016*, which is based on the BRENDA database (*Chang et al., 2021*) and proteomics-based measurements (*Schmidt et al., 2016*). We note that most available kinetic constants for enzymes have been obtained under *in vitro* conditions, which can be very different from those of the cytosol (*García-Contreras et al., 2012*). When comparing flux levels against co-substrate pool sizes, we used the matching, measured pool-sizes from *Gerosa et al., 2015*. All the data used in this analysis is provided in the *Supplementary file 1*, and through a dedicated repository (*West et al., 2023*).

## Acknowledgements

This project is funded by the Biotechnology and Biological Sciences Research Council (BBSRC) (grant BB/T010150/1). EF acknowledges funding from the Novo Nordisk Foundation (grant NNF18OC0052483), while OSS acknowledges support from the Gordon and Betty Moore Foundation (grant https://doi.org/10.37807/GBMF9200).

We would like to thank Wenying Shou for constructive comments on an earlier version of this manuscript, and Dan Davidi for his help with datasets of reaction fluxes and enzyme abundances.

## Additional information

### Funding

| Funder | Grant reference number | Author |
|---|---|---|
| Biotechnology and Biological Sciences Research Council | BB/T010150/1 | Robert West |
| Novo Nordisk | F18OC0052483 | Elisenda Feliu |
| Gordon and Betty Moore Foundation | GBMF9200 | Orkun S Soyer |

The funders had no role in study design, data collection and interpretation, or the decision to submit the work for publication.

### Author contributions

Robert West, Conceptualization, Resources, Software, Formal analysis, Writing – review and editing; Hadrien Delattre, Conceptualization, Investigation, Writing – review and editing; Elad Noor, Resources, Formal analysis, Investigation, Methodology, Writing – review and editing; Elisenda Feliu, Formal analysis, Methodology, Writing – review and editing; Orkun S Soyer, Conceptualization, Supervision, Investigation, Writing – original draft, Project administration, Writing – review and editing

### Author ORCIDs

Robert West ⬚ http://orcid.org/0000-0001-9348-0258

Hadrien Delattre http://orcid.org/0000-0002-5488-8370
Elad Noor http://orcid.org/0000-0001-8776-4799
Elisenda Feliu http://orcid.org/0000-0001-7205-6511
Orkun S Soyer http://orcid.org/0000-0002-9504-3796

**Decision letter and Author response**
Decision letter https://doi.org/10.7554/eLife.84379.sa1
Author response https://doi.org/10.7554/eLife.84379.sa2

## Additional files

### Supplementary files
• Supplementary file 1. Enzyme kinetics, flux, metabolite concentration, and enzyme abundance data associated with flux analyses.

• MDAR checklist

### Data availability
All used data and models for cases shown in Figure 4 and 5 are made available via a dedicated repository (https://doi.org/10.5281/zenodo.7565439) and the following Github page: https://github.com/OSS-Lab/CoSubstrateDynamics/tree/v1.0.0.

The following dataset was generated:

| Author(s) | Year | Dataset title | Dataset URL | Database and Identifier |
|---|---|---|---|---|
| West R, Delattre H, Noor E, Feliu E, Soyer O | 2023 | Co-substrate dynamics analysis software and data | https://doi.org/10.5281/zenodo.7565439 | Zenodo, 10.5281/zenodo.7565439 |

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

## Appendix 1

## The mathematical approach and modelling setting

As explained in the main text, we are interested in understanding the effect of co-substrate cycling on the flux through metabolic pathways, such as those shown in *Appendix 1—figure 1*.

In these Appendices, we use a generic co-substrate pair, denoted as $A_0$, $A_1$. We consider the synthesis or degradation of the co-substrate pair, or consider it as a conserved moiety, i.e. having a fixed total concentration. Our generic co-substrate pair, $A_0$ and $A_1$, can be taken as representing a specific co-substrate, such as NAD(H), but note that the mathematical analyses presented would be applicable to any co-substrate pair in natural metabolic pathways (as discussed in the main text).

For our analyses, we consider a generalised model of a linear metabolic pathway, as well as additional metabolic pathway structures. Throughout the presented analyses, we consider reactions to be either enzyme mediated or not, and when they are enzyme mediated, we consider them either to be reversible or irreversible. In the former case, the enzymatic conversions are shown as $M_{i-1} \rightleftharpoons M_i$, while in the latter case, they are shown as $M_{i-1} \longrightarrow M_i$. These notations do not show enzyme complexes explicitly, but we use enzymatic rates derived from reaction schemes accounting for enzyme complexes (see below).

In certain models, we consider some, or all, cycling reactions of the co-substrate to occur independently of the enzymatic reactions involved in the metabolic pathway, for example due to hydrolysis reactions. We refer to this type of recycling as free conversion, for example in the case of a generic co-substrate considered here, we have:

$$A_0 \underset{k_6}{\overset{k_5}{\rightleftharpoons}} A_1.$$

We talk about irreversible co-substrate conversion, if $k_5 = 0$ or $k_6 = 0$, that is, only conversion in one direction is considered. We talk about no free co-substrate conversion, if $k_5 = k_6 = 0$, that is, the co-substrate cycling is only related through the reactions in the metabolic pathway.

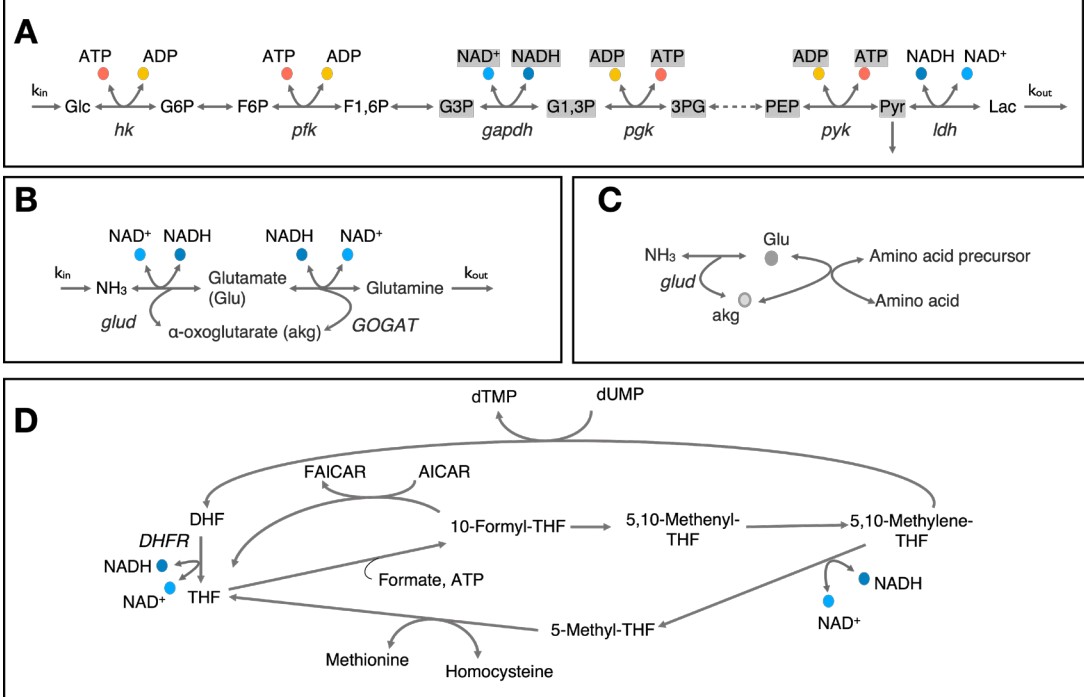

**Appendix 1—figure 1.** Cartoon representation of select metabolic pathways involving co-substrate cycling. (**A**) Cartoon representation of upper glycolysis pathway. Note that stoichiometric balance across the pathway changes after F1,6P and metabolites and co-substrates highlighted with gray background have a stoichiometry *Appendix 1—figure 1 continued on next page*

*Appendix 1—figure 1 continued*

of 2. (**B & C**) Cartoon representation of nitrogen assimilation via glutamate and involvement of glutamate cycling in amino acid biosynthesis. (**D**) Cartoon representation of metabolite cycles involved in one-carbon metabolism around tetrahydorfolate. Enzyme and metabolite name abbreviations are: Glc – Glucose, G6P – Glucose-6-phosphate, F6P – Fructose-6-phosphate, F1,6P – Fructose-1,6-biphosphate, G3P – glyceraldehyde-3-phosphate, G1,3P – 1,3-biphospho-D-glycerate, 3 PG – 3-phospho-D-glycerate, PEP – Phosphoenolpyruvate, Pyr – Pyruvate, Lac – Lactate, DHF – Dihydrofolate, THF - Tetrahydrofolate, AICAR - 5-amino-4-imidazolecarboxamide ribotide, FAICAR - 5'-phosphoribosyl-formamido-carboxamide, hk – hexokinase, pfk – phosphofructokinase, gapdh – glyceraldehyde-3-phosphate dehydrogenase, pgk – phosphoglycerate kinase, pyk – phophoenolpyruvate kinase, ldh – lactate dehydrogenase, glud – glutamate dehydrogenase, GOGAT – glutamate synthase, DHFR – dihydrofolate reducatase.

## Enzyme kinetics

Each metabolic pathway is modelled using either Michaelis-Menten (irreversible case) or Haldane (reversible case) enzyme kinetics, for the individual reactions it comprises. The general kinetics can be expressed as follows, where we let $a_0, a_1$ denote the concentrations of the co-substrate pair $A_0$ and $A_1$, respectively and $m_i$ to denote the concentration of $M_i$, the $i$-th metabolite in the pathway.

In the case of a reversible, enzymatic reaction involving a co-substrate and assuming simultaneous binding of both substrates to the enzyme, we have the following reaction scheme:

$$M_{i-1} + A_0 + E \underset{k_{\text{off}}}{\overset{k_{\text{on}}}{\rightleftharpoons}} EM_{i-1}A_0 \underset{k_{\text{off},1}}{\overset{k_{\text{on},1}}{\rightleftharpoons}} EM_iA_1 \underset{k_{\text{off},2}}{\overset{k_{\text{on},2}}{\rightleftharpoons}} M_i + A_1 + E.$$

For this reversible reaction scheme, the rate of production of $M_i$ takes the form

$$v = \frac{E_i L_{i-1} m_{i-1} a_0 - F_i K_i m_i a_1}{K_i L_{i-1} + K_i m_i a_1 + L_{i-1} m_{i-1} a_0}.$$

Likewise, for the reversible enzymatic conversion $M_{i-1} \rightleftharpoons M_i$, we have the following reaction scheme:

$$M_{i-1} + E \underset{k_{\text{off}}}{\overset{k_{\text{on}}}{\rightleftharpoons}} EM_{i-1} \underset{k_{\text{off},1}}{\overset{k_{\text{on},1}}{\rightleftharpoons}} EM_i \underset{k_{\text{off},2}}{\overset{k_{\text{on},2}}{\rightleftharpoons}} M_i + E.$$

The rate of production of $M_i$ is given by

$$v = \frac{E_i L_{i-1} m_{i-1} - F_i K_i m_i}{K_i L_{i-1} + K_i m_i + L_{i-1} m_{i-1}}.$$

In both of these reversible rate equations, the parameters $K$ and $L$ are equivalent to the Haldane coefficients $K_S$ and $K_P$, respectively and are given by

$$K_i = \frac{k_{\text{on},1}k_{\text{on},2} + k_{\text{off}}k_{\text{on},2} + k_{\text{off}}k_{\text{off},1}}{k_{\text{on}}(k_{\text{on},2} + k_{\text{off},1} + k_{\text{on},1})} \quad \text{and} \quad L_{i-1} = \frac{k_{\text{on},1}k_{\text{on},2} + k_{\text{off}}k_{\text{on},2} + k_{\text{off}}k_{\text{off},1}}{k_{\text{off},2}(k_{\text{off},1} + k_{\text{on},1} + k_{\text{off}})}. \tag{5}$$

When there are two substrates that take part in the reaction, the $k_{\text{on}}$ and $k_{\text{off},2}$ parameters are composite parameters composed of two binding coefficients, one for each substrate. This does not affect the derivations, so for convenience we use $K_S$ and $K_P$.

The parameters $E$ and $F$ correspond to the Haldane coefficients $k_{\text{cat}+}$ and $k_{\text{cat}-}$, multiplied by the total enzyme concentration (denoted $E_{\text{tot}}$, below) and are given by

$$E_i = E_{\text{tot}} \frac{k_{\text{on},1}k_{\text{on},2}}{k_{\text{on},2} + k_{\text{off},1} + k_{\text{on},1}} \quad \text{and} \quad F_i = E_{\text{tot}} \frac{k_{\text{off}}k_{\text{off},1}}{k_{\text{off}} + k_{\text{off},1} + k_{\text{on},1}}. \tag{6}$$

For the irreversible enzymatic reaction, the reaction schemes simplify to:

$$M_{i-1} + A_0 + E \underset{k_{\text{off}}}{\overset{k_{\text{on}}}{\rightleftharpoons}} EM_{i-1}A_0 \overset{k_{\text{cat}}}{\longrightarrow} M_i + A_1 + E.$$

$$M_{i-1} + E \underset{k_{\text{off}}}{\overset{k_{\text{on}}}{\rightleftharpoons}} EM_{i-1} \overset{k_{\text{cat}}}{\longrightarrow} M_i + E.$$

And the rate of production for the two cases are given by

$$v = \frac{E_i\, m_{i-1}\, a_0}{K_i + m_{i-1} a_0}, \qquad \text{and} \qquad v = \frac{E_i\, m_{i-1}}{K_i + m_{i-1}},$$

where $E_i$ is the catalytic rate coefficient of the $i$-th enzyme multiplied by its total concentration, and $K_i$ is its Michaelis-Menten coefficient. Again, when there are two substrates, the $k_{\text{on}}$ parameter is a composite parameter composed of two binding coefficients, one for each substrate. As in the reversible case, this does not affect the derivations, so we use $K_i$ for convenience. Influx and outflux follow non-enzymatic reaction kinetics, with reaction rate constants as indicated by the labels of the reactions.

## Analytical approach

Our mathematical analysis is primarily concerned with finding conditions on the kinetic parameters, if any, that imply that the system has a positive steady state. This is different than system reduction, for example as done in the analyses leading to Michaelis-Menten kinetics. Our analysis distinctively solves the entire system for steady states and determines conditions on kinetic parameters to satisfy the steady state equations.

Thus, for each of the metabolic pathway motifs we consider, we build the ODEs defining the rates of change of variables, find the conservation laws among variables, and consider a system of equations whose solutions are the steady states of the ODEs constrained by the conservation laws. We then follow one of two strategies. We first attempt to solve all equations for all concentrations. For some systems, we readily get an expression in terms of the parameters of the system. For other systems, this approach is not possible. In this case, using all equations in the system but one, we solve for the steady states of all concentrations but one. This gives all concentrations in terms of the remaining concentration, say $x$. Plugging these expressions in the remaining equation of the system, we obtain a final equation whose solutions characterize the steady states of the system. We need then to study when the solutions obtained this way are positive.

We are also interested in proving if a given system has a positive steady state for all parameter combinations, and that this steady state is stable. When there is one positive steady state, we find the Hurwtiz determinants associated with the characteristic polynomial of the Jacobian of the system of ODEs, evaluated at the steady state. If these are all positive, then the steady state is asymptotically stable (*Torres and Feliu, 2021*).

To decide on the existence of a steady state, throughout the analysis, we will use repeatedly the following lemma, which is a consequence of the well-known Descartes' rule of signs.

Lemma 1. Let $p(x)$ be a univariate polynomial of degree two, with negative leading term. If at some value $T$, we have $p(T) > 0$, then $p$ has a root in the interval $(0, T)$ if and only if the independent term of $p$ is negative.

*Proof.* The Descartes' rule of signs establishes that the number of positive roots of a polynomial cannot exceed the number $\tau$ of sign changes in the sequence of coefficients ignoring zero coefficients, and the difference between $\tau$ and the number of positive roots is an even number. As the polynomial $p$ in the statement attains positive values, it must have some positive coefficient. Furthermore, as the degree two polynomial has negative leading term, the sequence of the sign of terms (when terms are ordered from lowest exponent to highest) is one of the following $++-$, $+--$, $-+-$, $+0-$, $0+-$.

If the independent term is positive or zero, then the sign sequence is one of $++-$, $+--$, $+0-$, $0+-$. In this case, there is one sign change in the sequence, and it follows that the polynomial has exactly one positive root. As $p(0) > 0$, $p(T) > 0$ and $p$ becomes negative as $x$ goes to $+\infty$, the root must be in the interval $(T, +\infty)$.

If the independent term is negative, then the sign sequence is $-+-$. The polynomial is negative both at 0 and at $+\infty$. As $p(T) > 0$, there must be a positive root in $(0, T)$ and one in $(T, +\infty)$, and there cannot be more by the Descartes' rule of signs. From this, the statement of the lemma follows.

## Appendix 2

### Considering co-substrate pool size

We first consider cycling of a generic co-substrate $A_0$ and $A_1$, with biosynthesis and degradation of both forms.

$$0 \underset{k_2}{\overset{k_1}{\rightleftharpoons}} A_0 \qquad 0 \underset{k_4}{\overset{k_3}{\rightleftharpoons}} A_1 \qquad A_0 \underset{k_6}{\overset{k_5}{\rightleftharpoons}} A_1.$$

We suppose that the biosynthesis occurs at a constant rate, while degradation and cycling are proportional to the concentration of the relevant chemical species. Writing $a_0 = [A_0]$ and $a_1 = [A_1]$, the differential equations for these concentrations are:

$$\frac{da_0}{dt} = k_1 - (k_2 + k_5)a_0 + k_6 a_1$$
$$\frac{da_1}{dt} = k_3 - (k_4 + k_6)a_1 + k_5 a_0.$$

Since all the terms are linear or constant, the steady state values are the solutions of the linear equation:

$$\begin{pmatrix} -(k_2 + k_5) & k_6 \\ k_5 & -(k_4 + k_6) \end{pmatrix} \begin{pmatrix} a_0 \\ a_1 \end{pmatrix} = \begin{pmatrix} -k_1 \\ -k_3 \end{pmatrix}.$$

The steady states are then found to be:

$$a_0 = \frac{k_1 k_4 + k_1 k_6 + k_3 k_6}{k_4 k_5 + k_2 k_4 + k_2 k_6}, \qquad a_1 = \frac{k_2 k_3 + k_3 k_5 + k_1 k_5}{k_4 k_5 + k_2 k_4 + k_2 k_6}.$$

If we consider the case where the synthesis and degradation rates of the different forms of the co-substrate (i.e. cycled metabolite) are the same, i.e. $k_1 = k_3 = k_s$ and $k_2 = k_4 = k_d$, these equations simplify to:

$$a_0 = \frac{k_s(k_d + 2k_6)}{k_d(k_d + k_5 + k_6)}, \qquad a_1 = \frac{k_s(k_d + 2k_5)}{k_d(k_d + k_5 + k_6)},$$

and the eigenvalues of the Jacobian of the system evaluated at this steady state are always real and negative. When $k_d$ is sufficiently small compared to co-substrate conversion rates, it can be safely neglected in the brackets, resulting in the expression of steady state formulas as:

$$a_0 = \frac{2k_6 k_s/k_d}{k_5 + k_6}, \qquad a_1 = \frac{2k_5 k_s/k_d}{k_5 + k_6},$$

We can compare the above expressions with those obtained from the case, where we assume a constant pool size of the cycled metabolite (i.e. $k_1 = k_2 = k_3 = k_4 = 0$). In that case, the steady states are $a_0 = Tk_6/(k_5 + k_6)$ and $a_1 = Tk_5/(k_5 + k_6)$, where $T$ is the total pool size. Thus, under the limit of degradation rates being much smaller than conversion rates, the two cases will be identical and co-substrates will act as a conserved moiety for the rest of the metabolic system.

If we now assume that the cycling of co-substrates is an enzymatic reaction and make the same simplifying assumptions as above that $k_1 = k_3 = k_s$ and $k_2 = k_4 = k_d$, the ODEs for the system are:

$$\frac{da_0}{dt} = k_s - k_d a_0 - \frac{a_0 E_a L_a - a_1 F_a K_a}{a_0 L_a + a_1 K_a + K_a L_a},$$

$$\frac{da_1}{dt} = k_s - k_d a_1 + \frac{a_0 E_a L_a - a_1 F_a K_a}{a_0 L_a + a_1 K_a + K_a L_a}.$$

The only real and positive steady state is now found to be:

$$a_0 = (K_a L_a k_d + K_a (F_a + 3k_s) + L_a (E_a - k_s)$$

$$- \sqrt{-4 K_a k_s (K_a - L_a)(2(F_a + k_s) + k_d L_a) + (F_a K_a + 3 K_a k_s + E_a L_a + K_a k_d L_a - k_s L_a)^2})/(2 k_d (K_a - L_a)),$$

$$a_1 = (- K_a L_a k_d - K_a (F_a - 3k_s) - L_a (E_a + 3k_s)$$

$$+ \sqrt{-4 K_a k_s (K_a - L_a)(2(F_a + k_s) + k_d L_a) + (F_a K_a + 3 K_a k_s + E_a L_a + K_a k_d L_a - k_s L_a)^2})/(2 k_d (K_a - L_a)).$$

This is stable as long as all parameters are positive. Note that in the case of $K_a = L_a$, the steady state solutions converge to a real number less than infinity by l'Hopital's Rule. Also, note that the sum $a_0 + a_1$ is constant as in the non-enzymatic case presented above. Thus, whether the metabolite cycling is considered as a non-enzymatic or enzymatic (i.e. following Michaelis-Menten kinetics) reaction, the co-substrates will act as a conserved moiety for the rest of the metabolic system in both cases.

## Appendix 3

### Single reaction models

In this section, we derive results for a single reaction of two metabolites, involving co-substrate cycling or not, as presented in the main text.

### Enzymatic reaction with co-substrate cycling

We first reconsider the case where all reactions, including the off-pathway cycling, are enzymatic (hence are modelled with Michaelis-Menten kinetics). The reactions are:

$$0 \xrightarrow{k_{\text{in}}} M_0 \qquad M_0 + \text{A}_0 \rightleftharpoons M_1 + \text{A}_1 \qquad M_1 \xrightarrow{k_{\text{out}}} 0 \qquad \text{A}_0 \rightleftharpoons \text{A}_1.$$

This corresponds to the motif depicted in **Figure 1A** of the main text, and the resulting ODEs are:

$$\frac{dm_0}{dt} = k_{\text{in}} - \frac{E_1 L_0 m_0 a_0 - F_1 K_1 m_1 a_1}{K_1 m_1 a_1 + L_0 a_0 m_0 + K_1 L_0}$$

$$\frac{dm_1}{dt} = \frac{E_1 L_0 m_0 a_0 - F_1 K_1 m_1 a_1}{K_1 m_1 a_1 + L_0 a_0 m_0 + K_1 L_0} - k_{\text{out}} m_1$$

$$\frac{da_0}{dt} = -\frac{E_1 L_0 m_0 a_0 - F_1 K_1 m_1 a_1}{K_1 m_1 a_1 + L_0 a_0 m_0 + K_1 L_0} - \frac{E_a L_a a_0 - F_a K_a a_1}{K_a a_1 + L_a a_0 + K_a L_a}$$

$$\frac{da_1}{dt} = \frac{E_1 L_0 m_0 a_0 - F_1 K_1 m_1 a_1}{K_1 m_1 a_1 + L_0 a_0 m_0 + K_1 L_0} + \frac{E_a L_a a_0 - F_a K_a a_1}{K_a a_1 + L_a a_0 + K_a L_a}.$$

This ODE system has one conservation law, namely the sum of $a_0$ and $a_1$ is constant:

$$a_0 + a_1 = A_{\text{tot}}.$$

The steady states of the system are:

$$m_0 = \frac{K_1 k_{\text{in}}}{(E_1 - k_{\text{in}})\left(F_a A_{\text{tot}} - k_{\text{in}}(L_a + A_{\text{tot}})\right)} \alpha,$$

$$m_1 = \frac{k_{\text{in}}}{k_{\text{out}}},$$

$$a_0 = \frac{K_a\left(F_a A_{\text{tot}} - k_{\text{in}}(L_a + A_{\text{tot}})\right)}{K_a(F_a - k_{\text{in}}) + L_a(E_a + k_{\text{in}})},$$

$$a_1 = \frac{L_a\left(K_a k_{\text{in}} + T(E_a + k_{\text{in}})\right)}{K_a(F_a - k_{\text{in}}) + L_a(E_a + k_{\text{in}})}.$$

where, we introduced the composite parameter $\alpha$, as follows:

$$\alpha = \frac{k_{\text{out}} L_0\left(K_a(F_a - k_{\text{in}}) + L_a(E_a + k_\in)\right) + (F_1 + k_{\text{in}})L_a(K_a k_{\text{in}} + k_{\text{in}} A_{\text{tot}} + E_a A_{\text{tot}})}{K_a L_0 k_{\text{out}}}. \tag{7}$$

For the steady state equations given above to be positive, the following conditions must be satisfied:

$$k_{\text{in}} < E_1 \quad \text{and} \quad k_{\text{in}} < \frac{F_a A_{\text{tot}}}{L_a + A_{\text{tot}}}. \tag{8}$$

Note that as $\frac{F_a A_{\text{tot}}}{L_a + A_{\text{tot}}} < F_a$, the second condition readily implies $k_{\text{in}} < F_a$ and $\alpha$ is positive. When there is a positive steady state, then it is asymptotically stable.

If the main pathway is irreversible, but the co-substrate reaction is still reversible, the ODEs describing the system dynamics are:

$$\frac{dm_0}{dt} = k_{in} - \frac{E_1 m_0 a_0}{K_1 + m_0 a_0},$$

$$\frac{dm_1}{dt} = \frac{E_1 m_0 a_0}{K_1 + m_0 a_0} - k_{out} m_1,$$

$$\frac{da_0}{dt} = -\frac{E_1 m_0 a_0}{K_1 + m_0 a_0} - \frac{E_a L_a a_0 - F_a K_a a_1}{K_a a_1 + L_a a_0 + K_a L_a},$$

$$\frac{da_1}{dt} = \frac{E_1 m_0 a_0}{K_1 + m_0 a_0} + \frac{E_a L_a a_0 - F_a K_a a_1}{K_a a_1 + L_a a_0 + K_a L_a}.$$

The steady state of the system is:

$$m_0 = \frac{K_1 k_{in}}{(E_1 - k_{in})(F_a A_{tot} - k_{in}(L_a + A_{tot}))} \alpha,$$

$$m_1 = \frac{k_{in}}{k_{out}},$$

$$a_0 = \frac{K_a(F_a A_{tot} - k_{in}(L_a + A_{tot}))}{K_a (F_a - k_{in}) + L_a (E_a + k_{in})},$$

$$a_1 = \frac{L_a (K_a k_{in} + A_{tot}(E_a + k_{in}))}{K_a (F_a - k_{in}) + L_a (E_a + k_{in})}.$$

where the composite parameter $\alpha$ is defined (differently to the reversible case) as:

$$\alpha = \frac{K_a (F_a - k_{in}) + L_a (E_a + k_{in})}{K_a}. \tag{9}$$

For this to be positive the same conditions as in the reversible case must be satisfied:

$$k_{in} < E_1 \quad \text{and} \quad k_{in} < \frac{F_a A_{tot}}{L_a + A_{tot}}. \tag{10}$$

When these are satisfied, $\alpha$ is positive, and the positive steady state is asymptotically stable.

If the main pathway is irreversible, and the co-substrate reaction only flows from $A_1$ to $A_0$, the ODEs describing the system dynamics are:

$$\frac{dm_0}{dt} = k_{in} - \frac{E_1 m_0 a_0}{K_1 + m_0 a_0},$$

$$\frac{dm_1}{dt} = \frac{E_1 m_0 a_0}{K_1 + m_0 a_0} - k_{out} m_1,$$

$$\frac{da_0}{dt} = -\frac{E_1 m_0 a_0}{K_1 + m_0 a_0} + \frac{F_a a_1}{L_a + a_1},$$

$$\frac{da_1}{dt} = \frac{E_1 m_0 a_0}{K_1 + m_0 a_0} - \frac{F_a a_1}{L_a + a_1}.$$

The steady state of the system is:

$$m_0 = \frac{K_1 k_{\text{in}}}{(E_1 - k_{\text{in}})(F_a A_{\text{tot}} - k_{\text{in}}(L_a + A_{\text{tot}}))} \alpha,$$

$$m_1 = \frac{k_{\text{in}}}{k_{\text{out}}},$$

$$a_0 = A_{\text{tot}} - \frac{k_{\text{in}} L_a}{F_a - k_{\text{in}}},$$

$$a_1 = \frac{k_{\text{in}} L_a}{F_a - k_{\text{in}}}.$$

where the composite parameter $\alpha$ is defined as:

$$\alpha = F_a - k_{\text{in}} \tag{11}$$

For these steady states to be positive the same conditions as in the other two cases must be satisfied:

$$k_{\text{in}} < E_1 \quad \text{and} \quad k_{\text{in}} < \frac{F_a A_{\text{tot}}}{L_a + A_{\text{tot}}}. \tag{12}$$

When these are satisfied, the positive steady state is asymptotically stable.

## Enzymatic reaction with co-substrate cycling, biosynthesis and degradation

To extend the previous analysis we consider the same simple scenario where a pathway involves a single co-substrate consuming reaction and a back reaction re-generating the co-substrate, with the addition of synthesis and degradation of the co-substrate forms. This system is comprised of the following reactions:

$$0 \xrightarrow{k_{\text{in}}} M_0 \qquad M_0 + A_0 \rightleftharpoons M_1 + A_1 \qquad M_1 \xrightarrow{k_{\text{out}}} 0$$
$$A_0 \underset{k_6}{\overset{k_5}{\rightleftharpoons}} A_1 \qquad 0 \underset{k_8}{\overset{k_7}{\rightleftharpoons}} A_0 \qquad 0 \underset{k_{10}}{\overset{k_9}{\rightleftharpoons}} A_1 \qquad .$$

The resulting system of ODEs is:

$$\frac{dm_0}{dt} = k_{\text{in}} - \frac{E_1 L_0 m_0 a_0 - F_1 K_1 m_1 a_1}{K_1 m_1 a_1 + L_0 a_0 m_0 + K_1 L_0}$$

$$\frac{dm_1}{dt} = \frac{E_1 L_0 m_0 a_0 - F_1 K_1 m_1 a_1}{K_1 m_1 a_1 + L_0 a_0 m_0 + K_1 L_0} - k_{\text{out}} m_1$$

$$\frac{da_0}{dt} = -\frac{E_1 L_0 m_0 a_0 - F_1 K_1 m_1 a_1}{K_1 m_1 a_1 + L_0 a_0 m_0 + K_1 L_0} - (k_5 + k_8) a_0 + k_6 a_1 + k_7$$

$$\frac{da_1}{dt} = \frac{E_1 L_0 m_0 a_0 - F_1 K_1 m_1 a_1}{K_1 m_1 a_1 + L_0 a_0 m_0 + K_1 L_0} + k_5 a_0 - (k_6 + k_{10}) a_1 + k_9.$$

The steady state concentrations are then given by:

$$a_0 = \frac{k_{10}(k_7 - k_{\text{in}}) + k_6(k_7 + k_9)}{k_{10}k_5 + k_{10}k_8 + k_6k_8}$$

$$a_1 = \frac{k_8(k_9 + k_{\text{in}}) + k_5(k_7 + k_9)}{k_{10}k_5 + k_{10}k_8 + k_6k_8}$$

$$m_0 = \frac{K_1 k_{\text{in}} \left((F_1 + k_{\text{in}})(k_8(k_9 + k_{\text{in}}) + k_5(k_7 + k_9)) + k_{\text{out}}L_0(k_{10}k_5 + k_{10}k_8 + k_6k_8)\right)}{L_0 k_{\text{out}} \left(E_1 - k_{\text{in}}\right) \left(k_{10}(k_7 - k_{\text{in}}) + k_6(k_7 + k_9)\right)}$$

$$m_1 = \frac{k_{\text{in}}}{k_{\text{out}}}.$$

These expressions are positive if and only if

$$k_{\text{in}} < k_7 + \frac{k_6}{k_{10}}(k_7 + k_9) \qquad \text{and} \qquad k_{\text{in}} < E_1.$$

If the main path is irreversible, similarly to the previous section, the ODEs describing the system's dynamics, are:

$$\frac{dm_0}{dt} = k_{\text{in}} - \frac{E_1 m_0 a_0}{K_1 + m_0 a_0}$$

$$\frac{dm_1}{dt} = \frac{E_1 m_0 a_0}{K_1 + m_0 a_0} - k_{\text{out}}m_1$$

$$\frac{da_0}{dt} = -\frac{E_1 m_0 a_0}{K_1 + m_0 a_0} - (k_5 + k_8)a_0 + k_6 a_1 + k_7$$

$$\frac{da_1}{dt} = \frac{E_1 m_0 a_0}{K_1 + m_0 a_0} + k_5 a_0 - (k_6 + k_{10})a_1 + k_9.$$

The steady state in this case is:

$$a_0 = \frac{k_{10}(k_7 - k_{\text{in}}) + k_6(k_7 + k_9)}{k_{10}k_5 + k_{10}k_8 + k_8k_6}$$

$$a_1 = \frac{k_8(k_9 + k_{\text{in}}) + k_5(k_7 + k_9)}{k_{10}k_5 + k_{10}k_8 + k_8k_6}$$

$$m_0 = \frac{K_1 k_{\text{in}}(k_{10}k_5 + k_{10}k_8 + k_8k_6)}{(E_1 - k_{\text{in}})(k_{10}(k_7 - k_{\text{in}}) + k_6(k_7 + k_9))}$$

$$m_1 = \frac{k_{\text{in}}}{k_{\text{out}}}.$$

These expressions are positive if and only if

$$k_{\text{in}} < k_7 + \frac{k_6}{k_{10}}(k_7 + k_9) \qquad \text{and} \qquad k_{\text{in}} < E_1.$$

Comparing these results with those of Subsection C.1, we see some similar themes. Firstly, the total amount of co-substrate $a_0 + a_1$ at steady state is a constant, even when it is not explicitly considered to be a conserved moiety. Secondly, one of the conditions for a positive steady state is $k_{\text{in}} < E_1$, and any other conditions take the form $k_{\text{in}} < f$, where $f$ is a function of the parameters controlling synthesis, degradation and the turnover of the co-substrate. Thus, in the analysis that follows, we only consider the case of conserved co-substrate, as adding synthesis and degradation only affects the quantitative results, not the qualitative behaviour.

## Enzymatic reaction without co-substrate cycling

Considering an enzymatic reaction without co-substrates, the reactions are:

$$0 \xrightarrow{k_{\text{in}}} M_0 \qquad M_0 \underset{k_2}{\overset{k_1}{\rightleftharpoons}} M_1 \qquad M_1 \xrightarrow{k_{\text{out}}} 0.$$

The resulting system of ODEs is:

$$\frac{dm_0}{dt} = k_{\text{in}} - \frac{E_1 L_0 m_0 - F_1 K_1 m_1}{K_1 m_1 + L_0 m_0 + K_1 L_0}$$

$$\frac{dm_1}{dt} = \frac{E_1 L_0 m_0 - F_1 K_1 m_1}{K_1 m_1 + L_0 m_0 + K_1 L_0} - k_{\text{out}} m_1.$$

In this case, there is no conservation law, and the steady states of the system are:

$$m_0 = \frac{K_1 k_{\text{in}} \left( F_1 + k_{\text{in}} + k_{\text{out}} L_0 \right)}{k_{\text{out}} L_0 \left( E_1 - k_{\text{in}} \right)}, \qquad m_1 = \frac{k_{\text{in}}}{k_{\text{out}}}.$$

These expressions are positive if and only if $k_{\text{in}} < E_1$ and the steady state is asymptotically stable when this holds.

If the main path is irreversible, the ODEs describing system dynamics are:

$$\frac{dm_0}{dt} = k_{\text{in}} - \frac{E_1 m_0}{K_1 + m_0}$$

$$\frac{dm_1}{dt} = \frac{E_1 m_0}{K_1 + m_0} - k_{\text{out}} m_1,$$

and the steady state is:

$$m_0 = \frac{K_1 k_{\text{in}}}{E_1 - k_{\text{in}}}, \qquad m_1 = \frac{k_{\text{in}}}{k_{\text{out}}}.$$

Again, expressions are positive if and only if $k_{\text{in}} < E_1$ and the steady state is asymptotically stable when this holds.

From this we see that the condition $k_{\text{in}} < E_1$ for stability in the enzymatic system with co-substrate cycling is a result of the reaction being enzymatic.

## Non-enzymatic reaction with co-substrate cycling

Consider a theoretical case of non-enzymatic reactions for the main reaction:

$$0 \xrightarrow{k_{\text{in}}} M_0 \qquad M_0 + A_0 \underset{k_2}{\overset{k_1}{\rightleftharpoons}} M_1 A_1 \qquad M_1 \xrightarrow{k_{\text{out}}} 0 \qquad A_0 \underset{k_4}{\overset{k_3}{\rightleftharpoons}} A_1.$$

The resulting system of ODEs, describing system dynamics, is:

$$\frac{dm_0}{dt} = k_{\text{in}} - k_1 m_0 a_0 + k_2 m_1 a_1$$

$$\frac{dm_1}{dt} = k_1 m_0 a_0 - k_2 m_1 a_1 - k_{\text{out}} m_1$$

$$\frac{da_0}{dt} = -k_1 m_0 a_0 + k_2 a_1 m_1 - k_3 a_0 + k_4 a_1$$

$$\frac{da_1}{dt} = k_1 m_0 a_0 - k_2 m_1 a_1 + k_3 a_0 - k_4 a_1$$

There is the conservation law

$$a_0 + a_1 = A_{\text{tot}}.$$

The steady state of the system is:

$$m_0 = \frac{k_{\text{in}}(k_2(k_{\text{in}} + k_3 A_{\text{tot}}) + k_{\text{out}}(k_3 + k_4))}{k_1 k_{\text{out}}(k_4 A_{\text{tot}} - k_{\text{in}})}$$

$$m_1 = \frac{k_{\text{in}}}{k_{\text{out}}},$$

$$a_0 = \frac{k_4 A_{\text{tot}} - k_{\text{in}}}{k_3 + k_4},$$

$$a_1 = \frac{k_3 A_{\text{tot}} + k_{\text{in}}}{k_3 + k_4}.$$

These expressions are positive if and only if $k_{\text{in}} < k_4 A_{\text{tot}}$, and when this is satisfied the steady state is asymptotically stable.

If the system reactions are irreversible, the ODEs describing the system dynamics are:

$$\frac{dm_0}{dt} = k_{\text{in}} - k_1 m_0 a_0$$

$$\frac{dm_1}{dt} = k_1 m_0 a_0 - k_{\text{out}} m_1$$

$$\frac{da_0}{dt} = -k_1 m_0 a_0 + k_4 a_1$$

$$\frac{da_1}{dt} = k_1 m_0 a_0 - k_4 a_1$$

and the steady state is:

$$m_0 = \frac{k_4 k_{\text{in}}}{k_1(k_4 A_{\text{tot}} - k_{\text{in}})},$$

$$m_1 = \frac{k_{\text{in}}}{k_{\text{out}}},$$

$$a_0 = \frac{k_4 A_{\text{tot}} - k_{\text{in}}}{k_4},$$

$$a_1 = \frac{k_{\text{in}}}{k_4}.$$

As in the reversible case, these expressions are positive when $k_{\text{in}} < k_4 A_{\text{tot}}$ and the steady state is asymptotically stable when this holds.

Hence, we see that introducing co-substrate cycling always introduces a new constraint into the system, even in this simple case where the cycling is not enzymatic.

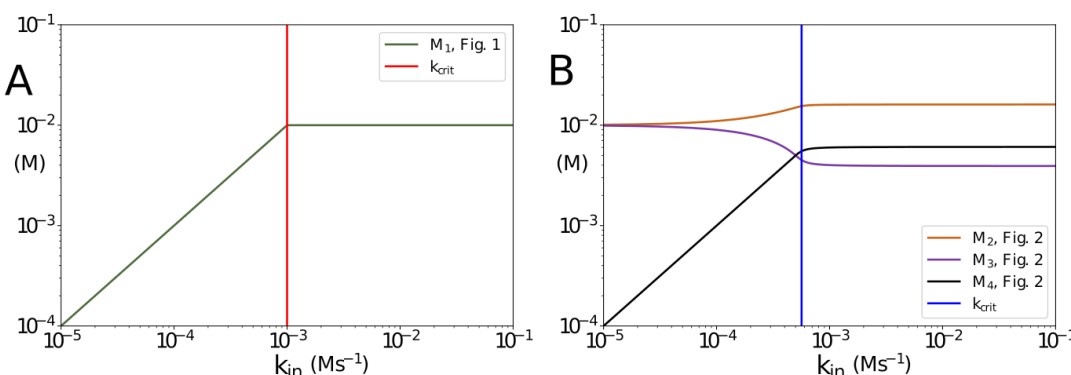

**Appendix 3—figure 1.** Concentrations of non-accumulating metabolites vs $k_{in}$ in the models presented in *Figures 1 and 2* of the main text. (**A**) Concentration of non-accumulating metabolite $M_1$ from *Figure 1*. Red line shows the critical $k_{in}$ value, after which the concentration of $M_1$ remains constant (**B**) Concentrations of non-accumulating metabolites from the models presented in Figure 2. Blue line shows the threshold value for $k_{in}$. Once the threshold value for $k_{in}$ is reached the concentrations of the non-accumulating metabolites do not change. $A_{tot} = 10^{-1}$ in panel (**A**) and $10^{-6}$ in panel (**B**). All other parameters are the same as their counterparts in the main text.

## Appendix 4

### Linear, arbitrary length, pathway model with co-substrate cycling

We consider next a linear, generic pathway of $n+1$ metabolites, and comprising the following reaction mechanism:

$$0\xrightarrow{k_{\text{in}}}M_0 \qquad M_0 + A_0 \rightleftharpoons M_1 + A_1 \qquad M_{n-1} + A_1 \rightleftharpoons M_n + A_0$$

$$M_1 \rightleftharpoons M_2 \cdots \quad M_{i-1} \rightleftharpoons M_i \rightleftharpoons M_{i+1} \cdots \rightleftharpoons M_{n-1} \qquad M_n \xrightarrow{k_{\text{out}}}0. \tag{13}$$

#### Steady states of the linear pathway model: the case with n=3

We first consider the dynamics of model (**13**) with $n=3$, as this is the minimal pathway length where the system makes biochemical sense. This system has the form:

$$0\xrightarrow{k_{\text{in}}}M_0 \quad M_0 + A_0 \rightleftharpoons M_1 + A_1 \quad M_1 \rightleftharpoons M_2$$

$$M_3 \xrightarrow{k_{\text{out}}}0 \quad M_2 + A_1 \rightleftharpoons M_3 + A_0. \tag{14}$$

Considering reversible reaction kinetics, the evolution of the concentrations of the species in time is modelled by the following ODE system:

$$\frac{dm_0}{dt} = k_{\text{in}} - \frac{E_1 L_0 m_0 a_0 - F_1 K_1 m_1 a_1}{K_1 m_1 a_1 + L_0 m_0 a_0 + K_1 L_0}$$

$$\frac{dm_1}{dt} = \frac{E_1 L_0 m_0 a_0 - F_1 K_1 m_1 a_1}{K_1 m_1 a_1 + L_0 m_0 a_0 + K_1 L_0} - \frac{E_2 L_1 m_1 - F_2 K_2 m_2}{K_2 m_2 + L_1 m_1 + K_2 L_1}$$

$$\frac{dm_2}{dt} = \frac{E_2 L_1 m_1 - F_2 K_2 m_2}{K_2 m_2 + L_1 m_1 + K_2 L_1} - \frac{E_3 L_2 m_2 a_1 - F_3 K_3 m_3 a_0}{K_3 m_3 a_0 + L_2 m_2 a_1 + K_3 L_2}$$

$$\frac{dm_3}{dt} = \frac{E_3 L_2 m_2 a_1 - F_3 K_3 m_3 a_0}{K_3 m_3 a_0 + L_2 m_2 a_1 + K_3 L_2} - k_{\text{out}} m_3$$

$$\frac{da_0}{dt} = -\frac{E_1 L_0 m_0 a_0 - F_1 K_1 m_1 a_1}{K_1 m_1 a_1 + L_0 m_0 a_0 + K_1 L_0} + \frac{E_3 L_2 m_2 a_1 - F_3 K_3 m_3 a_0}{K_3 m_3 a_0 + L_2 m_2 a_1 + K_3 L_2}$$

$$\frac{da_1}{dt} = \frac{E_1 L_0 m_0 a_0 - F_1 K_1 m_1 a_1}{K_1 m_1 a_1 + L_0 m_0 a_0 + K_1 L_0} - \frac{E_3 L_2 m_2 a_1 - F_3 K_3 m_3 a_0}{K_3 m_3 a_0 + L_2 m_2 a_1 + K_3 L_2}.$$

The system has two conservation laws:

$$a_0 + a_1 = A_{\text{tot}}, \qquad m_1 + m_2 + a_0 = W. \tag{15}$$

Solving recursively the steady state equations given by $\frac{dm_0}{dt} + \cdots + \frac{dm_3}{dt} = 0$, $\frac{dm_3}{dt} = 0$, $\frac{dm_2}{dt} + \frac{dm_3}{dt} = 0$, $\frac{dm_1}{dt} + \frac{dm_2}{dt} + \frac{dm_3}{dt} = 0$ for $m_3, m_2, m_1, m_0$ and the first conservation law, we obtain

$$m_0 = \frac{K_1 \left((F_1 + k_{\text{in}})m_1 a_1 + L_0 k_{\text{in}}\right)}{a_0 L_0 \left(E_1 - k_{\text{in}}\right)},$$

$$m_1 = \frac{K_2 \left((F_2 + k_{\text{in}})m_2 + L_1 k_{\text{in}}\right)}{L_1 \left(E_2 - k_{\text{in}}\right)},$$

$$m_2 = \frac{k_{\text{in}} K_3 \left((F_3 + k_{\text{in}})a_0 + L_2 k_{\text{out}}\right)}{k_{\text{out}} a_1 L_2 \left(E_3 - k_{\text{in}}\right)},$$

$$m_3 = \frac{k_{\text{in}}}{k_{\text{out}}}, \qquad a_1 = A_{\text{tot}} - a_0. \tag{16}$$

By substituting recursively $m_2$ in $m_1$ and $m_1$ in $m_0$, we obtain expressions of all variables at steady state in terms of $a_0$. For these to be positive, it needs to hold that $0 < a_0 < A_{\text{tot}}$, and further

$$k_{\text{in}} < \min(E_1, E_2, E_3). \tag{17}$$

We substitute (**16**) into the remaining conservation law, and obtain a polynomial in $a_0$ whose roots in the interval $(0, T)$ are in one-to-one correspondence with the positive steady states, provided (**17**) holds. The polynomial is

$$
\begin{aligned}
p(a_0) &= -L_1 L_2 k_{\text{out}} \left(E_3 - k_{\text{in}}\right) \left(E_2 - k_{\text{in}}\right) a_0^2 + \left(L_1 L_2 k_{\text{out}} \left(A_{\text{tot}} + W\right) \left(E_3 - k_{\text{in}}\right) \left(E_2 - k_{\text{in}}\right)\right) \\
&\quad - K_2 L_1 L_2 k_{\text{in}} k_{\text{out}} \left(E_3 - k_{\text{in}}\right) + K_3 k_{\text{in}} \left(F_3 + k_{\text{in}}\right) \left(E_2 L_1, + F_2 K_2\right) \\
&\quad + K_3 k_{\text{in}}^2 \left(K_2 - L_1\right) \left(F_3 + k_{\text{in}},\right) )a_0 + L_2 K_2 \, \alpha,
\end{aligned}
$$

where

$$\alpha = k_{\text{in}}(k_{\text{in}} + F_2)K_2 K_3 + k_{\text{in}}A_{\text{tot}}(E_3 - k_{\text{in}})K_2 L_1 + k_{\text{in}}(E_2 - k_{\text{in}})K_3 L_1 - A_{\text{tot}} W(E_3 - k_{\text{in}})(E_2 - k_{\text{in}})L_1. \tag{18}$$

As $a_0 = A_{\text{tot}}$, we have $p(A_{\text{tot}}) > 0$, and hence, by Lemma 1, the system has positive steady states if and only if $\alpha < 0$.

Note that at $k_{\text{in}} = 0$, $\alpha = -A_{\text{tot}} W(E_3 - k_{\text{in}})(E_2 - k_{\text{in}})L_1 < 0$. Hence, for $k_{\text{in}}$ small enough, $\alpha < 0$ and also (**17**) holds, and the system has a positive steady state. The steady state value of $a_0$ is found by solving the quadratic equation $p(a_0) = 0$ and considering the smallest root.

We note that $\alpha$ is a polynomial in $k_{\text{in}}$ of degree 2 with negative independent term. The sign of the leading term depends on the choice of parameters. If the minimum in (**17**) is attained at $k_{\text{in}} = E_2$ or $k_{\text{in}} = E_3$, then at this value of $k_{\text{in}}$, $\alpha$ is positive independently of the rest of the parameters. Hence, in the region of interest $\alpha$ is negative only in an interval of the form $(0, B)$, where $B$ is the smallest positive root of $\alpha$. The bound for positive steady states is now given as

$$k_{\text{in}} < \min(E_1, B).$$

If the minimum is attained at $B$, then, as $k_{\text{in}}$ approaches $B$, the smallest positive root of $p(a_0)$ approaches 0 as $\alpha$ approaches zero. Using this in (**16**), we obtain that

$$m_0 \xrightarrow[k_{\text{in}} \longrightarrow B]{} \infty, \qquad m_1 \xrightarrow[k_{\text{in}} \longrightarrow B]{} \frac{B k_{\text{out}} \left(F_2 K_3 + L_1 A_{\text{tot}} \left(E_3 - B\right) + B K_3\right)}{L_1 A_{\text{tot}} \left(E_3 - B\right) \left(E_2 - B\right)},$$

$$m_2 \xrightarrow[k_{\text{in}} \longrightarrow B]{} \frac{B K_3}{A_{\text{tot}} \left(E_3 - B\right)} \qquad m_3 \xrightarrow[k_{\text{in}} \longrightarrow B]{} \frac{B}{k_{\text{out}}}. \tag{19}$$

If $E_1 < B$, then $k_{\text{in}}$ needs to be smaller than $E_1$. When $k_{\text{in}}$ approaches $E_1$, $a_0$ approaches the root of $p$ when $k_{\text{in}} = E_1$, which is not zero. Then $m_0$ still approaches $\infty$ as it has $E_1 - k_{\text{in}}$ in the denominator, while $m_1, m_2$ converge to some positive value.

## Steady states of the linear pathway model: the general case

We consider the dynamics of model (**13**) with $n$ taking any positive value strictly larger than 1, and positioning the co-substrate recycling in arbitrary places.

$$
\begin{aligned}
&0 \xrightarrow{k_{\text{in}}} M_0 \\
&M_0 \rightleftharpoons M_1 \rightleftharpoons \cdots \rightleftharpoons M_{\ell-1} \rightleftharpoons M_\ell \qquad M_\ell + A_0 \rightleftharpoons M_{\ell+1} + A_1 \\
&M_{\ell+1} \rightleftharpoons \cdots \rightleftharpoons M_{\rho-1} \rightleftharpoons M_\rho \qquad\quad M_\rho + A_1 \rightleftharpoons M_{\rho+1} + A_0 \\
&M_{\rho+1} \rightleftharpoons \cdots \rightleftharpoons M_{n-1} \rightleftharpoons M_n \\
&M_n \xrightarrow{k_{\text{out}}} 0.
\end{aligned} \tag{20}
$$

For the system to make sense biochemically, we require $0 \le \ell < \rho - 1 \le n - 1$.

To write the equations in a simple format, we write

$$x_i = m_i, \qquad i \ne \ell, \rho, \qquad x_\ell = m_\ell a_0, \quad x_\rho = m_\rho a_1$$

and

$$y_i = m_i, \qquad i \neq \ell + 1, \rho + 1, \qquad y_{\ell+1} = m_{\ell+1} a_1, \quad y_{\rho+1} = m_{\rho+1} a_0.$$

We start with the reversible reaction kinetics. Then the associated ODE system becomes:

$$\frac{dm_0}{dt} = k_{\text{in}} - \frac{E_1 L_0 x_0 - F_1 K_1 y_1}{K_1 L_0 + L_0 x_0 + K_1 y_1}$$

$$\vdots \qquad\qquad \vdots$$

$$\frac{dm_i}{dt} = \frac{E_i L_{i-1} x_{i-1} - F_i K_i y_i}{K_i L_{i-1} + K_i y_i + L_{i-1} x_{i-1}} - \frac{E_{i+1} L_i x_i - F_{i+1} K_{i+1} y_{i+1}}{K_{i+1} L_i + K_{i+1} y_{i+1} + L_i x_i} \qquad i = 1, \ldots, n-1$$

$$\vdots \qquad\qquad \vdots$$

$$\frac{dm_n}{dt} = \frac{E_n L_{n-1} x_{n-1} - F_n K_n y_n}{K_n L_{n-1} + K_n y_n + L_{n-1} x_{n-1}} - k_{\text{out}} m_n$$

$$\frac{da_0}{dt} = -\frac{E_{\ell+1} L_\ell x_\ell - F_{\ell+1} K_{\ell+1} y_{\ell+1}}{K_{\ell+1} L_\ell + L_\ell x_\ell + K_{\ell+1} y_{\ell+1}} + \frac{E_{\rho+1} L_\rho x_\rho - F_{\rho+1} K_{\rho+1} y_{\rho+1}}{K_{\rho+1} L_\rho + L_\rho x_\rho + K_{\rho+1} y_{\rho+1}}$$

$$\frac{da_1}{dt} = \frac{E_{\ell+1} L_\ell x_\ell - F_{\ell+1} K_{\ell+1} y_{\ell+1}}{K_{\ell+1} L_\ell + L_\ell x_\ell + K_{\ell+1} y_{\ell+1}} - \frac{E_{\rho+1} L_\rho x_\rho - F_{\rho+1} K_{\rho+1} y_{\rho+1}}{K_{\rho+1} L_\rho + L_\rho x_\rho + K_{\rho+1} y_{\rho+1}}.$$

This ODE system has precisely two conservation laws:

$$a_0 + a_1 = A_{\text{tot}}, \qquad m_{\ell+1} + \cdots + m_\rho + a_0 = W.$$

Note that we have the following equalities:

$$\frac{dm_0}{dt} + \cdots + \frac{dm_i}{dt} = k_{\text{in}} - \frac{E_{i+1} L_i x_i - F_{i+1} K_{i+1} y_{i+1}}{K_{i+1} L_i + K_{i+1} y_{i+1} + L_i x_i}, \qquad i = 0, \ldots, n-1 \tag{21}$$

$$\frac{dm_0}{dt} + \cdots + \frac{dm_n}{dt} = k_{\text{in}} - k_{\text{out}} m_n \tag{22}$$

For $i = 1, \ldots, n-2$, by solving (21) for $x_i$, we obtain the following recursive formulas:

$$x_i = \frac{K_{i+1}(k_{\text{in}} L_i + (k_{\text{in}} + F_{i+1}) y_{i+1})}{(E_{i+1} - k_{\text{in}}) L_i}, \qquad i = 0, \ldots, n-1. \tag{23}$$

Finally, from (22) and the conservation law $a_0 + a_1 = A_{\text{tot}}$, we obtain

$$x_n = \frac{k_{\text{in}}}{k_{\text{out}}}, \qquad a_1 = A_{\text{tot}} - a_0. \tag{24}$$

These expressions are all positive if and only if $0 < a_0 < A_{\text{tot}}$ and $k_{\text{in}} < E_i$ for $i = 1, \ldots, n$. Note that the value of $m_i$ can now be found recursively from $m_n$ using (23), as we show next.

Recall that $0 \leq \ell < \rho - 1 \leq n - 1$. Then, it holds

$$y_i = x_i \epsilon_i, \qquad \epsilon_i = 1, i \neq \ell, \ell+1, \rho, \rho+1, \qquad \epsilon_\ell = \epsilon_{\rho+1}^{-1} = \frac{1}{a_0}, \quad \epsilon_{\ell+1} = \epsilon_\rho^{-1} = a_1.$$

We write for shortness

$$x_i = \frac{z_i + b_i x_{i+1}}{c_i}, \qquad i = 0, \ldots, n-1,$$

where for $i = 0, \ldots, n-1$,

$$z_i = k_{\text{in}} K_{i+1} L_i \qquad b_i = \epsilon_{i+1}(k_{\text{in}} + F_{i+1}) K_{i+1} \qquad c_i = (E_{i+1} - k_{\text{in}}) L_i. \tag{25}$$

Then we claim that

$$x_i = \frac{\left(\sum_{j=i}^{n-1} z_j (b_i \cdots b_{j-1})(c_{j+1} \cdots c_{n-1})\right) + (b_i \cdots b_{n-1}) x_n}{c_i \cdots c_{n-1}}, \qquad i = 1, \ldots, n-1. \tag{26}$$

We prove this by descending induction on $i$. Note that products over an empty index equal 1. For instance, $b_j \cdots b_{j-1}$. For $i = n - 1$, (**26**) agrees with (**23**). Assume the formula is true for some , and we prove it for $i - 1$. To do so, we use (**23**) and the induction hypothesis for $x_i$. For $n - 1 > i - 1 \geq 0$, we have

$$
x_{i-1} \quad = \frac{z_{i-1} + b_{i-1}x_i}{c_{i-1}} = \frac{z_{i-1} + b_{i-1}\frac{\left(\sum_{j=i}^{n-1} z_j(b_i\cdots b_{j-1})(c_{j+1}\cdots c_{n-1})\right) + (b_i\cdots b_{n-1})x_n}{c_i\cdots c_{n-1}}}{c_{i-1}}
$$

$$
= \frac{z_{i-1}c_i\cdots c_{n-1} + b_{i-1}\left(\left(\sum_{j=i}^{n-1} z_j(b_i\cdots b_{j-1})(c_{j+1}\cdots c_{n-1})\right) + (b_i\cdots b_{n-1})x_n\right)}{c_{i-1}c_i\cdots c_{n-1}}
$$

$$
= \frac{z_{i-1}c_i\cdots c_{n-1} + \left(\sum_{j=i}^{n-1} z_j(b_{i-1}\cdots b_{j-1})(c_{j+1}\cdots c_{n-1})\right) + (b_{i-1}\cdots b_{n-1})x_n}{c_{i-1}\cdots c_{n-1}}
$$

$$
= \frac{\left(\sum_{j=i-1}^{n-1} z_j(b_{i-1}\cdots b_{j-1})(c_{j+1}\cdots c_{n-1})\right) + (b_{i-1}\cdots b_{n-1})x_n}{c_{i-1}\cdots c_{n-1}}.
$$

This is (**26**) for $i - 1$. Hence, the formula holds. Finally, we can use that $x_n = m_n = \frac{k_{\text{in}}}{k_{\text{out}}}$ to obtain:

$$
x_i = \frac{k_{\text{out}}\left(\sum_{j=i}^{n-1} z_j(b_i\cdots b_{j-1})(c_{j+1}\cdots c_{n-1})\right) + (b_i\cdots b_{n-1})k_{\text{in}}}{k_{\text{out}}\,c_i\cdots c_{n-1}}, \qquad i = 0, \ldots, n - 1. \tag{27}
$$

Let

$$
\bar{b}_{u,j} := \prod_{i=u}^{j}(k_{\text{in}} + F_{i+1})K_{i+1}, \qquad \bar{c}_{u,j} := c_u\cdots c_j = \prod_{i=u}^{j}(E_{i+1} - k_{\text{in}})L_i.
$$

This gives:

$$
m_i = \frac{k_{\text{out}}\left(\sum_{j=i}^{n-1} z_j(b_i\cdots b_{j-1})\bar{c}_{j+1,n-1}\right) + (b_i\cdots b_{n-1})k_{\text{in}}}{k_{\text{out}}\,\bar{c}_{i,n-1}}, \qquad i \neq \ell, \rho, \tag{28}
$$

$$
m_\ell = \frac{k_{\text{out}}\left(\sum_{j=i}^{n-1} z_j(b_i\cdots b_{j-1})\bar{c}_{j+1,n-1}\right) + (b_i\cdots b_{n-1})k_{\text{in}}}{a_0 k_{\text{out}}\,\bar{c}_{\ell,n-1}}, \tag{29}
$$

$$
m_\rho = \frac{k_{\text{out}}\left(\sum_{j=i}^{n-1} z_j(b_i\cdots b_{j-1})\bar{c}_{j+1,n-1}\right) + (b_i\cdots b_{n-1})k_{\text{in}}}{(T - a_0)k_{\text{out}}\,\bar{c}_{\rho,n-1}}. \tag{30}
$$

Remember that $b_\ell, b_{\ell+1}, b_\rho, b_{\rho+1}$ depend on $a_0, a_1$, while the rest of $b$'s do not. For a product of the form $b_u\cdots b_j$ with $u \leq j$, we have the following:
If $u \leq \ell - 1$:

$$
b_u\cdots b_j = \begin{cases} \prod_{i=u}^{j}(k_{\text{in}} + F_{i+1})K_{i+1} & j < \ell - 1, \text{ or } \rho \leq j \\ \frac{1}{a_0}\prod_{i=u}^{j}(k_{\text{in}} + F_{i+1})K_{i+1} & j = \ell - 1, \text{ or } j = \rho - 1 \\ \frac{a_1}{a_0}\prod_{i=u}^{j}(k_{\text{in}} + F_{i+1})K_{i+1} & \ell - 1 < j < \rho - 1. \end{cases}
$$

If $u = \ell$:

$$b_u \cdots b_j = \begin{cases} a_1 \prod_{i=u}^{j}(k_{\text{in}} + F_{i+1})K_{i+1} & j < \rho - 1 \\ \prod_{i=u}^{j}(k_{\text{in}} + F_{i+1})K_{i+1} & j = \rho - 1 \\ a_0 \prod_{i=u}^{j}(k_{\text{in}} + F_{i+1})K_{i+1} & \rho \leq j. \end{cases}$$

If $\ell < u \leq \rho - 1$:

$$b_u \cdots b_j = \begin{cases} \prod_{i=u}^{j}(k_{\text{in}} + F_{i+1})K_{i+1} & j < \rho - 1 \\ \frac{1}{a_1} \prod_{i=u}^{j}(k_{\text{in}} + F_{i+1})K_{i+1} & j = \rho - 1 \\ \frac{a_0}{a_1} \prod_{i=u}^{j}(k_{\text{in}} + F_{i+1})K_{i+1} & \rho \leq j. \end{cases}$$

If $u = \rho$:

$$b_u \cdots b_j = a_0 \prod_{i=u}^{j}(k_{\text{in}} + F_{i+1})K_{i+1}.$$

If $u > \rho$:

$$b_u \cdots b_j = \prod_{i=u}^{j}(k_{\text{in}} + F_{i+1})K_{i+1}.$$

Summarising $b_u \cdots b_j$, equals

$$\begin{cases} \prod_{i=u}^{j}(k_{\text{in}} + F_{i+1})K_{i+1} & u \leq \ell - 1, j < \ell - 1, \text{ or } u \leq \ell - 1, \rho \leq j, \text{ or } u = \ell, j = \rho - 1, \\ & \text{or } \ell < u \leq \rho - 1, j < \rho - 1, \text{ or } u > \rho \\ \frac{1}{a_0} \prod_{i=u}^{j}(k_{\text{in}} + F_{i+1})K_{i+1} & j = \ell - 1, \text{ or } u \leq \ell - 1, j = \rho - 1 \\ \frac{a_1}{a_0} \prod_{i=u}^{j}(k_{\text{in}} + F_{i+1})K_{i+1} & u \leq \ell - 1, \ell - 1 < j < \rho - 1 \\ a_1 \prod_{i=u}^{j}(k_{\text{in}} + F_{i+1})K_{i+1} & u = \ell, j < \rho - 1 \\ a_0 \prod_{i=u}^{j}(k_{\text{in}} + F_{i+1})K_{i+1} & u = \ell, \rho \leq j, \text{ or } u = \rho \\ \frac{1}{a_1} \prod_{i=u}^{j}(k_{\text{in}} + F_{i+1})K_{i+1} & \ell < u \leq \rho - 1, j = \rho - 1 \\ \frac{a_0}{a_1} \prod_{i=u}^{j}(k_{\text{in}} + F_{i+1})K_{i+1} & \ell < u \leq \rho - 1, \rho \leq j. \end{cases}$$

Observe that for all $i$, the summand of $\sum_{j=i}^{n-1} z_j(b_i \cdots b_{j-1})(c_{j+1} \cdots c_{n-1})$ corresponding to $i = j$ is

$$z_i(c_{i+1} \cdots c_{n-1}),$$

which does not depend either on $a_0$ or $a_1$.

In particular, we deduce that $m_i$ for $i \leq \ell - 1$, the term $(b_i \cdots b_{n-1})$ does not depend on $a_0, a_1$, and in the sum $\sum_{j=i}^{n-1} z_j(b_i \cdots b_{j-1})(c_{j+1} \cdots c_{n-1})$ there are summands involving $\frac{1}{a_0}$, for eample when $j = \ell$. Hence $m_i$, for $i \leq \ell - 1$, goes to infinity as $a_0$ goes to zero. Note that $a_1$ goes to $T$ when $a_0$ goes to zero. When $i = \ell$, the denominator of $m_i$ is multiplied by $a_0$. As the numerator has at least one term that is not multiple of $a_0$, $m_\ell$ goes to infinity as well. We conclude that

$$m_i \xrightarrow[a_0 \to 0]{} \infty, \qquad 0 \leq i \leq \ell.$$

When $i \geq \ell$, then no summand in the numerator of $m_i$ in (27–29) that involves $\frac{1}{a_0}$, and neither the denominator has $a_0$ as factor. As the numerator has at least one term that is not multiple of $a_0$, $m_i$ goes to a finite value as $a_0$ goes to zero.

$$m_i \xrightarrow[a_0 \to 0]{} \text{number}, \qquad i \geq \ell.$$

The number can be found using (*Equations 28–30*) and is a function of the parameters of the system, not involving $W$.

We consider now the remaining equation, namely the conservation law $m_{\ell+1} + \cdots + m_\rho + a_0 = W$, to determine the value of $a_0$. We have

$$\sum_{i=\ell+1}^{\rho} m_i + a_0 - W \quad = \sum_{i=\ell+1}^{\rho-1} \frac{k_{\text{out}}\left(\sum_{j=i}^{n-1} z_j(b_i \cdots b_{j-1})\bar{c}_{j+1,n-1}\right) + (b_i \cdots b_{n-1})k_{\text{in}}}{k_{\text{out}}\,\bar{c}_{i,n-1}}$$

$$+ \frac{k_{\text{out}}\left(\sum_{j=\rho}^{n-1} z_j(b_\rho \cdots b_{j-1})\bar{c}_{j+1,n-1}\right) + (b_\rho \cdots b_{n-1})k_{\text{in}}}{a_1 k_{\text{out}}\,\bar{c}_{\rho,n-1}} + a_0 - W = (*).$$

We have $(b_i \cdots b_{n-1}) = \frac{a_0}{a_1}\bar{b}_{i,n-1}$ when $\ell < i < \rho$ and $(b_\rho \cdots b_{n-1}) = a_0 \bar{b}_{\rho,n-1}$. Also

$$\sum_{j=\rho}^{n-1} z_j(b_\rho \cdots b_{j-1})\bar{c}_{j+1,n-1} = z_\rho \bar{c}_{\rho+1,n-1} + \sum_{j=\rho+1}^{n-1} a_0 z_j \bar{b}_{\rho,j-1}\bar{c}_{j+1,n-1}.$$

Finally, for $\ell + 1 \le i \le \rho - 1$,

$$\sum_{j=i}^{n-1} z_j(b_i \cdots b_{j-1})\bar{c}_{j+1,n-1} = \sum_{j=i}^{\rho-1} z_j(b_i \cdots b_{j-1})\bar{c}_{j+1,n-1} + z_\rho(b_i \cdots b_{\rho-1})\bar{c}_{\rho+1,n-1} + \sum_{j=\rho+1}^{n-1} z_j(b_i \cdots b_{j-1})\bar{c}_{j+1,n-1}$$

$$= \sum_{j=i}^{\rho-1} z_j \bar{b}_{i,j-1}\bar{c}_{j+1,n-1} + z_\rho \frac{1}{a_1}\bar{b}_{i,\rho-1}\bar{c}_{\rho+1,n-1} + \frac{a_0}{a_1}\sum_{j=\rho+1}^{n-1} z_j \bar{b}_{i,j-1}\bar{c}_{j+1,n-1}.$$

This gives

$$(*) \quad = \sum_{i=\ell+1}^{\rho-1} \frac{k_{\text{out}}\left(\sum_{j=i}^{\rho-1} z_j \bar{b}_{i,j-1}\bar{c}_{j+1,n-1} + z_\rho \frac{1}{a_1}\bar{b}_{i,\rho-1}\bar{c}_{\rho+1,n-1} + \frac{a_0}{a_1}\sum_{j=\rho+1}^{n-1} z_j \bar{b}_{i,j-1}\bar{c}_{j+1,n-1}\right) + \frac{a_0}{a_1}\bar{b}_{i,n-1}k_{\text{in}}}{k_{\text{out}}\,\bar{c}_{i,n-1}}$$

$$+ \frac{k_{\text{out}}\left(z_\rho \bar{c}_{\rho+1,n-1} + \sum_{j=\rho+1}^{n-1} a_0 z_j \bar{b}_{\rho,j-1}\bar{c}_{j+1,n-1}\right) + a_0 \bar{b}_{\rho,n-1}k_{\text{in}}}{a_1 k_{\text{out}}\,\bar{c}_{\rho,n-1}} + a_0 - W$$

$$= \sum_{i=\ell+1}^{\rho-1} \frac{k_{\text{out}}\left(a_1 \sum_{j=i}^{\rho-1} z_j \bar{b}_{i,j-1}\bar{c}_{j+1,n-1} + z_\rho \bar{b}_{i,\rho-1}\bar{c}_{\rho+1,n-1} + a_0 \sum_{j=\rho+1}^{n-1} z_j \bar{b}_{i,j-1}\bar{c}_{j+1,n-1}\right) + a_0 \bar{b}_{i,n-1}k_{\text{in}}}{a_1 k_{\text{out}}\,\bar{c}_{i,n-1}}$$

$$+ \frac{k_{\text{out}}\left(z_\rho \bar{c}_{\rho+1,n-1} + \sum_{j=\rho+1}^{n-1} a_0 z_j \bar{b}_{\rho,j-1}\bar{c}_{j+1,n-1}\right) + a_0 \bar{b}_{\rho,n-1}k_{\text{in}}}{a_1 k_{\text{out}}\,\bar{c}_{\rho,n-1}} + a_0 - W.$$

Hence $(*)$ vanishes if and only if

$$0 \quad = \sum_{i=\ell+1}^{\rho-1} k_{\text{out}}\left(0, 0, 1 a_1 \sum_{j=i}^{\rho-1} z_j \bar{b}_{i,j-1}\bar{c}_{\ell+1,i-1}\bar{c}_{j+1,n-1} + z_\rho \bar{b}_{i,\rho-1}\bar{c}_{\ell+1,i-1}\bar{c}_{\rho+1,n-1} + 0, 0, 1 a_0 \sum_{j=\rho+1}^{n-1} z_j \bar{b}_{i,j-1}\bar{c}\right.$$

$$+ 0, 0, 1 a_0 \bar{b}_{i,n-1}\bar{c}_{\ell+1,i-1}k_{\text{in}} + k_{\text{out}}\left(z_\rho \bar{c}_{\ell+1,\rho-1}\bar{c}_{\rho+1,n-1} + \sum_{j=\rho+1}^{n-1} 0, 0, 1 a_0 z_j \bar{b}_{\rho,j-1}\bar{c}_{\ell+1,\rho-1}\bar{c}_{j+1,n-1}\right)$$

$$+ 0, 0, 1 a_0 \bar{b}_{\rho,n-1}\bar{c}_{\ell+1,\rho-1}k_{\text{in}} + (0, 0, 1 a_0 - W)k_{\text{out}}\,\bar{c}_{\ell+1,n-1} 0, 0, 1 a_1.$$

$$\tag{31}$$

As $a_1 = A_{\text{tot}} - a_0$, this is a degree 2 polynomial in $a_0$. The leading term comes from the term $(0, 0, 1a_0 - W)k_{\text{out}}\bar{c}_{\ell+1,n-1}\,0, 0, 1a_1$, and is

$$-k_{\text{out}}\bar{c}_{\ell+1,n-1}a_0^2,$$

which is negative. The independent term is

$$\sum_{i=\ell+1}^{\rho-1} k_{\text{out}}\left(A_{\text{tot}}\sum_{j=i}^{\rho-1} z_j\bar{b}_{i,j-1}\bar{c}_{\ell+1,i-1}\bar{c}_{j+1,n-1} + z_\rho\bar{b}_{i,\rho-1}\bar{c}_{\ell+1,i-1}\bar{c}_{\rho+1,n-1}\right)$$
$$+k_{\text{out}}z_\rho\bar{c}_{\ell+1,\rho-1}\bar{c}_{\rho+1,n-1} - A_{\text{tot}}Wk_{\text{out}}\bar{c}_{\ell+1,n-1}.$$

We divide by $k_{\text{out}}$ and define

$$\alpha = \sum_{i=\ell+1}^{\rho-1}\bar{c}_{\ell+1,i-1}\left(A_{\text{tot}}\sum_{j=i}^{\rho-1}z_j\bar{b}_{i,j-1}\bar{c}_{j+1,n-1} + z_\rho\bar{b}_{i,\rho-1}\bar{c}_{\rho+1,n-1}\right) + z_\rho\bar{c}_{\ell+1,\rho-1}\bar{c}_{\rho+1,n-1} - A_{\text{tot}}W\bar{c}_{\ell+1,n-1},$$

where recall from (**25**) that

$$z_i = k_{\text{in}}K_{i+1}L_i \qquad b_i = \epsilon_{i+1}(k_{\text{in}} + F_{i+1})K_{i+1} \qquad c_i = (E_{i+1} - k_{\text{in}})L_i.$$

When $a_0 = A_{\text{tot}}$, all terms multiplying $a_1$ vanish, and then the polynomial is a sum of positive terms, hence positive. By Lemma 1, the system has a positive steady state if and only if

$$\alpha < 0.$$

Note that when $k_{\text{in}} = 0$ this inequality holds, as all terms with $z_j$ vanish. When $k_{\text{in}}$ approaches one of $E_i$ with $\ell + 1 < i \leq n$, the negative term of $\alpha$ approaches zero, and the inequality does not hold.

For example, for $n = 3$, $\ell = 0$ and $\rho = 2$, $\alpha$ was found in (**18**). For $n = 4$, $\ell = 0$ and $\rho = 2$, we have

$$\begin{aligned}\alpha &= -A_{\text{tot}}W(E_2 - k_{\text{in}})L_1(E_3 - k_{\text{in}})L_2(E_4 - k_{\text{in}}) + k_{\text{in}}K_4(E_2 - k_{\text{in}})L_1(E_3 - k_{\text{in}})L_2\\&+A_{\text{tot}}k_{\text{in}}k_{\text{out}}L_1(E_4 - k_{\text{in}})(E_3 - k_{\text{in}})L_2 + A_{\text{tot}}k_{\text{in}}K_3L_2(E_4 - k_{\text{in}})(F_2 + k_{\text{in}})k_{\text{out}}\\&+k_{\text{in}}K_4(F_3 + k_{\text{in}})K_3(F_2 + k_{\text{in}})k_{\text{out}} + k_{\text{in}}K_3L_2(E_2 - k_{\text{in}})L_1(E_4 - k_{\text{in}})A_{\text{tot}}\\&+k_{\text{in}}K_4(E_2 - k_{\text{in}})L_1(F_3 + k_{\text{in}})K_3.\end{aligned}$$

Let $B$ be the smallest positive root of $\alpha = 0$ as a polynomial in $k_{\text{in}}$, if it exists, or take $B = \infty$ if not. Similarly, let $B'$ be the second such root, if it exists, or $B' = \infty$ if not. If the smallest of $E_j$ is attained for some $E_i$ with $\ell + 1 < i \leq n$, then $\alpha$ is positive at $k_{\text{in}} = E_i$. In that case, as $\alpha$ is negative at $k_{\text{in}} = 0$, there is at least one value of $k_{\text{in}} < E_i$ for all $i$ such that $\alpha = 0$ and hence $B$ is finite. This shows that $\min(E_1, \ldots, E_n, B) = \min(E_1, \ldots, E_{\ell+1}, B)$.

Putting it all together, we have shown that for all

$$k_{\text{in}} < \min(E_1, \ldots, E_{\ell+1}, B) \tag{32}$$

the system has positive steady states, and if

$$k_{\text{in}} \geq \min(E_1, \ldots, E_{\ell+1}, B), \quad \text{or} \quad k_{\text{in}} < \min(E_1, \ldots, E_{\ell+1}), \quad B < k_{\text{in}} < B',$$

Taking condition (**32**), if the minimum is attained at $B$, then when $k_{\text{in}}$ approaches $B$, the first positive root of the polynomial in (**3031**, ) approaches zero (as $\alpha$ goes to zero). As this determines the steady state value of $a_0$, we see that $a_0$ approaches zero, and the $m_i$ converge to the values given above. Specifically,

$$m_j \xrightarrow[k_{\text{in}} \to B]{} \infty, \qquad \text{for } 0 \leq j \leq \ell, \qquad m_j \xrightarrow[k_{\text{in}} \to B]{} \text{number}, \qquad \text{for } j \geq \ell.$$

By the comment above, (**32**) cannot be attained at $E_i$ with $\ell + 1 < i \leq n$. If (**32**) is attained at $E_i$ with $i \leq \ell + 1$, then as $k_{\text{in}}$ approaches this minimum, $a_0$ converges to a number. In this case, the concentrations that tend to infinity are those with $E_i - k_{\text{in}}$ in the denominator:

$$m_j \xrightarrow[k_{\text{in}} \to E_i]{} \infty, \qquad \text{for } 0 \leq j < i, \qquad m_j \xrightarrow[k_{\text{in}} \to E_i]{} \text{number}, \qquad \text{for } j \geq i.$$

## Appendix 5

### Multiple co-substrate cycling along a single pathway – mimicking the case seen in glycolysis, combined with fermentation

In this section, we consider a scenario of intra-pathway cycling with two different co-substrates. This is a simplified version of the case seen in the combined pathways of glycolysis and fermentation. The reaction scheme we consider comprises:

$$0 \xrightarrow{k_{\text{in}}} M_0 \qquad M_0 + \text{ATP} \rightleftharpoons M_1 + \text{ADP} \qquad M_1 + \text{NAD} \rightleftharpoons M_2 + \text{NADH}$$
$$M_4 \xrightarrow{k_{\text{out}}} 0 \qquad M_2 + \text{ADP} \rightleftharpoons M_3 + \text{ATP} \qquad M_3 + \text{NADH} \rightleftharpoons M_4 + \text{NAD}$$

(33)

We write $A_0 = \text{ATP}$, $A_1 = \text{ADP}$, $A_2 = \text{NAD}$, $A_3 = \text{NADH}$, for simplicity. The ODE system governing the dynamics of the network is:

$$\frac{dm_0}{dt} = k_{\text{in}} - \frac{E_1 L_0 a_0 m_0 - F_1 K_1 a_1 m_1}{K_1 a_1 m_1 + L_0 a_0 m_0 + K_1 L_0}$$

$$\frac{dm_1}{dt} = \frac{E_1 L_0 a_0 m_0 - F_1 K_1 a_1 m_1}{K_1 a_1 m_1 + L_0 a_0 m_0 + K_1 L_0} - \frac{E_2 L_1 a_2 m_1 - F_2 K_2 a_3 m_2}{K_2 a_3 m_2 + L_1 a_2 m_1 + K_2 L_1}$$

$$\frac{dm_2}{dt} = \frac{E_2 L_1 a_2 m_1 - F_2 K_2 a_3 m_2}{K_2 a_3 m_2 + L_1 a_2 m_1 + K_2 L_1} - \frac{E_3 L_2 a_1 m_2 - F_3 K_3 a_0 m_3}{K_3 a_0 m_3 + L_2 a_1 m_2 + K_3 L_2}$$

$$\frac{dm_3}{dt} = \frac{E_3 L_2 a_1 m_2 - F_3 K_3 a_0 m_3}{K_3 a_0 m_3 + L_2 a_1 m_2 + K_3 L_2} - \frac{E_4 L_3 a_3 m_3 - F_4 K_4 a_2 m_4}{K_4 a_2 m_4 + L_3 a_3 m_3 + K_4 L_3}$$

$$\frac{dm_4}{dt} = \frac{E_4 L_3 a_3 m_3 - F_4 K_4 a_2 m_4}{K_4 a_2 m_4 + L_3 a_3 m_3 + K_4 L_3} - k_{\text{out}} m_4$$

$$\frac{da_0}{dt} = -\frac{E_1 L_0 a_0 m_0 - F_1 K_1 a_1 m_1}{K_1 a_1 m_1 + L_0 a_0 m_0 + K_1 L_0} - \frac{E_3 L_2 a_1 m_2 - F_3 K_3 a_0 m_3}{K_3 a_0 m_3 + L_2 a_1 m_2 + K_3 L_2}$$

$$\frac{da_1}{dt} = \frac{E_1 L_0 a_0 m_0 - F_1 K_1 a_1 m_1}{K_1 a_1 m_1 + L_0 a_0 m_0 + K_1 L_0} - \frac{E_3 L_2 a_1 m_2 - F_3 K_3 a_0 m_3}{K_3 a_0 m_3 + L_2 a_1 m_2 + K_3 L_2}$$

$$\frac{da_2}{dt} = -\frac{E_2 L_1 a_2 m_1 - F_2 K_2 a_3 m_2}{K_2 a_3 m_2 + L_1 a_2 m_1 + K_2 L_1} + \frac{E_4 L_3 a_3 m_3 - F_4 K_4 a_2 m_4}{K_4 a_2 m_4 + L_3 a_3 m_3 + K_4 L_3}$$

$$\frac{da_3}{dt} = \frac{E_2 L_1 a_2 m_1 - F_2 K_2 a_3 m_2}{K_2 a_3 m_2 + L_1 a_2 m_1 + K_2 L_1} - \frac{E_4 L_3 a_3 m_3 - F_4 K_4 a_2 m_4}{K_4 a_2 m_4 + L_3 a_3 m_3 + K_4 L_3}.$$

There are four conservation laws:

$$a_0 + a_1 = A_{\text{tot}}, \qquad a_2 + a_3 = B_{\text{tot}}, \qquad m_1 + m_2 + a_0 = W, \qquad m_2 + m_3 + a_2 = M.$$

We consider the equations $\frac{dm_0}{dt} + \cdots + \frac{dm_4}{dt} = 0$, $\frac{dm_4}{dt} = 0$, $\frac{dm_3}{dt} + \frac{dm_4}{dt} = 0$, $\frac{dm_2}{dt} + \frac{dm_3}{dt} + \frac{dm_4}{dt} = 0$, $\frac{dm_1}{dt} + \frac{dm_2}{dt} + \frac{dm_3}{dt} + \frac{dm_4}{dt} = 0$, and solve them iteratively for $m_4, m_3, m_2, m_1, m_0$ and obtain:

$$m_4 = \frac{k_{\text{in}}}{k_{\text{out}}}, \qquad\qquad m_3 = \frac{k_{\text{in}} K_4 \left( (F_4 + k_{\text{in}}) a_2 + L_3 k_{\text{out}} \right)}{k_{\text{out}} a_3 L_3 (E_4 - k_{\text{in}})},$$

$$m_2 = \frac{K_3 \left( (F_3 + k_{\text{in}}) a_0 m_3 + L_2 k_{\text{in}} \right)}{a_1 L_2 (E_3 - k_{\text{in}})}, \qquad m_1 = \frac{K_2 \left( (F_2 + k_{\text{in}}) a_3 m_2 + L_1 k_{\text{in}} \right)}{a_2 L_1 (E_2 - k_{\text{in}})},$$

$$m_0 = \frac{K_1 \left( (F_1 + k_{\text{in}}) a_1 m_1 + L_0 k_{\text{in}} \right)}{a_0 L_0 (E_1 - k_{\text{in}})},$$

and $a_1 = A_{\text{tot}} - a_0$, $a_3 = B_{\text{tot}} - a_2$. As usual, a necessary condition for positive steady states is

$$k_{in} < \min(E_1, E_2, E_3, E_4). \tag{34}$$

For this system, we are left with two conservation laws that we have not used, and two steady state values that are still free. By plugging the expressions above into $m_1 + m_2 + a_0 = W$, and $m_2 + m_3 + a_2 = M$, we obtain a system of two polynomial equations in two variables. For some combinations of parameter values, there will be positive steady states, and for others, not, as the following examples shows.

By choosing

$$M = 1, \quad W = 1, \quad F_1 = 1 \quad F_2 = 1, \quad F_3 = 1, \quad F_4 = 2, \quad K_1 = 1 \quad K_2 = 0.1, \quad K_3 = 0.1,$$

$$K_4 = 2, \quad L_0 = 1 \quad L_1 = 1 \quad L_2 = 2, \quad L_3 = 10, \quad A_{tot} = 3, \quad B_{tot} = 2, \quad E_1 = 3, \quad E_2 = 3,$$

$$E_3 = 12, \quad E_4 = 12, \quad k_{in} = 2, \quad k_{out} = 10,$$

the polynomial system becomes

$$0 = 20000 a_2^2 a_0 - 60000 a_2^2 + 180560 a_2 - 60315.2 a_2 a_0 - 95200 + 32120 a_0$$

$$0 = 20000 a_2^2 a_0^2 - 40000 a_2 a_0^2 - 79996.64 a_2^2 a_0 + 164086.88 a_2 a_0 - 7928 a_0 + 59720 a_2^2 - 131680 a_2 + 24480$$

and the solutions are:

$$a_0 \sim 0.69, \qquad a_2 \sim 0.679,$$

which are in the desired interval.

If instead we replace $L_3$ by 1, we obtain a system whose solutions are

$$a_0 \sim 2.978, \qquad a_2 \sim -0.323,$$

and hence there are no positive steady states.

This shows that there is a condition for positive steady states to exist, and although (*34*) is necessary, it is not sufficient.

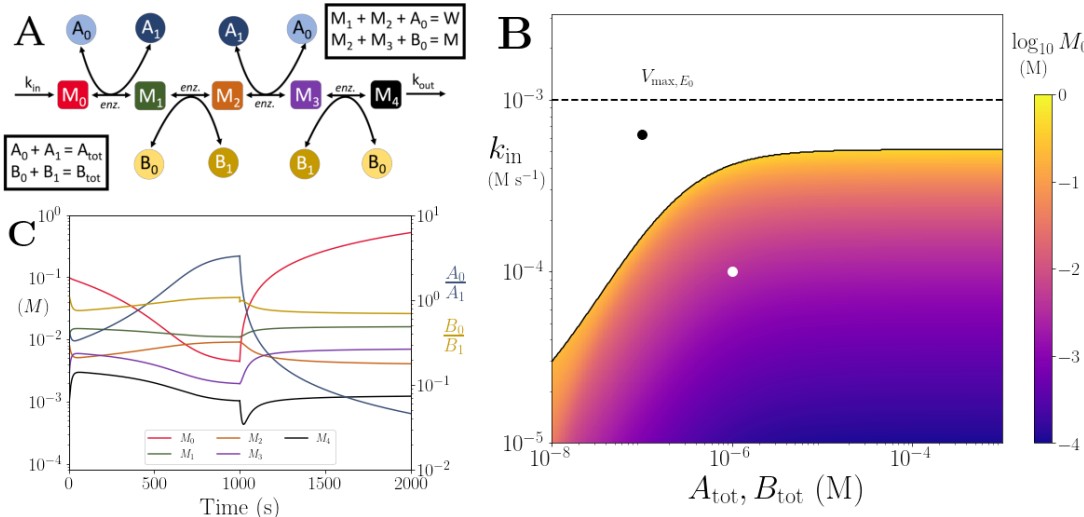

**Appendix 5—figure 1.** Motif, time series and threshold for the linear pathway model with multiple co-substrates. (**A**) Cartoon representation of a chain of reversible reactions with metabolite cycling of two different metabolites, as shown in *Equation 33*. Each cycled metabolite has two forms, and there is no pathway independent cycling. (**B**) Heatmap of of the steady state concentration of $M_0$ as a function of the metabolite pool sizes ($A_{tot} = B_{tot}$) and the inflow flux ($k_{in}$). White area shows the region where there is no steady state. (**C**) Concentrations of $M_{0-4}$, $A_0/A_1$ and $B_0/B_1$ ratios as a function of time. At $t = 1000$ s, parameters are switched from the white dot in panel (**B**) (where a steady state exists) to the black dot (where we see build up of $M_0$), and continued decline of $A_0$. The $B_0/B_1$ ratio remains constant however as these are still cycled by reactions after the build up. In (**B**) and (**C**) the other parameters are $k_{cat} = 100 s^{-1}$, $E_{tot} = 0.01 mM$, $K_m = 50 \mu M$ and $k_{out} = 0.1 s^{-1}$.

## Appendix 6

### Different stoichiometries for co-substrate cycling along a single pathway – mimicking the case seen in upper glycolysis

In this section, we consider a scenario of intra-pathway cycling with varying stoichiometry in the pathway. In particular, we consider a simplified version of the case seen in upper glycolysis. The reaction scheme we consider comprises:

$$0 \xrightarrow{k_{\text{in}}} M_0 \quad M_0 + \text{ATP} \rightleftharpoons M_1 + \text{ADP} \quad M_1 + \text{ATP} \rightleftharpoons 2M_2 + \text{ADP}$$
$$M_2 + \text{ADP} \rightleftharpoons M_3 + \text{ATP} \qquad M_3 \xrightarrow{k_{\text{out}}} 0. \tag{35}$$

We let $A_0 = \text{ATP}$ and $A_1 = \text{ADP}$ as usual.

With Michaelis-Menten kinetics, the ODE system is

$$\frac{dm_0}{dt} = k_{\text{in}} - \frac{E_1 L_0 m_0 a_0 - F_1 K_1 m_1 a_1}{K_1 m_1 a_1 + L_0 m_0 a_0 + K_1 L_0}$$

$$\frac{dm_1}{dt} = \frac{E_1 L_0 m_0 a_0 - F_1 K_1 m_1 a_1}{K_1 m_1 a_1 + L_0 m_0 a_0 + K_1 L_0} - \frac{E_2 L_1 m_1 a_0 - F_2 K_2 m_2^2 a_1}{K_2 m_2^2 a_1 + L_1 m_1 a_0 + K_2 L_1}$$

$$\frac{dm_2}{dt} = \frac{2(E_2 L_1 m_1 a_0 - F_2 K_2 m_2^2 a_1)}{K_2 m_2^2 a_1 + L_1 m_1 a_0 + K_2 L_1} - \frac{E_3 L_2 m_2 a_1 - F_3 K_3 m_3 a_0}{K_3 m_3 a_0 + L_2 m_2 a_1 + K_3 L_2}$$

$$\frac{dm_3}{dt} = \frac{E_3 L_2 m_2 a_1 - F_3 K_3 m_3 a_0}{K_3 m_3 a_0 + L_2 m_2 a_1 + K_3 L_2} - k_{\text{out}} m_3$$

$$\frac{da_0}{dt} = -\frac{E_1 L_0 m_0 a_0 - F_1 K_1 m_1 a_1}{K_1 m_1 a_1 + L_0 m_0 a_0 + K_1 L_0} - \frac{E_2 L_1 m_1 a_0 - F_2 K_2 m_2^2 a_1}{K_2 m_2^2 a_1 + L_1 m_1 a_0 + K_2 L_1} + \frac{E_3 L_2 m_2 a_1 - F_3 K_3 m_3 a_0}{K_3 m_3 a_0 + L_2 m_2 a_1 + K_3 L_2}$$

$$\frac{da_1}{dt} = \frac{E_1 L_0 m_0 a_0 - F_1 K_1 m_1 a_1}{K_1 m_1 a_1 + L_0 m_0 a_0 + K_1 L_0} + \frac{E_2 L_1 m_1 a_0 - F_2 K_2 m_2^2 a_1}{K_2 m_2^2 a_1 + L_1 m_1 a_0 + K_2 L_1} - \frac{E_3 L_2 m_2 a_1 - F_3 K_3 m_3 a_0}{K_3 m_3 a_0 + L_2 m_2 a_1 + K_3 L_2}.$$

The system has two conservation laws:

$$a_0 + a_1 = A_{\text{tot}}, \qquad m_1 + m_2 + a_0 = W.$$

By considering the equation $0 = 2\frac{dm_0}{dt} + 2\frac{dm_1}{dt} + \frac{dm_2}{dt} + \frac{dm_3}{dt} = k_{\text{out}} m_3 - 2k_{\text{in}}$, we obtain

$$m_3 = \frac{2k_{\text{in}}}{k_{\text{out}}}.$$

Upon substitution of this value of $m_3$ into $\frac{dm_3}{dt} = 0$, $\frac{dm_2}{dt} + \frac{dm_3}{dt} = 0$ and $2\frac{dm_1}{dt} + \frac{dm_2}{dt} + \frac{dm_3}{dt} = 0$, and solving recursively for $M_2, m_1, m_0$, we obtain the following steady state relations:

$$m_2 = \frac{2k_{\text{in}} K_3 \left((F_3 + 2k_{\text{in}})a_0 + L_2 k_{\text{out}}\right)}{k_{\text{out}} a_1 L_2 \left(E_3 - 2k_{\text{in}}\right)},$$

$$m_1 = -\frac{K_2 \left((F_2 + k_{\text{in}})a_1 m_2^2 + L_1 k_{\text{in}}\right)}{L_1 a_0 \left(E_2 - k_{\text{in}}\right)},$$

$$m_0 = -\frac{K_1 \left((F_1 + k_{\text{in}})a_1 m_1 + L_0 k_{\text{in}}\right)}{a_0 L_0 \left(E_1 - k_{\text{in}}\right)},$$

and recall $a_1 = A_{\text{tot}} - a_0$. We see that if $0 < a_0 < A_{\text{tot}}$, these expressions are all positive if and only if

$$k_{\text{in}} < \min(E_1, E_2, E_3/2). \tag{36}$$

As usual, we consider the remaining conservation law, $m_1 + m_2 + a_0 = W$, together with these expressions, to find a polynomial $p(a_0)$ whose roots in $(0, A_{tot})$ are in one-to-one correspondence with positive steady states, provided (**36**) holds. The polynomial has degree 3, positive leading term, negative independent term and term of degree 2, and is positive at $a_0 = A_{tot}$. With this information, we cannot immediately conclude that there is a positive root in $(0, A_{tot})$. But for positive steady states to exist, we need the term of degree 1 to be positive (this follows from Descartes' rule of signs, as usual), and this sets an extra condition on the parameters. When this happens, there will be two positive steady states.

Specifically, the term of degree 1 is

$$\alpha = L_2 k_{out} \Big( L_1 L_2 \left( E_3 - 2k_{in} \right)^2 k_{out} \left( A_{tot} W (E_2 - k_{in}) + K_2 k_{in} \right)$$
$$- 2K_3 L_1 L_2 \left( E_2 - k_{in} \right) \left( E_3 - 2k_{in} \right) k_{in} k_{out}$$
$$- 8K_2 K_3^2 k_{in}^2 \left( F_3 + 2k_{in} \right) \left( F_2 + k_{in} \right) \Big),$$

which depends on $A_{tot}$ and $W$. To summarize for positive steady states to exist, we need

$$k_{in} < E_1, \qquad k_{in} < E_2, \qquad k_{in} < \tfrac{E_3}{2}, \qquad \alpha > 0.$$

If $k_{in}$ is small enough, then the conditions hold as $\alpha > 0$ at $k_{in} = 0$.

However, these conditions are not sufficient. To see this, by inspecting the term $\alpha$, one can see that if $K_2, K_3$ are small enough, then there will be two positive steady states, while if they are larger, there will be none. We have verified that both scenarios occur. For example, fix the parameter values to be

$$A_{tot} = 1, \quad W = 2, \quad F_1 = 1, \quad F_2 = 2, \quad F_3 = 3, \quad L_0 = 1 \quad L_1 = 1,$$
$$L_2 = 2, \quad K_1 = 1 \quad k_{in} = 1, \quad k_{out} = 1, \quad E_1 = 2, \quad E_2 = 2, \quad E_3 = 3,$$

and note that (35) holds. With $K_2 = K_3 = 0.1$, the polynomial of interest becomes $p(a_0) = 4a_0^3 - 14.300a_0^2 + 7.360a_0 - 0.448$, and so $\alpha > 0$. The polynomial has two positive roots under $A_{tot} = 1$, namely 0.07 and 0.537. For $K_2 = 0.2, K_3 = 0.3$, the polynomial is $p(a_0) = 4a_0^3 - 23.400a_0^2 + 2.080a_0 - 1.664$, and although $\alpha > 0$, it has no root in the interval $(0, 1)$.

An analogous reasoning holds for the irreversible system.

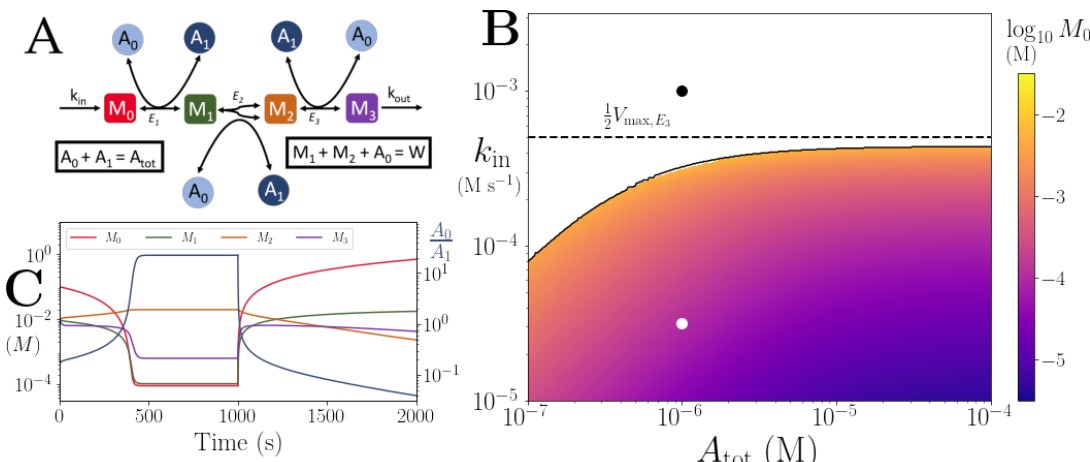

**Appendix 6—figure 1.** Motif, time series and threshold for a model with differential co-substrate stoichiometry, as seen in glycolysis. (**A**) Reaction system modelled in Appendix 6. Branching arrowhead from $M_1$ to $M_2$ indicates that two $M_2$ molecules are produced/used in the forward/backward reactions. (**B**) Heatmap of the steady state $M_0$ concentration. In the white area there is no steady state due to continual build up of $M_0$. Dashed line shows the smallest limit imposed by **Equation 35**. (**C**) Time series of $M_{0\to4}$ and $A_0/A_1$. At $t = 1000$ s, parameters are switched from the white dot in panel (**B**) (where a steady state exists) to the black dot (where we see build up of $M_0$). Note that in the build up regime, $M_2$ reduces as well as $A_0/A_1$, as $M_2$ production is dependent on the presence of $A_0$. In (**B**) and (**C**) the other parameters are $k_{cat} = 100s^{-1}$, $E_{tot} = 0.01mM$, $K_m = 50\mu M$ and $k_{out} = 0.1s^{-1}$.

## Appendix 7

### Co-substrate cycling along with metabolite cycling – mimicking the case seen in nitrogen assimilation

In this section, we consider a scenario of intertwined, co-substrate cycling within a pathway. In particular, we consider a simplified version of the case seen in nitrogen assimilation where NAD(P)H cycling co-occurs together with a cycling from alpha-ketoglutarate to glutamate (mediated by glutamate dehydrogenase) and back from glutamate to alpha-ketoglutarate and glutamine (mediated by glutamate synthase). The simplified reaction scheme we consider comprises:

$$0 \xrightarrow{k_{in}} NH3 \qquad M_0 + NADH + NH3 \rightleftharpoons M_1 + NAD$$
$$M_2 \xrightarrow{k_{out}} 0 \qquad M_1 + NAD \rightleftharpoons M_0 + M_2 + NADH. \tag{37}$$

Denoting $A_0 = $ NADH, $A_1 = $ NAD, $A_2 = $ NH3, the ODE system is:

$$\frac{dm_0}{dt} = -\frac{E_1 L_0 a_0 a_2 m_0 - F_1 K_1 a_1 m_1}{L_0 a_0 a_2 m_0 + K_1 a_1 m_1 + K_1 L_0} + \frac{E_2 L_1 a_1 m_1 - F_2 K_2 a_0 m_0 m_2}{K_2 a_0 m_0 m_2 + L_1 a_1 m_1 + K_2 L_1}$$

$$\frac{dm_1}{dt} = \frac{E_1 L_0 a_0 a_2 m_0 - F_1 K_1 a_1 m_1}{L_0 a_0 a_2 m_0 + K_1 a_1 m_1 + K_1 L_0} - \frac{-F_2 K_2 a_0 m_0 m_2 + E_2 L_1 a_1 m_1}{K_2 a_0 m_0 m_2 + L_1 a_1 m_1 + K_2 L_1}$$

$$\frac{dm_2}{dt} = \frac{-F_2 K_2 a_0 m_0 m_2 + E_2 L_1 a_1 m_1}{K_2 a_0 m_0 m_2 + L_1 a_1 m_1 + K_2 L_1} - k_{out} m_2$$

$$\frac{da_0}{dt} = -\frac{E_1 L_0 a_0 a_2 m_0 - F_1 K_1 a_1 m_1}{L_0 a_0 a_2 m_0 + K_1 a_1 m_1 + K_1 L_0} + \frac{-F_2 K_2 a_0 m_0 m_2 + E_2 L_1 a_1 m_1}{K_2 a_0 m_0 m_2 + L_1 a_1 m_1 + K_2 L_1}$$

$$\frac{da_1}{dt} = \frac{E_1 L_0 a_0 a_2 m_0 - F_1 K_1 a_1 m_1}{L_0 a_0 a_2 m_0 + K_1 a_1 m_1 + K_1 L_0} - \frac{-F_2 K_2 a_0 m_0 m_2 + E_2 L_1 a_1 m_1}{K_2 a_0 m_0 m_2 + L_1 a_1 m_1 + K_2 L_1}$$

$$\frac{da_2}{dt} = k_{in} - \frac{E_1 L_0 a_0 a_2 m_0 - F_1 K_1 a_1 m_1}{L_0 a_0 a_2 m_0 + K_1 a_1 m_1 + K_1 L_0}.$$

There are three conservation laws:

$$a_0 + m_1 = M, \qquad a_0 + a_1 = A_{tot}, \qquad m_0 + a_1 = W.$$

From $\frac{da_2}{dt} + \frac{dm_1}{dt} + \frac{dm_2}{dt} = 0$ we get

$$m_2 = \frac{k_{in}}{k_{out}}$$

as expected. From $\frac{dm_2}{dt} = 0$ and $\frac{da_2}{dt} = 0$ we get

$$m_1 = \frac{k_{in} K_2 \left( F_2 a_0 m_0 + k_{in} a_0 m_0 + L_1 k_{out} \right)}{k_{out} a_1 L_1 \left( E_2 - k_{in} \right)}, \qquad a_2 = \frac{K_1 \left( F_1 a_1 m_1 + a_1 k_{in} m_1 + L_0 k_{in} \right)}{m_0 a_0 L_0 \left( E_1 - k_{in} \right)}.$$

From the second and third conservation laws, we get

$$a_1 = A_{tot} - a_0, \qquad m_0 = W - a_1 = W - A_{tot} + a_0.$$

This gives that for a positive steady state, we need $k_{in} < E_1, k_{in} < E_2$ and $\max(A_{tot} - W, 0) < a_0 < A_{tot}$. Note that $A_{tot} - W$ is the constant value of $a_0 - m_0$ along trajectories.

Plugging these expressions into the first conservation law, we have that steady states are in one-to-one correspondence with the solutions to

$$M = a_0 + \frac{k_{in} K_2 \left( F_2 a_0 \left( a_0 - A_{tot} + W \right) + a_0 k_{in} \left( a_0 - A_{tot} + W \right) + L_1 k_{out} \right)}{k_{out} \left( -a_0 + A_{tot} \right) L_1 (E_2 - k_{in})}. \tag{38}$$

We first note that the derivative of the right hand side of (38) with respect to $a_0$ is:

$$1 + \frac{k_\text{in} K_2 \left( F_2 \left( a_0 - A_\text{tot} + W \right) + F_2 a_0 + k_\text{in} \left( a_0 - A_\text{tot} + W \right) + a_0 k_\text{in} \right)}{k_\text{out} \left( -a_0 + A_\text{tot} \right) L_1 (E_2 - k_\text{in})}$$

$$+ \frac{k_\text{in} K_2 \left( F_2 a_0 \left( a_0 - A_\text{tot} + W \right) + a_0 k_\text{in} \left( a_0 - A_\text{tot} + W \right) + L_1 k_\text{out} \right)}{k_\text{out} \left( -a_0 + A_\text{tot} \right)^2 L_1 (E_2 - k_\text{in})}$$

As $A_\text{tot} - a_0 > 0$ and $a_0 - A_\text{tot} - W > 0$, this function is positive in the interval of interest. Therefore, the right hand side of (38) is an increasing function of $a_0$ when $\max(A_\text{tot} - W, 0) < a_0 < A_\text{tot}$. It follows that if (38) has a solution, it has exactly one.

Rewriting (38) as a polynomial equation, steady states are in one-to-one correspondence with the roots in the interval $\max(A_\text{tot} - W, 0) < a_0 < A_\text{tot}$ of the following polynomial

$$p(a_0) = \left( -L_1(E_2 - k_\text{in}) k_\text{out} + K_2 k_\text{in}(F_2 + k_\text{in}) \right) a_0^2 + (L_1(E_2 - k_\text{in}) k_\text{out}(M + T)$$
$$+ (W - A_\text{tot}) K_2 k_\text{in}(F_2 + k_\text{in})) a_0 - L_1 k_\text{out} \left( M A_\text{tot}(E_2 - k_\text{in}) - K_2 k_\text{in} \right).$$

When $a_0 = A_\text{tot}$, the polynomial is positive.

Case 1: $A_\text{tot} - W \leq 0$. In this case we want $0 < a_0 < A_\text{tot}$. If the independent term of $p$ is negative, then $p$ has exactly one solution in $(0, A_\text{tot})$. So, if $MT(E_2 - k_\text{in}) - K_2 k_\text{in} < 0$, we have one positive steady state. This is equivalent to

$$k_\text{in} < \frac{M A_\text{tot} E_2}{M A_\text{tot} + K_2}.$$

If the independent term is positive or zero, then we note that the degree 1 term is also positive. So either $p$ has all coefficients positive and no positive roots, or the leading term is negative, in which case there is one root larger than $A_\text{tot}$. Therefore, no positive steady states in this case.

Case 2: $A_\text{tot} - W > 0$. In this case, we want $A_\text{tot} - W < a_0 < A_\text{tot}$. We find that $p(A_\text{tot} - W)$ is

$$p(A_\text{tot} - W) = -L_1 k_\text{out} \left( W(M - A_\text{tot} + W)(E_2 - k_\text{in}) - K_2 k_\text{in} \right).$$

If this is negative, then there is one positive steady state, as $p$ is positive at $A_\text{tot}$. The condition is

$$k_\text{in} < \frac{W E_2 \left( M - A_\text{tot} + W \right)}{W(M - A_\text{tot} + W) + K_2}.$$

Note that $M + W - A_\text{tot} = a_0 + m_1 + m_0 + a_1 - a_0 - a_1 = m_0 + m_1$, and hence needs to be positive. This is assumed here.

If $p(A_\text{tot} - W) \geq 0$, then we are in the situation where $p$ is nonnegative both at $A_\text{tot} - W$ and $A_\text{tot}$, so, if there are roots in the interval $(A_\text{tot} - W, A_\text{tot})$, there must be two. This contradicts that we already showed that there was at most one. So, in this case, no steady states.

To summarize, there is one positive steady state if and only if $k_\text{in} < E_1$ and

$$k_\text{in} < \frac{M A_\text{tot} E_2}{M A_\text{tot} + K_2} \qquad \text{and} \qquad A_\text{tot} - W \leq 0,$$

or

$$k_\text{in} < \frac{W E_2 \left( M - A_\text{tot} + W \right)}{W(M - A_\text{tot} + W) + K_2}, \qquad A_\text{tot} - W > 0, \qquad \text{and} \qquad M + W - A_\text{tot} > 0.$$

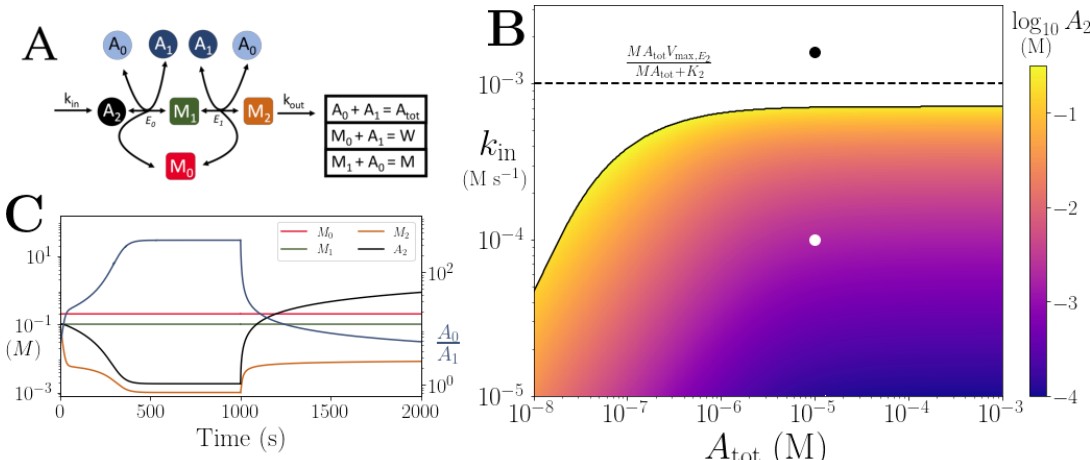

**Appendix 7—figure 1.** Motif, time series and threshold for model with both co-substrate and metabolite cycling, mimicking nitrogen assimilation. (**A**) Reaction system modelled in Appendix 7. (**B**) Heatmap of the steady state $A_2$ concentration. In the white area there is no steady state due to continual build up of $A_2$. Dashed line shows the limit $k_{in} < \frac{MTE_2}{MT+K_2}$, which is the smallest limit for these parameters. (**C**) Time series of $M_{0\rightarrow2}$, $A_2$ and $A_0/A_1$. At $t = 1000$ s, parameters are switched from the white dot in panel (**B**) (where a steady state exists) to the black dot (where there is continual build up of $A_2$). In (**B**) and (**C**) the other parameters are $k_{cat} = 100s^{-1}$, $E_{tot} = 0.01mM$, $K_m = 50\mu M$ and $k_{out} = 0.1s^{-1}$.

## Appendix 8

### Analysis of existing flux data against predicted limits

To support the presented theory, we have analysed existing flux data – compiled from experiments and using flux balance analysis modelling – against predicted limits. The details and main results of this analysis are presented in the main text, under the results and methods sections. Here, we provide further analysis results

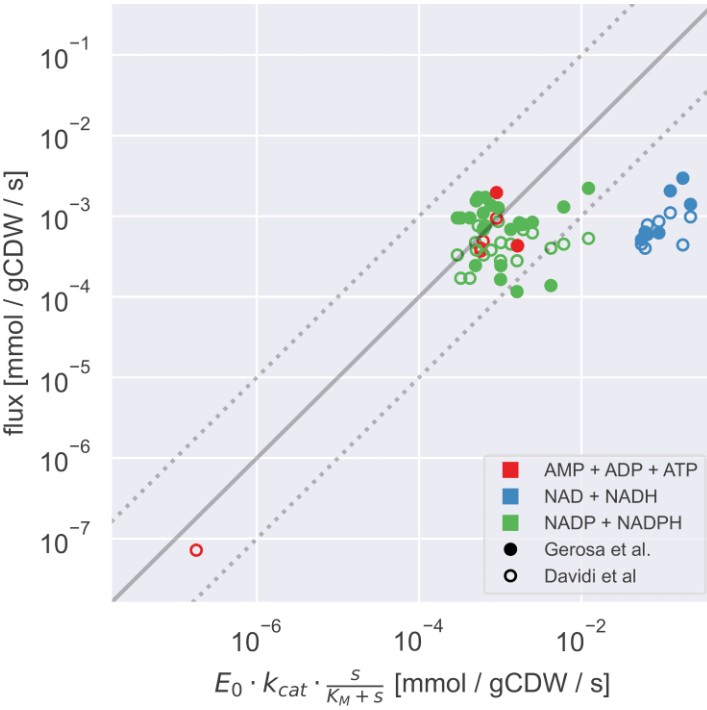

**Appendix 8—figure 1.** Measured flux values (from *Davidi et al., 2016*; *Gerosa et al., 2015*) plotted against the calculated primary enzyme kinetic threshold (first part of *Eq. (1)* of the main text) adjusted by substrate affinity of the enzyme. Note that the flux data shown here is a subset of the flux data presented in (*Figure 3* of the main text), focusing only on those where the main substrate concentration was experimentally measured and the relevant $K_M$ is known. For both panels, the solid line indicates the equivalence of the two values and the dashed lines indicate 10-fold change interval on this, as a guide to the eye. Point colour indicates the nature of co-substrate involved and fill state indicates the data source (as shown on the inset).

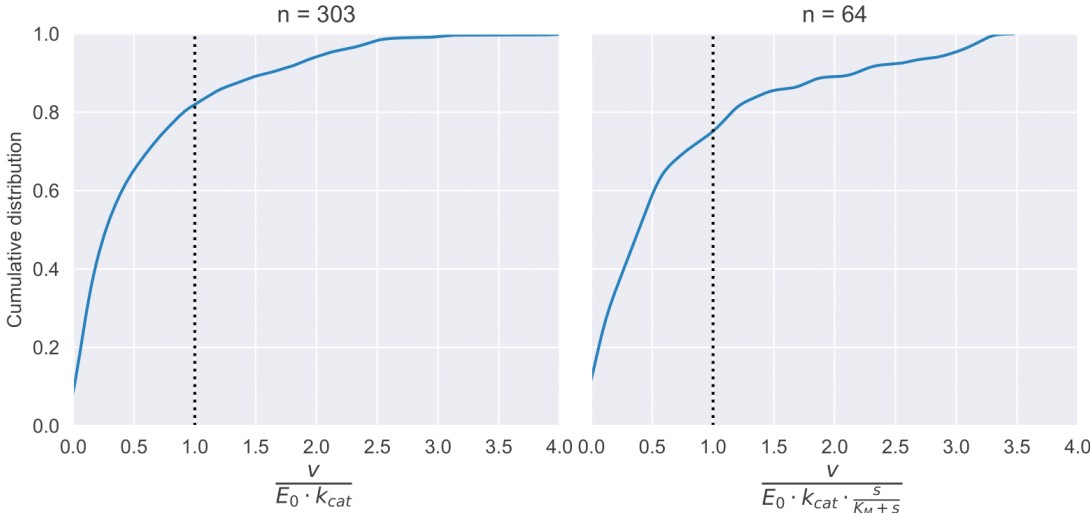

**Appendix 8—figure 2.** Cumulative distribution of the ratio of observed flux (measured or FBA-predicted) to enzyme kinetic based limit ($V_{max}$) (left panel) or to enzyme kinetic based limit accounting for substrate affinity (right panel).

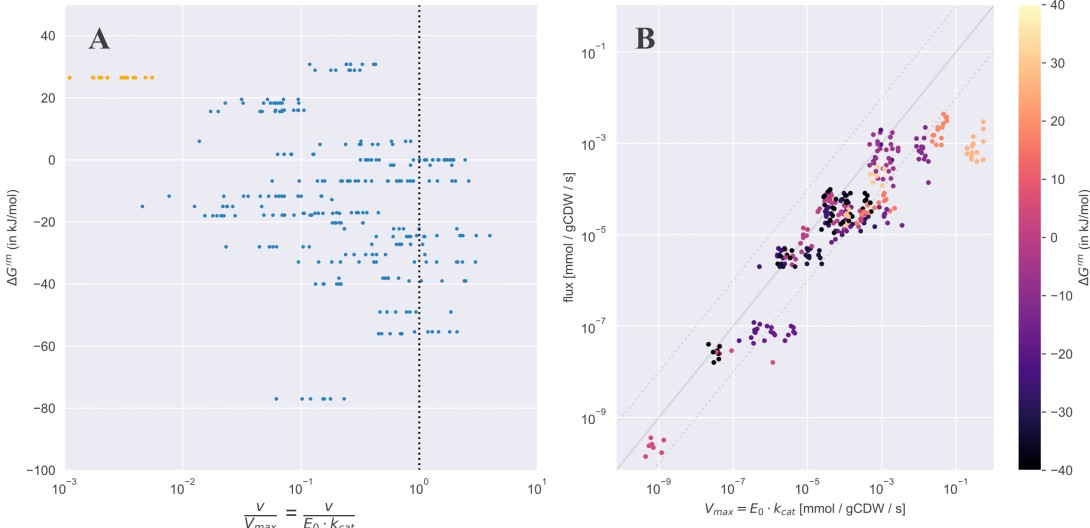

**Appendix 8—figure 3.** Measured flux values (from *Davidi et al., 2016*; *Gerosa et al., 2015*) plotted against $\Delta G'^m$ which is the standard Gibbs energy of the reaction assuming all metabolites are at 1 mM concentrations (note this can be different than the usual definition of $\Delta G'^\circ$ where concentrations are set to 1 M – a concentration that is not reflective of standard physiological conditions). (**A**) shows a scatter plot of the ratio between the observed flux and the enzyme kinetic based limit ($V_{max}$) against the $\Delta G'^m$. The points highlighted in orange are for the malate dehydrogenase reaction. (**B**) A scatter plot of the enzyme kinetic based limit against the measured flux (same data as in *Figure 3* of the main text), where the $\Delta G'^m$ is shown using a colourmap.

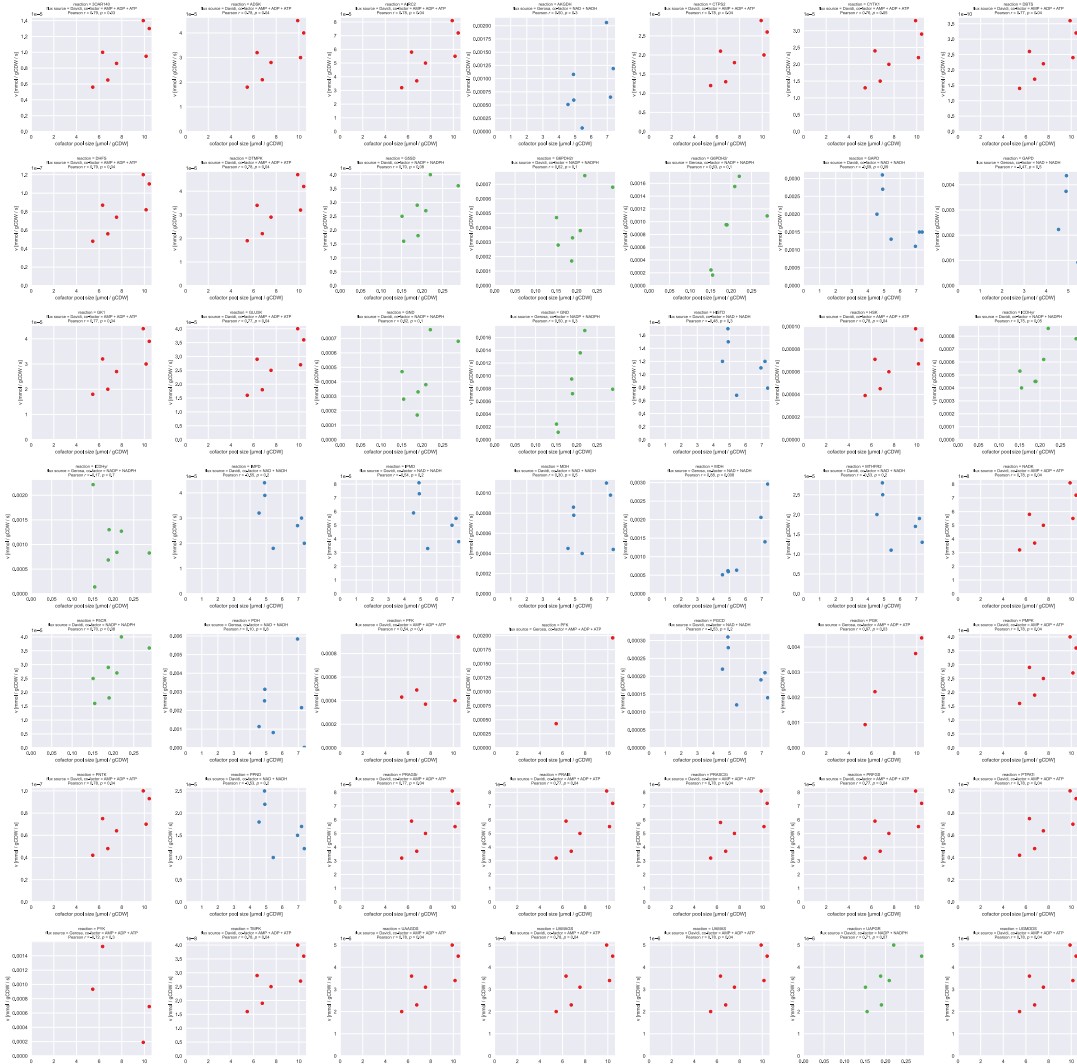

**Appendix 8—figure 4.** Measured flux values under different experimental conditions (from *Davidi et al., 2016*; *Gerosa et al., 2015*) for select reactions plotted against the corresponding co-substrate pool size. See *Supplementary file 1* for short notations for reactions. Each point on each panel is a separate flux measurement under a different environmental condition, where the co-substrate pool size is also measured. Point colours represent co-substrate type, with red for AMP +ADP + ATP, blue for NAD +NADH, and green for NADP +NADPH. Normalised RMSE of the best linear fit and the p-value are shown in the panel title.

**Appendix 8—table 1.** Correlation coefficients between observed flux (measured or FBA-predicted) and co-substrate pool size.
For reaction ID and descriptions, see *Supplementary file 1*.

|  | Reaction | Flux source | Number of conditions | Pearson-r | p-Value |
|---|---|---|---|---|---|
| 25 | MDH | Gerosa | 7 | 0.874310 | 0.010042 |
| 33 | PGK | Gerosa | 4 | 0.973547 | 0.026453 |
| 0 | 3OAR140 | Davidi | 7 | 0.794377 | 0.032850 |
| 7 | DHFS | Davidi | 7 | 0.787719 | 0.035438 |
| 46 | UAMAS | Davidi | 7 | 0.783767 | 0.037025 |

*Appendix 8—table 1 Continued on next page*

*Appendix 8—table 1 Continued*

| | Reaction | Flux source | Number of conditions | Pearson-r | p-Value |
|---|---|---|---|---|---|
| 45 | UAMAGS | Davidi | 7 | 0.783767 | 0.037025 |
| 44 | UAAGDS | Davidi | 7 | 0.783767 | 0.037025 |
| 48 | UGMDDS | Davidi | 7 | 0.783767 | 0.037025 |
| 2 | AIRC2 | Davidi | 7 | 0.783523 | 0.037125 |
| 39 | PRASCSi | Davidi | 7 | 0.783523 | 0.037125 |
| 27 | NADK | Davidi | 7 | 0.783523 | 0.037125 |
| 19 | HSK | Davidi | 7 | 0.783042 | 0.037321 |
| 4 | CTPS2 | Davidi | 7 | 0.782303 | 0.037624 |
| 41 | PTPATi | Davidi | 7 | 0.781795 | 0.037833 |
| 35 | PNTK | Davidi | 7 | 0.781795 | 0.037833 |
| 8 | DTMPK | Davidi | 7 | 0.780177 | 0.038501 |
| 43 | TMPK | Davidi | 7 | 0.777454 | 0.039643 |
| 34 | PMPK | Davidi | 7 | 0.777454 | 0.039643 |
| 1 | ADSK | Davidi | 7 | 0.777418 | 0.039658 |
| 15 | GLU5K | Davidi | 7 | 0.774461 | 0.040918 |
| 37 | PRAGSr | Davidi | 7 | 0.773900 | 0.041159 |
| 40 | PRFGS | Davidi | 7 | 0.773900 | 0.041159 |
| 38 | PRAIS | Davidi | 7 | 0.773900 | 0.041159 |
| 14 | GK1 | Davidi | 7 | 0.772918 | 0.041584 |
| 6 | DBTS | Davidi | 7 | 0.771900 | 0.042026 |
| 5 | CYTK1 | Davidi | 7 | 0.762125 | 0.046410 |
| 20 | ICDHyr | Davidi | 7 | 0.743003 | 0.055681 |
| 47 | UAPGR | Davidi | 7 | 0.702799 | 0.078196 |
| 9 | G5SD | Davidi | 7 | 0.692830 | 0.084415 |
| 28 | P5CR | Davidi | 7 | 0.692830 | 0.084415 |
| 12 | GAPD | Davidi | 7 | –0.681203 | 0.091988 |
| 11 | G6PDH2r | Gerosa | 7 | 0.629982 | 0.129423 |
| 10 | G6PDH2r | Davidi | 7 | 0.617909 | 0.139204 |
| 16 | GND | Davidi | 7 | 0.617909 | 0.139204 |
| 22 | IMPD | Davidi | 7 | –0.539918 | 0.210941 |
| 23 | IPMD | Davidi | 7 | –0.535973 | 0.214953 |
| 36 | PPND | Davidi | 7 | –0.520516 | 0.231016 |
| 26 | MTHFR2 | Davidi | 7 | –0.519049 | 0.232569 |
| 32 | PGCD | Davidi | 7 | –0.517475 | 0.234241 |
| 3 | AKGDH | Gerosa | 7 | 0.498666 | 0.254643 |
| 17 | GND | Gerosa | 7 | 0.498636 | 0.254676 |
| 42 | PYK | Gerosa | 4 | –0.718821 | 0.281179 |
| 18 | HISTD | Davidi | 7 | –0.469323 | 0.288020 |

*Appendix 8—table 1 Continued on next page*

*Appendix 8—table 1 Continued*

|     | Reaction | Flux source | Number of conditions | Pearson-r | p-Value |
| --- | --- | --- | --- | --- | --- |
| 30 | PFK | Davidi | 5 | 0.537260 | 0.350444 |
| 24 | MDH | Davidi | 7 | 0.311471 | 0.496502 |
| 13 | GAPD | Gerosa | 4 | −0.451817 | 0.548183 |
| 21 | ICDHyr | Gerosa | 7 | −0.169510 | 0.716347 |
| 29 | PDH | Gerosa | 7 | 0.098015 | 0.834403 |
| 31 | PFK | Gerosa | 2 | nan | nan |

## Appendix 9

## Pathway branching into two pathways with independent co-substrates

We consider a scenario where two pathways share a common upstream metabolite, as shown in the motif in *Figure 4A* in the main text. Each branch has its own conserved moiety that is cycled and all reactions are reversible. The reaction system is as follows:

$$0 \xrightarrow{k_{in}} M_0 \qquad M_0 \rightleftharpoons M_{1,2} \qquad M_0 \rightleftharpoons M_{1,1}$$
$$M_{1,2} + A_0 \rightleftharpoons M_{2,2} + A_1 \qquad M_{1,1} + B_0 \rightleftharpoons M_{2,1} + B_1 \qquad A_1 \rightleftharpoons A_0 \tag{39}$$
$$B_1 \rightleftharpoons B_0 \qquad M_{2,2} \xrightarrow{k_{out,2}} 0 \qquad M_{2,1} \xrightarrow{k_{out,1}} 0.$$

The ODE system is:

$$\frac{dm_0}{dt} = k_{in} - \frac{E_{1,1}L_0 m_0 - F_{1,1}K_{1,1}m_{1,1}}{K_{1,1}L_0 + K_{1,1}m_{1,1} + L_0 m_0} - \frac{E_{1,2}L_0 m_0 - F_{1,2}K_{1,2}m_{1,2}}{K_{1,2}L_0 + K_{1,2}m_{1,2} + L_0 m_0}$$

$$\frac{dm_{1,2}}{dt} = \frac{E_{1,2}L_0 m_0 - F_{1,2}K_{1,2}m_{1,2}}{K_{1,2}L_0 + K_{1,2}m_{1,2} + L_0 m_0} - \frac{E_{2,2}L_{1,2}m_{1,2}a_0 - F_{2,2}K_{2,2}m_{2,2}a_1}{K_{2,2}L_{1,2} + K_{2,2}m_{2,2}a_1 + L_{1,2}m_{1,2}a_0}$$

$$\frac{dm_{1,1}}{dt} = \frac{E_{1,1}L_0 m_0 - F_{1,1}K_{1,1}m_{1,1}}{K_{1,1}L_0 + K_{1,1}m_{1,1} + L_0 m_0} - \frac{E_{2,1}L_{1,1}m_{1,1}b_0 - F_{2,1}K_{2,1}m_{2,1}b_1}{K_{2,1}L_{1,1} + K_{2,1}m_{2,1}b_1 + L_{1,1}m_{1,1}b_0}$$

$$\frac{dm_{2,2}}{dt} = \frac{E_{2,2}L_{1,2}m_{1,2}a_0 - F_{2,2}K_{2,2}m_{2,2}a_1}{K_{2,2}L_{1,2} + K_{2,2}m_{2,2}a_1 + L_{1,2}m_{1,2}a_0} - m_{2,2}k_{out,2}$$

$$\frac{dm_{2,1}}{dt} = \frac{E_{2,1}L_{1,1}m_{1,1}b_0 - F_{2,1}K_{2,1}m_{2,1}b_1}{K_{2,1}L_{1,1} + K_{2,1}m_{2,1}b_1 + L_{1,1}m_{1,1}b_0} - m_{2,1}k_{out,1} \tag{40}$$

$$\frac{da_0}{dt} = -\frac{E_a L_a a_0 - F_a K_a a_1}{K_a L_a + K_a a_1 + L_a a_0} - \frac{E_{2,2}L_{1,2}m_{1,2}a_0 - F_{2,2}K_{2,2}m_{2,2}a_1}{K_{2,2}L_{1,2} + K_{2,2}m_{2,2}a_1 + L_{1,2}m_{1,2}a_0}$$

$$\frac{da_1}{dt} = \frac{E_a L_a a_0 - F_a K_a a_1}{K_a L_a + K_a a_1 + L_a a_0} + \frac{E_{2,2}L_{1,2}m_{1,2}a_0 - F_{2,2}K_{2,2}m_{2,2}a_1}{K_{2,2}L_{1,2} + K_{2,2}m_{2,2}a_1 + L_{1,2}m_{1,2}a_0}$$

$$\frac{db_0}{dt} = -\frac{E_b L_b b_0 - F_b K_b b_1}{K_b L_b + K_b b_1 + L_b b_0} - \frac{E_{2,1}L_{1,1}m_{1,1}b_0 - F_{2,1}K_{2,1}m_{2,1}b_1}{K_{2,1}L_{1,1} + K_{2,1}m_{2,1}b_1 + L_{1,1}m_{1,1}b_0}$$

$$\frac{db_1}{dt} = \frac{E_b L_b b_0 - F_b K_b b_1}{K_b L_b + K_b b_1 + L_b b_0} + \frac{E_{2,1}L_{1,1}m_{1,1}b_0 - F_{2,1}K_{2,1}m_{2,1}b_1}{K_{2,1}L_{1,1} + K_{2,1}m_{2,1}b_1 + L_{1,1}m_{1,1}b_0}.$$

While it is not possible to solve this system directly, we examine through simulations the effect of varying the influx of the shared upstream metabolite, $k_{in}$ along with the ratio of the pool sizes, $A_{tot}/B_{tot}$ in the main text (*Figure 4* and *Appendix 9—figure 1*), while varying $k_{in}$ with the ratio of moiety back-cycling rates $k_a/k_b$ is presented in *Appendix 9—figure 2*.

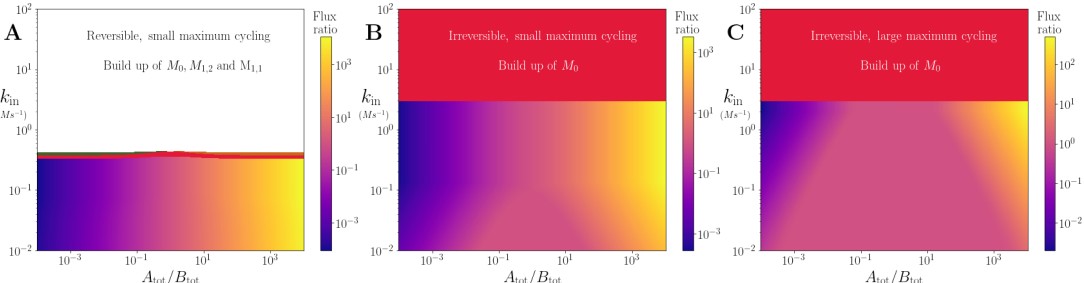

**Appendix 9—figure 1.** Effect of varying $A_{tot}/B_{tot}$ and $k_{in}$ in the branching system shown in *Figure 4A*, on the ratio of flux into $M_{2,2}/M_{2,1}$ for reversible and irreversible dynamics and different $V_{max,A}$ and $V_{max,B}$ values. Downstream metabolite has higher flux when it's respective co-substrate has a higher concentration. (**A**) Reversible reactions with $V_{max,A} = V_{max,B} = 0.1$ (i.e. one order of magnitude smaller than the other reaction rates). (**B**) Irreversible

reactions with $V_{\max,A} = V_{\max,B} = 0.1$ (i.e. one order of magnitude smaller than the other reaction rates). (C) Reversible reactions with $V_{\max,A} = V_{\max,B} = 10$ (i.e. one order of magnitude larger than the other reaction rates).

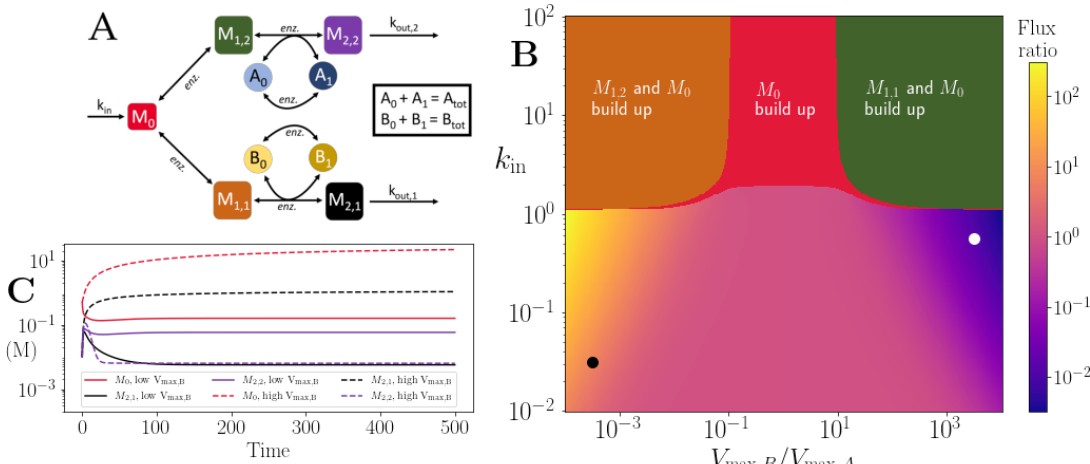

**Appendix 9—figure 2.** Effect of varying $V_{\max,B}/V_{\max,A}$ (i.e maximum back-cycling) ratio and $k_{in}$ in the branching system shown in *Figure 4A*. (**A**) Motif showing the dynamics. (**B**) Heatmap showing the ratio of flux into $M_{2,2}/M_{2,1}$. Downstream metabolite has higher flux when the back-cycling rate of its co-substrate is higher, with a greater effect for higher $k_{in}$ values. (**C**) Time series of downstream metabolites and shared precursor. Solid/dashed lines show parameters for black/white dot in (**B**).

## Appendix 10

### Independent pathways coupled by co-substrate cycling

We consider a scenario with two pathways that when isolated take the form examined in Appendix 3. However, we now suppose that they are coupled by the conserved moiety, that is $A_0 \to A_1$ in the 'forward' direction of the first pathway, while $A_1 \to A_0$ in the 'forward' direction of the other. The metabolite reactions are reversible, and there is pathway-independent cycling of the shared conserved moiety. The motif is shown in *Figure 5A* in the main text, and the reactions are as follows:

$$0 \xrightarrow{k_{\text{in},1}} M_{0,1} \quad M_{0,1} + A_0 \rightleftharpoons M_{1,1} + A_1 \quad M_{1,1} \xrightarrow{k_{\text{out},1}} 0$$
$$0 \xrightarrow{k_{\text{in},2}} M_{0,2} \quad M_{0,2} + A_1 \rightleftharpoons M_{1,2} + A_0 \quad M_{1,2} \xrightarrow{k_{\text{out},2}} 0 \tag{41}$$
$$A_0 \rightleftharpoons A_1.$$

The resulting system of ODEs is:

$$\frac{dm_{0,1}}{dt} = k_{\text{in},1} - \frac{E_{1,1} L_{0,1} m_{0,1} a_0 - F_{1,1} K_{1,1} m_{1,1} a_1}{K_{1,1} L_{0,1} + K_{1,1} m_{1,1} a_1 + L_{0,1} m_{0,1} a_0}$$

$$\frac{dm_{1,1}}{dt} = \frac{E_{1,1} L_{0,1} m_{0,1} a_0 - F_{1,1} K_{1,1} m_{1,1} a_1}{K_{1,1} L_{0,1} + K_{1,1} m_{1,1} a_1 + L_{0,1} m_{0,1} a_0} - k_{\text{out},1} m_{1,1}$$

$$\frac{dm_{0,2}}{dt} = k_{\text{in},2} - \frac{E_{1,2} L_{0,2} m_{0,2} a_1 - F_{1,2} K_{1,2} m_{1,2} a_0}{K_{1,2} L_{0,2} + K_{1,2} m_{1,2} a_0 + L_{0,2} m_{0,2} a_1}$$

$$\frac{dm_{1,2}}{dt} = \frac{E_{1,2} L_{0,2} m_{0,2} a_1 - F_{1,2} K_{1,2} m_{1,2} a_0}{K_{1,2} L_{0,2} + K_{1,2} m_{1,2} a_0 + L_{0,2} m_{0,2} a_1} - k_{\text{out},2} m_{1,2}$$

$$\frac{da_0}{dt} = \frac{E_{1,2} L_{0,2} m_{0,2} a_1 - F_{1,2} K_{1,2} m_{1,2} a_0}{K_{1,2} L_{0,2} + K_{1,2} m_{1,2} a_0 + L_{0,2} m_{0,2} a_1} \tag{42}$$
$$\quad - \frac{E_{1,1} L_{0,1} m_{0,1} a_0 - F_{1,1} K_{1,1} m_{1,1} a_1}{K_{1,1} L_{0,1} + K_{1,1} m_{1,1} a_1 + L_{0,1} m_{0,1} a_0} - \frac{E_a L_a a_0 - F_a K_a a_1}{K_a L_a + K_a a_1 + L_a a_0}$$

$$\frac{da_1}{dt} = - \frac{E_{1,2} L_{0,2} m_{0,2} a_1 - F_{1,2} K_{1,2} m_{1,2} a_0}{K_{1,2} L_{0,2} + K_{1,2} m_{1,2} a_0 + L_{0,2} m_{0,2} a_1}$$
$$\quad + \frac{E_{1,1} L_{0,1} m_{0,1} a_0 - F_{1,1} K_{1,1} m_{1,1} a_1}{K_{1,1} L_{0,1} + K_{1,1} m_{1,1} a_1 + L_{0,1} m_{0,1} a_0} + \frac{E_a L_a a_0 - F_a K_a a_1}{K_a L_a + K_a a_1 + L_a a_0}.$$

This system has one conservation law

$$a_0 + a_1 = A_{\text{tot}},$$

giving that at steady state, $a_1 = A_{\text{tot}} - a_0$. By solving the steady state equations $\frac{dm_{0,1}}{dt} = \frac{dm_{1,1}}{dt} = \frac{dm_{0,2}}{dt} = \frac{dm_{1,2}}{dt} = 0$ for $m_{0,1}, m_{0,2}, m_{1,1}, m_{1,2}$, we obtain

$$m_{0,1} = \frac{K_{1,1} k_{\text{in},1} \left( (F_{1,1} + k_{\text{in},1}) a_1 + L_{0,1} k_{\text{out},1} \right)}{k_{\text{out},1} L_{0,1} a_0 \left( E_{1,1} - k_{\text{in},1} \right)}, \quad m_{1,1} = \frac{k_{\text{in},1}}{k_{\text{out},1}},$$

$$m_{0,2} = \frac{K_{1,2} k_{\text{in},2} \left( (F_{1,2} + k_{\text{in},2}) a_0 + L_{0,2} k_{\text{out},2} \right)}{k_{\text{out},2} L_{0,2} a_1 \left( E_{1,2} - k_{\text{in},2} \right)}, \quad m_{1,2} = \frac{k_{\text{in},2}}{k_{\text{out},2}}.$$

The expressions are positive provided $E_{1,1} > k_{\text{in},1}$, $E_{1,2} > k_{\text{in},2}$, and $a_0, a_1 > 0$.

Finally, we use $\frac{da_0}{dt} = 0$ to solve for $a_0$ and obtain

$$a_0 = \frac{K_a \left((k_{\text{in},2} - k_{\text{in},1})(L_a + A_{\text{tot}}) + A_{\text{tot}}F_a\right)}{(k_{\text{in},2} - k_{\text{in},1})(K_a - L_a) + E_a L_a + F_a K_a}. \tag{43}$$

For all quantities to be positive at steady state, we require $0 < a_0 < A_{\text{tot}}$. By subtracting from $A_{\text{tot}}$ the value of $a_0$ at steady state, we obtain

$$A_{\text{tot}} - a_0 = L_a\left((A_{\text{tot}} + K_a)(k_{\text{in},1} - k_{\text{in},2}) + A_{\text{tot}}E_a\right). \tag{44}$$

To summarize, there is a positive steady state if and only if $E_{1,1} > k_{\text{in},1}$, $E_{1,2} > k_{\text{in},2}$, and (**44**) and (**43**) are positive. For (**44**) to be positive,

$$\frac{A_{\text{tot}}E_a}{A_{\text{tot}} + K_a} > k_{\text{in},2} - k_{\text{in},1}, \qquad \text{that is} \qquad -\frac{A_{\text{tot}}E_a}{A_{\text{tot}} + K_a} < k_{\text{in},1} - k_{\text{in},2}, \tag{45}$$

and either

$$\frac{A_{\text{tot}}F_a}{A_{\text{tot}} + L_a} > k_{\text{in},1} - k_{\text{in},2}, \qquad E_a L_a + F_a K_a > (k_{\text{in},1} - k_{\text{in},2})(K_a - L_a) \tag{46}$$

or

$$\frac{A_{\text{tot}}F_a}{A_{\text{tot}} + L_a} < k_{\text{in},1} - k_{\text{in},2}, \qquad E_a L_a + F_a K_a < (k_{\text{in},1} - k_{\text{in},2})(K_a - L_a) \tag{47}$$

By analysing these cases, we obtain all possible scenarios for a positive steady state to exist, and these are dictated by the difference $k_{\text{in},1} - k_{\text{in},2}$.

For example, if $k_{\text{in},1} = k_{\text{in},2}$, then (**45**) and (**46**) hold directly. If $K_a = L_a$, then we require $\frac{A_{\text{tot}}F_a}{A_{\text{tot}} + L_a} > k_{\text{in},1} - k_{\text{in},2} > -\frac{A_{\text{tot}}E_a}{A_{\text{tot}} + K_a}$ to hold.

If $K_a > L_a$, the conditions lead to either

$$\min\left(\frac{A_{\text{tot}}F_a}{A_{\text{tot}} + L_a}, \frac{E_a L_a + F_a K_a}{(K_a - L_a)}\right) > k_{\text{in},1} - k_{\text{in},2} > -\frac{A_{\text{tot}}E_a}{A_{\text{tot}} + K_a}$$

or

$$\max\left(\frac{A_{\text{tot}}F_a}{A_{\text{tot}} + L_a}, \frac{E_a L_a + F_a K_a}{(K_a - L_a)}, -\frac{A_{\text{tot}}E_a}{A_{\text{tot}} + K_a}\right) < k_{\text{in},1} - k_{\text{in},2}.$$

If $K_a < L_a$, the conditions lead to either

$$\frac{A_{\text{tot}}F_a}{A_{\text{tot}} + L_a} > k_{\text{in},1} - k_{\text{in},2} > \max\left(-\frac{A_{tot}E_a}{A_{\text{tot}} + K_a}, \frac{E_a L_a + F_a K_a}{(K_a - L_a)}\right)$$

or

$$\max\left(\frac{A_{\text{tot}}F_a}{A_{\text{tot}} + L_a}, -\frac{A_{\text{tot}}E_a}{A_{\text{tot}} + K_a}\right) < k_{\text{in},1} - k_{\text{in},2} < \frac{E_a L_a + F_a K_a}{(K_a - L_a)}.$$

A key consequence of these conditions is that coupled pathways can admit higher influx values without upstream metabolite build up, due to the cycling enzyme condition now depending on the difference between the $k_{\text{in}}$ values rather than the values themselves. This occurs when the limit due to the pathway enzyme is larger than that of the cycling enzyme, so there is always a range of $A_{\text{tot}}$ where this applies. An example is shown in *Appendix 10—figure 1*.

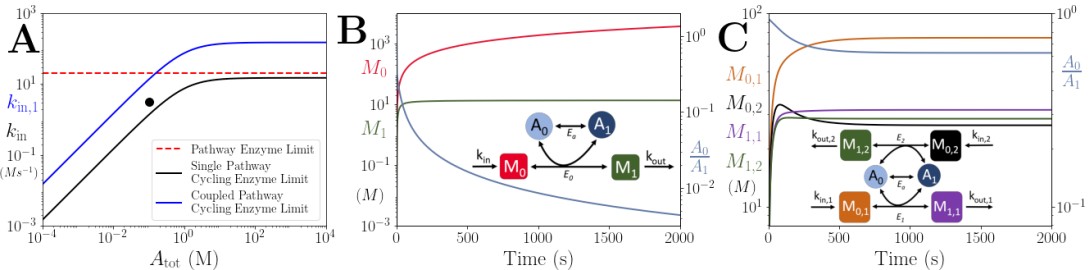

**Appendix 10—figure 1.** Comparison of the onset of instability in the coupled pathway case vs the single pathway case. (**A**) Pathway enzyme and cycling enzyme limits in the single and coupled pathway cases (see legend, pathway enzyme limit is the same for each case). For the coupled pathway, $k_{in,2}$ is set to $0.9k_{in,1}$ Since the limiting factor depends on the difference between $k_{in,1}$ and $k_{in,2}$, higher $k_{in}$ values are possible without build up of upstream metabolites if all other parameters are kept the same. (**B**) Time series of $M_0$, $M_1$ and $A_0/A_1$ in the single pathway case where the parameters for $A_{tot}$ and $k_{in}$ are set by the black dot in (**A**). Here we see build up $M_0$ because $k_{in} > A_{tot}F_a/(A_{tot} + L_a)$. (**C**) Time series of $M_{0,1}$, $M_{0,2}$, $M_{1,1}$, $M_{1,2}$ and $A_0/A_1$ in the coupled pathway case where the parameters for $A_{tot}$, $k_{in,1}$ and $k_{in,2}$ are set by the black dot in (**A**). Here we see the system admits a steady state because $k_{in,1} - k_{in,2} < A_{tot}F_a/(A_{tot} + L_a)$ The parameters in all panels are: $\frac{A_{tot}F_a}{A_{tot}L_a} > k_{in,1} - k_{in,2} > \max\left(-\frac{A_{tot}E_a}{A_{tot}K_a}, \frac{E_aL_a+F_aK_a}{(K_a-L_a)}\right), K_1 = L_0 = K_{1,1} = K_{1,2} = L_{0,1} = L_{0,2} = 1\ M$ ,$k_{out} = k_{out,1} = k_{out,2} = 0.1\ s^{-1}$, $E_a = F_a = 15\ Ms^{-1}$. $K_a = L_a = 1\ M$.

To complement this steady state analysis with dynamics of this system, we use numerical simulations to study the system. In particular, we consider the effect of randomly fluctuating influxes on the downstream metabolites. The analysis is achieved by fixing the average of the influxes, while drawing the log-ratio (*i.e.* $\log_{10} k_{in,1}/k_{in,2}$) from a standard normal distribution with mean $\mu = 0$ and variance $\sigma^2 = 1$. A new log-ratio is drawn after waiting a time that is drawn from an exponential distribution with mean $\tau$. Example time-series are shown in *Figure 5(C and D)* of the main text. The log-ratio is chosen as the variable instead of simply the ratio as it allows us to examine large variations, while keeping the effect on each pathway symmetric. This random process can be thought of as the discrete-time analogue of the Ornstein-Uhlenbeck process: it has the same steady state distribution, mean, variance and correlation function as its continuous-time counterpart, while being much easier and faster to implement as part of a larger system where the other equations are deterministic.

The effect of different values of $\tau$ and total pool size $A_{tot}$ is shown in *Figure 5* of the main text, and is further explored in *Appendix 10—figure 2*. These analyses show that the effect of increasing the pathway independent moiety cycling rate is to reduce the critical pool size above which the downstream metabolites are anti-correlated. Furthermore, this behaviour is observed whether the metabolite reactions are irreversible or reversible. Since the transition from correlated to anti-correlated is so sharp, relatively small changes in the total amount of cycled co-substrate, combined with noisy influx rates, could lead to the downstream metabolites changing from being correlated to anti-correlated, or vice-versa.

*Appendix 10—figure 2* also further demonstrates the effect of changing $E_a$, $\tau$ and $A_{tot}$ on the time series of the products $M_{1,1}$ and $M_{1,2}$ and the co-substrate ratio $A_0/A_1$. For parameters represented by the black dot in the first row (panels (**E**) - (**H**)), the products are correlated when $E_a \leq 1$, and become uncorrelated when $E_a = 10$. The system is still responsive to the noise, but with less variation compared to the other rows because it is limited by the low $A_{tot}$. For parameters represented by the grey dot in the first row (panels (**I**) - (**L**)), the products switch from correlated to anti-correlated as $E_a$ increases, and remain responsive to the noise. The variation in $A_0/A_1$ reduces from left to right because $E_a$ increases. For parameters represented by the white dot in the first row (panels (**M**) - (**P**)), the products remain anti-correlated and responsive to the noise. The variation in $A_0/A_1$ reduces from left to right but is not as pronounced for small $E_a$ because $A_{tot}$ is very large.

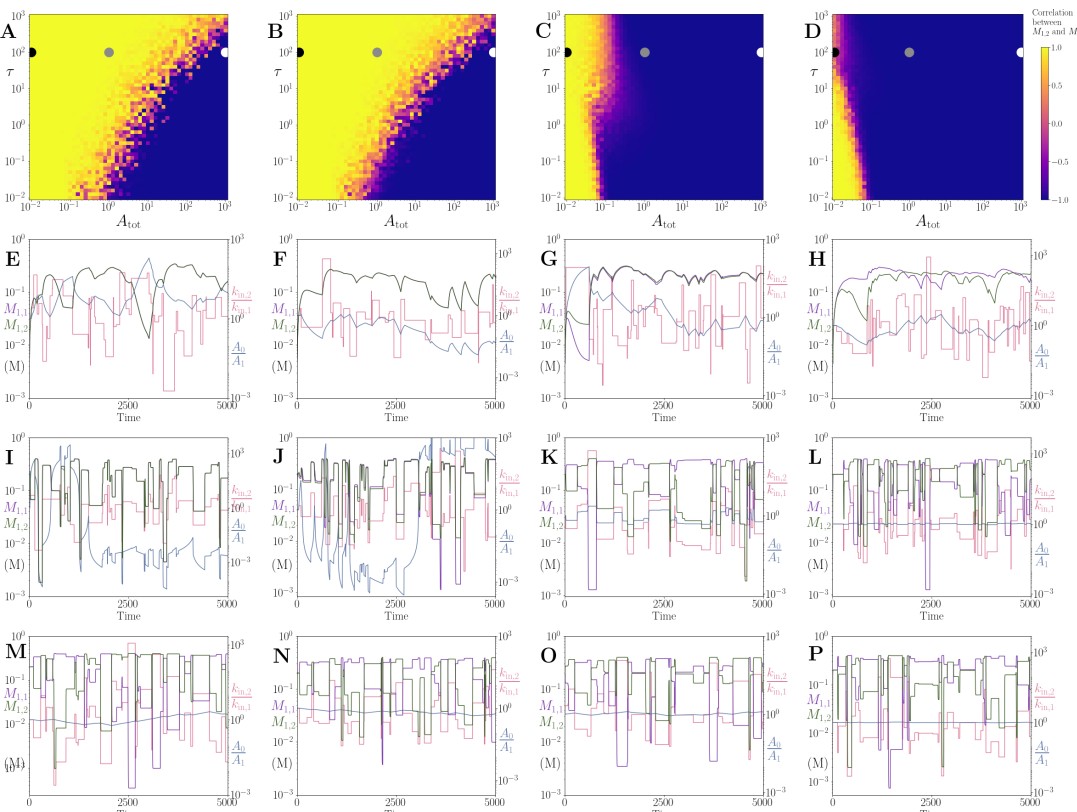

**Appendix 10—figure 2.** Correlation between products of two coupled pathways, using a model with irreversible reaction rates. (**A**) - (**D**) show results for $E_a = F_a = 0, 0.01, 1$ and $10$, respectively. As in the *Figure 5* of the main text, all other parameters are set to 1, apart from the $k_{out}$ values that are each 0.5. As $E_a$ increases, the products become anti-correlated at smaller $A_{tot}$ values. This is also true for the model using reversible reaction rates (see *Figure 5* of the main text). (**E**) - (**P**) show time series for the products $M_{1,1}$ and $M_{1,2}$ (purple and green respectively) on the left axis and the ratios $A_0/A_1$ and $k_{in,2}/k_{in,1}$ (blue and pink respectively) on the right axis. Panels in the same column share the same $E_a$ value as the heatmap in the top (*e.g.* $E_a = 0$ in (**E**), (**I**) and (**M**)). Panels (**E**) - (**H**), (**I**) - (**L**) and (**M**) - (**P**) have $A_{tot}$ and $\tau$ given by the black, grey and white dots respectively. See text.

## Appendix 11

### Table of parameters

**Appendix 11—table 1.** List of parameters together with the biological meanings and units. $M$ and $t$ stand for Molar and time, respectively. Where different notation is used between this and the main text, the equivalent parameters have been made clear.

| Parameter | Biological meaning and notes | Units |
|---|---|---|
| $E_i$, $F_i$ | Catalytic rate coefficients for forward and backward reactions respectively. In the main text we only use the forward rate and denote it $V_{\mathrm{max},E_i}$. | $Mt^{-1}$ |
| $K_i$, $L_{i-1}$ | Michaelis-Menten coefficients for forward and backward reactions, respectively. In the main text, we only use the forward rate and denote it $K_{M,E_i}$. | $M^n$, where $n$ is the number of different substrates involved in the reaction. |
| $k_{\mathrm{in}}$ | Influx rate. | $Mt^{-1}$ |
| $k_{\mathrm{out}}$ | Outflow rate. | $t^{-1}$ |

