## [Editor Report]

This manuscript presents an important mathematical analysis of metabolic "co-substrates" and how their cycling can affect metabolic fluxes. Through mathematical analysis of simple network motifs, it shows the impact of co-substrate cycling on constraining metabolic fluxes. The combination of mathematical modeling and comparisons with existing data from previous studies offers convincing support for the potential biological relevance of co-substrate cycling. The work will be of interest to researchers who study microbial metabolism and metabolic engineering.

---

## [Decision Letter]

**Decision letter after peer review:**

Thank you for submitting your article "Dynamics of co-substrate pools can constrain and regulate metabolic fluxes" for consideration by *eLife*. Your article has been reviewed by 3 peer reviewers, including Babak Momeni as Reviewing Editor and Reviewer #1, and the evaluation has been overseen by Aleksandra Walczak as the Senior Editor. The following individuals involved in review of your submission have agreed to reveal their identity: Jonas Cremer (Reviewer #2); Silvio Waschina (Reviewer #3).

Essential revisions:

1) Please include the terminology and basic concepts in the introduction to improve the accessibility of the paper.

2) Please revise the manuscript to explicitly include the relevant model assumptions and details of the setups (see individual reviewer comments).

3) Please further address the limitations of the study in the Discussion section (see comments from Reviewer #3).

*Reviewer #1 (Recommendations for the authors):*

1. Although the definition of cycling is somewhat intuitive, I do not think the formal definition offered in lines 84 and 85 is accurate. I'd suggest revising it (e.g. with A -> B -> C and D -> B -> E, B is produced and consumed via different reactions, but with no cycling).

2. One of the main theses in this manuscript is that limitations by conserved pools in cycles will impact the overall metabolite dynamics. It would be nice to have an estimate of how often cycles will impose a limitation and where they should be expected in the metabolic network. Figure 3 and associated supplementary figures perhaps contain this information, but if possible, a more explicit estimate would be informative.

3. In Figure 3B-D, I do not think that the correct model for the relation between the pool size and the normalized flux is a linear dependency (presumably when the pool size is large enough, the flux will saturate). Should this be taken into account instead of a regression line (and RMSE) for representing whether the model fits the existing data?

4. On lines 245-245, I think the second hypothesis should be phrased more explicitly. It is not immediately clear what "balancing" means.

5. In Figure 5, a simpler configuration can be imagined in which there is no direct reaction between A0 and A1. Presumably such a configuration creates an even stronger dependency between the two pathways through cycling A0 to A1 and vice versa. I am curious why the authors did not choose such a configuration. Alternatively, with the A0 <-> A1 reaction included, what is the impact of the rate of this reaction on the correlation between the two pathways?

*Reviewer #2 (Recommendations for the authors):*

– Overall, the paper is well written and I was able to follow. However, it also assumes that readers are familiar with a lot of different terminology/examples of metabolic analysis. Below are some remarks that the authors might want to consider. For example, I think the introduction would benefit from a more detailed introduction of terms. For example, a clearer definition of co-substrates in the introduction would be helpful. Similarly, a better introduction of the term flux would help and yield is not introduced such that it remains unclear for readers to see what the tradeoffs are.

– line 46 onwards: What is the evidence? Can a sentence or two be added to clarify?

– line 84 onwards: It seems this paragraph is still part of the introduction.

– Too early reference to Figure 1A (is it Figure S1A). Better to show a summary figure illustrating the different co-substrates mentioned?

– line 101 onwards: paragraph could be better embedded and it is currently hard to follow what the authors want to stress when the following discussion is primarily about steady state.

– line 113 onwards: I think it would be helpful to spell out in the main text what α is.

– line 180 onwards: I am missing background to understand the statement. It was always my impression that it is in general very hard to reliably measure reaction rates in-vivo (k_in_ and V_max_), so how can one establish that k_in_ < V_max_?

– ll 187 onwards: some essential details are missing to understand the type of dataset used (see above).

– It is also important to introduce in more detail the different experimental conditions for which the variation in ATP precursors has been observed. Could it be that ATP levels are merely changing because of the way the conditions change and not because fluxes vary as well?

*Reviewer #3 (Recommendations for the authors):*

Mathematical equations, transformations, resulting constraints, and conclusions are generally very clearly explained, which allows readers, who might be less-accustomed to kinetic modeling, to follow the line of arguments. One suggestion I have is to provide units of variables that are used in equations also in the text, or perhaps as a (supplementary) table. Specifically, I noticed that this could help readers already at equations (1). Here, the steady-state concentration of m1 is given by the ratio of the two rate parameters k[in] and k[out]. Since in the text, both parameters are only referred to as "flux rates", one could assume that both have the same unit, which would make their ratio (=m1 concentration) unit-less. Only with the units that appear later in figure 2 (k[in] in the figure itself; k[out] in the caption) it becomes clear that the two parameters have different units.

In addition, I understand that the authors claim, that flux-governing and -regulation through means of co-substrate cycling could be a general 'design principle' that is not limited to a specific group of organisms. However, I would suggest to provide more detail on the experimental/FBA data (i.e. fluxes, kinetic parameters obtained from BRENDA) that is used for the comparison with the model predictions. For instance, details on the organisms behind the experimental data that is mentioned e.g. in lines 187-208 and in Figure 3 could be added. This would help readers to put the authors' results in more biological context. Along those lines, the discussion could include a paragraph that addresses current limitations to further evaluate the developed mathematical framework. For instance, from which organisms do we have experimental data to study the potential role of co-substrate cycling using the developed framework? What future experiments/data types will further help to test the mathematical framework?

---

## [Author Response]

Essential revisions:1) Please include the terminology and basic concepts in the introduction to improve the accessibility of the paper.

This point is addressed. See revised text and also reply to recommendation 1 from reviewer 2, and recommendation 1 from reviewer 3.

2) Please revise the manuscript to explicitly include the relevant model assumptions and details of the setups (see individual reviewer comments).

This point is addressed. See revised text and specific replies to recommendations 4-6 from reviewer 1, recommendations 1 and 2 from reviewer 3, and recommendations 2, 7-9 from reviewer 2.

3) Please further address the limitations of the study in the Discussion section (see comments from Reviewer #3).

This point is addressed. See revised text and specific replies to reviewer 3 and the replies to general summary by reviewer 2 and their recommendation 9.

Reviewer #1 (Recommendations for the authors):1. Although the definition of cycling is somewhat intuitive, I do not think the formal definition offered in lines 84 and 85 is accurate. I'd suggest revising it (e.g. with A -> B -> C and D -> B -> E, B is produced and consumed via different reactions, but with no cycling).

We thank the reviewer for this comment. We have revised the definition, as:

“Certain pairs of metabolites can be interconverted via different reactions in the cell, thereby resulting in their ‘cycling’.”

2. One of the main theses in this manuscript is that limitations by conserved pools in cycles will impact the overall metabolite dynamics. It would be nice to have an estimate of how often cycles will impose a limitation and where they should be expected in the metabolic network. Figure 3 and associated supplementary figures perhaps contain this information, but if possible, a more explicit estimate would be informative.

We thank the reviewer for this suggestion. We explicitly stated that 80% of the analysed co-substrate linked reactions are possibly constrained by co-substrate dynamics. We have further highlighted three specific reactions in *E. coli*, where flux seems linked to co-substrate pool size. Further predictions would require more data, specifically for co-substrate related reactions. We also note that our analytical solutions provide an inequality of system parameters, which when satisfied will indicate flux limitation due to co-substrate cycling. These parameters, especially enzyme levels and co-substrate pool size will be condition dependent.

We have now included elements of this reply in the revised manuscript. See also reply to 2nd recommendation by reviewer 3.

3. In Figure 3B-D, I do not think that the correct model for the relation between the pool size and the normalized flux is a linear dependency (presumably when the pool size is large enough, the flux will saturate). Should this be taken into account instead of a regression line (and RMSE) for representing whether the model fits the existing data?

We agree with the reviewer that the dependency between the maximal flux and the pool size is not a linear relationship, but a saturating one as shown in Figure 1. We would expect this relation to hold if the only controlled parameter was the pool size (*A_tot_*) while the rest of the system is unchanged. In Figure 3B-D, we compare data points of the measured reaction flux at different environmental condition (carbon sources) which are likely to affect other system parameters in Equation 1. Therefore, it is difficult to describe the relationship quantitatively and we thus chose to look at correlations as a proxy. See also our replies to public review and recommendation 9 by reviewer 2.

4. On lines 245-245, I think the second hypothesis should be phrased more explicitly. It is not immediately clear what "balancing" means.

We thank the reviewer for this comment. We have now replaced ‘balancing’ with ‘coupling’ and revised this section to read;

“We hypothesised that such inter-pathway co-substrate cycling might cause; (1) the co-substrate related limit to relate to difference in pathway influxes, rather than the influx into one pathway, and (2) a coupling of the pathway output fluxes against influx fluctuations, such that the output fluxes remain correlated to each other, despite differences in influx levels.”

5. In Figure 5, a simpler configuration can be imagined in which there is no direct reaction between A0 and A1. Presumably such a configuration creates an even stronger dependency between the two pathways through cycling A0 to A1 and vice versa. I am curious why the authors did not choose such a configuration. Alternatively, with the A0 <-> A1 reaction included, what is the impact of the rate of this reaction on the correlation between the two pathways?

We thank the reviewer for this comment. We had already analysed the effect of the rate of the A_0_ to A_1_ reaction, which we called "pathway-independent turnover of the co-substrate” and labelled its rate as *V_max,Ea_*. This analysis had not considered the reaction rate being zero, i.e. no pathway-independent turnover. We have now extended this analysis to include this case, and also included a section in the results, summarising the findings from it (see Appendix – Figure 13).

In brief, we had shown in Figure 5B that the pathway outputs are coupled (i.e. correlated) for a range of pool sizes (*A_tot_*) despite fluctuations in their inputs. We also found that as *A_tot_* gets smaller, the two pathway outputs shift from being highly anti-correlated (blue region in Figure 5B) to being highly correlated (yellow region in Figure 5B). This 'correlated outputs regime' is further split into two sub-regimes, where the pathways are `responsive' or `unresponsive' to fluctuations in the inputs, as shown in Figure 5C/D.

We found that an increase in the rate of the pathway-independent turnover (*V_max,Ea_*) causes a reduction in the *A_tot_* value for which a transition to the highly correlated regime happens (see updated Appendix – Figure 13). The case for *V_max,Ea_* = 0 remains qualitatively the same as analysed in Figure 5. We have also checked that the systems remain 'responsive' to changes in the input as *V_max,Ea_* increase (Appendix – Figure 13).

We have now included these results in the main text in the corresponding section.

Reviewer #2 (Recommendations for the authors):– Overall, the paper is well written and I was able to follow. However, it also assumes that readers are familiar with a lot of different terminology/examples of metabolic analysis. Below are some remarks that the authors might want to consider. For example, I think the introduction would benefit from a more detailed introduction of terms. For example, a clearer definition of co-substrates in the introduction would be helpful. Similarly, a better introduction of the term flux would help and yield is not introduced such that it remains unclear for readers to see what the tradeoffs are.

We thank the reviewer for this useful comment. We have now revised the introduction section to better introduce the relevant terms. See also reply to second recommendation of reviewer 1.

– line 46 onwards: What is the evidence? Can a sentence or two be added to clarify?

We thank the reviewer for this useful comment. We have now expanded this section to read:

“There are, however, increasing number of studies suggesting that enzyme levels alone might not be sufficient to explain observed flux levels. For example, in Davidi et al. 2016, the maximal value of the apparent activities (k_app,max_) of an enzyme, derived using measured enzyme levels and fluxes under different conditions, was a good estimate for the specific activity of that enzyme in vitro (k_cat_). However, individual estimates from each condition (i.e. individual k_app_ values) were commonly lower than the specific activity – suggesting that the flux is limited by something other than enzyme levels under those conditions. Other studies have shown that metabolic flux changes, caused by perturbations in media conditions, are not explained solely by changes in expression levels of enzymes (Chubukov 2013, Gerosa 2015).”

– line 84 onwards: It seems this paragraph is still part of the introduction.

While the reviewer has a good point, we opted to keep this section within the results, as it relates to several figures in the Appendix that exemplify cases of co-substrate related cyclic reaction motifs in metabolism. We have, however, streamlined the two sections presented here into a single section. See also our reply to recommendation 5 below.

– Too early reference to Figure 1A (is it Figure S1A). Better to show a summary figure illustrating the different co-substrates mentioned?

We agree with the reviewer and have removed this reference to Figure 1A at the end of the first paragraph. See also next reply.

– line 101 onwards: paragraph could be better embedded and it is currently hard to follow what the authors want to stress when the following discussion is primarily about steady state.

We thank the reviewer for this comment. In light of it, we have revised this section and combined it with the previous section (the title of which is changed to: Co-substrate cycling represents a ubiquitous motif in metabolism with co-substrate pools acting as ‘conserved moieties’).

The point we wanted to highlight is that the pool size can still be defined by a constant, even under consideration of biosynthesis and uptake. We revised this section to read:

“We notice here that many of the co-substrate involving, cycling reactions can be abstracted as a simplified motif as shown in Figure 1A. This abstract representation highlights the fact that the total pool-size involving all the different forms of a cycled metabolite can become a conserved quantity. This would be the case even when we consider biosynthesis or environmental uptake of co-substrates, as the total concentration of a cycled metabolite across its different forms at steady state would then be given by a constant defined by the ratio of the influx and outflux rates (see Appendix, section 2 and 3). In other words, the cycled metabolite would become a ‘conserved moiety’ for the rest of the metabolic system and can have a constant ‘pool size’. Supporting this, temporal measurement of specific co-substrate pool sizes shows that ATP and GTP pools are constant under stable metabolic conditions, but can rapidly change in response to external perturbations, possibly through inter-conversions among pools rather than through biosynthesis (Walther 2010).”

– line 113 onwards: I think it would be helpful to spell out in the main text what α is.

We have now done so for the case presented in the main text (the irreversible Michealis-Menten model). For other models considered, we referred to the relevant Appendix section.

– line 180 onwards: I am missing background to understand the statement. It was always my impression that it is in general very hard to reliably measure reaction rates in-vivo (k_in_ and V_max_), so how can one establish that k_in_ < V_max_?

We thank the reviewer for this comment, which highlights that this section is not written clearly. The utilised experimental / FBA-based data does not include *k_in_* but rather the flux of select reactions. The value for *V_max_* is not measured, but rather calculated using in vitro enzyme kinetics measurements (for *k_cat_*) and measured expression level of enzymes (i.e. *E_tot_* from proteomics datasets). While there is no direct measurement of *k_in_*, the measured flux can be used as proxy for it, since the two will be equal at steady state. The fact that there is no observed accumulation of metabolites in the measured reactions, suggest that the analytically derived inequality holds and the influx into, and across, each reaction is below the analytically obtained limits. We have now revised this section as follows, to make this point more clear:

“Based on flux values that are either experimentally measured or predicted by flux balance analysis (FBA), many reactions from the central carbon metabolism of the model organism *Escherichia coli* are shown to have lower flux than expected from the kinetics of their immediate enzymes (i.e. V_max_) [Davidi et al. (2016)]. This finding is based on calculating V_max_ from in vitro measured k_cat_ values of specific enzymes and their in vivo levels based on proteomics studies in *E. coli* (see Methods). The flux and enzyme concentration data were from other studies which measured them during the exponential phase in *E. coli*, growing on minimal media supplemented with various carbon sources [Schmidt et al. (2016); Gerosa et al. (2015)]. If we consider measured fluxes for each reaction as a proxy for k_in_ (notice that these two would be equal at steady state), we can conclude from the fact that there were no observed substrate accumulation in these reactions, as the analysed reactions carrying fluxes below the first limit identified above in Equation 2. There could be several explanations for this observation of measured fluxes being lower than the limit set by measured enzyme kinetics and level. One simple explanation could be that there is a discrepancy between in vitro measured enzyme kinetics and in vivo realised ones. Alternatively, this discrepancy can be low, but the lower flux could be arising because there are additional limiting factors other than the enzymes mediating the main reaction. Among such additional limiting factors, substrate limitation and thermodynamic effects are shown to partially explain observed lower fluxes in some reactions [Davidi et al. (2016)]. Here, we highlight that the presented theory shows that an additional possible limitation could be the co-substrate pool size and turnover dynamics.”

– ll 187 onwards: some essential details are missing to understand the type of dataset used (see above).

We agree with the reviewer and added the relevant details to the text (see the answer to previous comment).

– It is also important to introduce in more detail the different experimental conditions for which the variation in ATP precursors has been observed. Could it be that ATP levels are merely changing because of the way the conditions change and not because fluxes vary as well?

We are not clear what the reviewer is suggesting about ATP precursors. We don't assume anything regarding changes to ATP precursors, but only use the measured values to show that there is a change in total ATP pool with changing conditions. We do not speculate about the reasons for this, however, we note that the changes in the total ATP pool size cannot be due to flux changes in reactions using the ATP pool members as co-substrates. The changes in pool size, however, can affect flux in those reactions, where co-substrate dynamics is limiting (as predicted by the theory). If this was the case, then we would expect a change in flux in those reactions with changing pool size. This is what we wanted to explore / show with Figure 3B-D. Indeed, there seems to be some reactions where this is the case. We also recognise that we cannot rule out other effects arising from pool size changes.

We have now included elements of this response in the revised text to make this point clear. (Regarding the experimental details, we believe they are covered by the answer to recommendation 7, above.)

Reviewer #3 (Recommendations for the authors):Mathematical equations, transformations, resulting constraints, and conclusions are generally very clearly explained, which allows readers, who might be less-accustomed to kinetic modeling, to follow the line of arguments. One suggestion I have is to provide units of variables that are used in equations also in the text, or perhaps as a (supplementary) table. Specifically, I noticed that this could help readers already at equations (1). Here, the steady-state concentration of m1 is given by the ratio of the two rate parameters k[in] and k[out]. Since in the text, both parameters are only referred to as "flux rates", one could assume that both have the same unit, which would make their ratio (=m1 concentration) unit-less. Only with the units that appear later in figure 2 (k[in] in the figure itself; k[out] in the caption) it becomes clear that the two parameters have different units.

We thank the reviewer for this useful comment. We have now included a table, Appendix – Table 2, listing the units of all the parameters, and referred to it in the Methods section of the main text.

In addition, I understand that the authors claim, that flux-governing and -regulation through means of co-substrate cycling could be a general 'design principle' that is not limited to a specific group of organisms. However, I would suggest to provide more detail on the experimental/FBA data (i.e. fluxes, kinetic parameters obtained from BRENDA) that is used for the comparison with the model predictions. For instance, details on the organisms behind the experimental data that is mentioned e.g. in lines 187-208 and in Figure 3 could be added. This would help readers to put the authors' results in more biological context. Along those lines, the discussion could include a paragraph that addresses current limitations to further evaluate the developed mathematical framework. For instance, from which organisms do we have experimental data to study the potential role of co-substrate cycling using the developed framework? What future experiments/data types will further help to test the mathematical framework?

We thank the reviewer for these useful comments. The source organisms for the data used was the model bacterium *Escherichia coli*. We have now included this information both in the *Methods* and the *Results section*s. See also reply to recommendation 7 by reviewer 2.

Additionally, we have included the following paragraph in the Discussion section to make the current limitations in the available data clear, and to highlight future experimental avenues possible:

“Comparing measured flux data against estimated flux values based on measured enzyme levels from proteomics and enzyme kinetics from in vitro studies, we have provided support that fluxes in co-substrate linked reactions could indeed by limited by co-substrate pool dynamics under physiological conditions. This analysis was based on the model organism *E. coli* and is limited even for this organism due to limited flux and proteomics data. For example, the data compiled here contained 14 co-substrate reactions with experimentally measured fluxes, but only half of these could be used due to lack of measurement on enzyme concentrations. We hope that the presented theory will provide motivation to further expand the available data sets, especially for reactions relating to co-substrate linked reactions. In this quest, we expect that the expansion of measurements to eukaryotic cells to be particularly challenging due to organelle-specific pools, but some progress is being made to achieve at least mitochondrial and cytosolic measurements [Chen et al. 2016]. Despite the current limitations, our data-based analyses highlighted three key reactions mediated by phosphoglycerate kinase (PGK), malate dehydrogenase (MDH), and glucose-6-phosphate dehydrogenase (G6PDH) and linking to ATP, NADH, and NADPH pools. Possible flux limitation of these reactions by co-substrate dynamics can also be subjected to further experimental study – as we discuss further below.”